# Rethinking Visual Counterfactual Explanations Through Region Constraint

**Bartlomiej Sobieski** *
University of Warsaw
b.sobieski@uw.edu.pl

**Jakub Grzywaczewski**
Warsaw University of Technology
jakub.grzywaczewski2.stud@pw.edu.pl

**Bartlomiej Sadlej**
University of Warsaw
b.sadlej@student.uw.edu.pl

**Matthew Tivnan**
Harvard Medical School
mtivnan@mgh.harvard.edu

**Przemyslaw Biecek**
University of Warsaw, Warsaw University of Technology
przemyslaw.biecek@pw.edu.pl

## Abstract

Visual counterfactual explanations (VCEs) have recently gained immense popularity as a tool for clarifying the decision-making process of image classifiers. This trend is largely motivated by what these explanations promise to deliver – indicate semantically meaningful factors that change the classifier's decision. However, we argue that current state-of-the-art approaches lack a crucial component – the *region constraint* – whose absence prevents from drawing explicit conclusions, and may even lead to faulty reasoning due to phenomenons like confirmation bias. To address the issue of previous methods, which modify images in a very entangled and widely dispersed manner, we propose *region-constrained* VCEs (RVCEs), which assume that only a predefined image region can be modified to influence the model's prediction. To effectively sample from this subclass of VCEs, we propose *Region-Constrained Counterfactual Schrödinger Bridges* (RCSB), an adaptation of a tractable subclass of Schrödinger Bridges to the problem of conditional inpainting, where the conditioning signal originates from the classifier of interest. In addition to setting a new state-of-the-art by a large margin, we extend RCSB to allow for *exact* counterfactual reasoning, where the predefined region contains *only* the factor of interest, and incorporating the user to actively interact with the RVCE by predefining the regions manually.

## 1 Introduction

Visual counterfactual explanations (VCEs) aim at explaining the decision-making process of an image classifier by modifying the input image in a semantically meaningful and minimal way so that its decision changes. Over time, they have become an independent research direction with the latest methods presenting impressive and visually appealing results. Nevertheless, in this work we show that they possess a fundamental flaw at a conceptual level – the lack of *region constraint* and its proper utilization.

Consider the image $\mathbf{x}^*$ in Fig. 1, which the classifier $f$ correctly predicts to be a jay. In essence, VCEs focus on semantically editing $\mathbf{x}^*$ so that the prediction of $f$ changes to some target class – bulbul in this case – hence providing an answer to a specific *what-if* question, through which the model's reasoning is explained. Consider now an example VCE for $\mathbf{x}^*$, denoted as $\mathbf{x}_{\text{VCE}}$, obtained with a recent state-of-the-art (SOTA) method. While $\mathbf{x}_{\text{VCE}}$ is successful at changing the prediction of $f$ and can be considered both realistic and semantically close to $\mathbf{x}^*$, answering *why* $f$ now predicts it as a bulbul is close to impossible.

---

*Corresponding author

The algorithm simultaneously modifies the bird's head and feathers, changes the texture of the branch and even modifies the copyright caption. The entanglement and dispersion of introduced changes hence leaves the question unanswered. We argue that to circumvent these fundamental difficulties, VCEs should be synthesized with a hard constraint on the *region*, where the changes are allowed to appear, while leaving the rest of the image unchanged. For example, consider the image $\mathbf{x}_{\mathbf{R}}^*$ with regions of the bird's head ($\mathbf{R}_1$) and body ($\mathbf{R}_2$) overlayed. Constraining the VCEs to introduce changes *only* to predetermined regions leads to two distinct explanations, $\mathbf{x}_{\mathbf{R}_1}$ and $\mathbf{x}_{\mathbf{R}_2}$, of why the decision changes to bulbul. By isolating the modified factors, the explanatory process greatly simplifies – one can now state with certainty that $f$'s new prediction is based either on the modified feathers ($\mathbf{x}_{\mathbf{R}_2}$) or the changed characteristics of its head ($\mathbf{x}_{\mathbf{R}_1}$). Region-constrained VCEs (RVCEs) allow, therefore, to reason about the model's thought process in a *causal* and principled manner, mitigating the potential *confirmation bias* and clarifying the explanatory process.

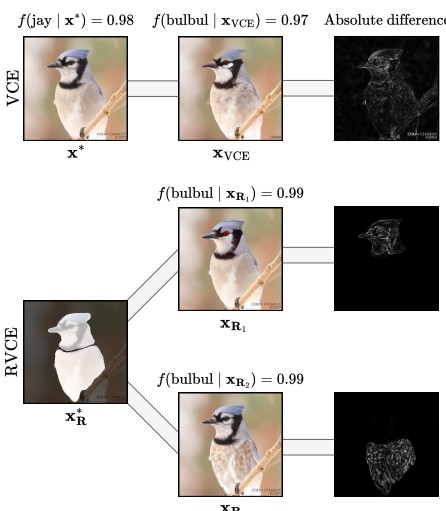

Figure 1: Previous methods create VCEs with unconstrained changes, making it virtually impossible to understand the decision-making process of a model. We propose *region-constrained* VCEs, establishing a new paradigm for comprehensible and actionable explanatory process.

By putting RVCEs in the spotlight, our work establishes new frontiers in the field of VCE generation. First, we define the objective of finding RVCEs as solving a conditional inpainting task. By building on top of the Image-to-Image Schrödinger Bridge (I²SB, Liu et al. (2023a)) approach and adapting it to the classifier guidance scheme, we develop an efficient algorithm which synthesizes RVCEs with extreme realism, sparsity and closeness to the original image. Specifically, we set a new quantitative state-of-the-art (SOTA) on ImageNet (Deng et al., 2009) with up to 4 times better scores in FID and 3 times better sFID (realism), up to 2 times higher COUT (sparsity), and match or exceed $S^3$ (similarity) and Flip Rate (efficiency) achieved by previous methods. Through large-scale experiments, we demonstrate that, besides a fully automated way of synthesizing meaningful and highly interpretable RVCEs, our approach, *Region-constrained Counterfactual Schrödinger Bridge* (RCSB), allows to infer causally about the model's change in prediction and enables the user to actively interact with the explanatory process by manually defining the region of interest. Moreover, our results highlight the importance of RVCEs in future research, indicating potential pitfalls of unconstrained methods that could lead to drawing misleading conclusions.

## 2 BACKGROUND & RELATED WORK

In this section, we introduce the necessary background knowledge connected with score-based generative models (SGMs) and I²SB, which forms the foundation of our method. We then present an overview of recent methods for VCE generation based on SGMs. For an extended literature review and detailed description of the theoretical basis, please refer to the Appendix.

**SGM.** Following the work of Song et al. (2021), SGMs can be constructed through the framework of stochastic differential equations (SDEs), where samples from a complex distribution $p_0$ (*e.g.*, natural images) are mapped to a Gaussian distribution $p_1$, while the model is trained to reverse this mapping. Formally, converting data to noise is performed by following the *forward* SDE (Eq. (1a)), while denoising happens through the *reverse* SDE (Eq. (1b), Anderson (1982)):

$$d\mathbf{x}_t = \mathbf{F}_t(\mathbf{x_t})dt + \sqrt{\beta_t}d\mathbf{w}, \tag{1a}$$

$$d\mathbf{x}_t = (\mathbf{F}_t(\mathbf{x_t}) - \beta_t\nabla_{\mathbf{x}_t}\log p(\mathbf{x}_t,t))dt + \sqrt{\beta_t}d\bar{\mathbf{w}}, \tag{1b}$$

where $\mathbf{x}_t$ is the noisy version of a clean image $\mathbf{x} \in \mathbb{R}^n$ for some $n \in \mathbb{N}$ at timestep $t \in [0, 1]$, $\mathbf{w}$ and $\bar{\mathbf{w}}$ denote the Wiener process and its reversed (in time) counterpart, $\mathbf{F}_t(\mathbf{x}_t) : \mathbb{R}^n \to \mathbb{R}^n$ is

the *drift* coefficient, $\beta_t \in \mathbb{R}$ is the *diffusion* coefficient and $\nabla_{\mathbf{x}_t} \log p(\mathbf{x}_t, t)$ is the *score function*. An SGM $\mathbf{s}_{\boldsymbol{\theta}}$, where $\boldsymbol{\theta}$ denotes the model's parameters, is trained to approximate the score, *i.e.*, $\mathbf{s}_{\boldsymbol{\theta}}(\mathbf{x}_t, t) \approx \nabla_{\mathbf{x}_t} \log p(\mathbf{x}_t, t)$. During sampling, denoising begins from pure noise $\mathbf{x}_1 \sim p_1$ and follows some discretized version of Eq. (1b) with the approximate score $\mathbf{s}_{\boldsymbol{\theta}}$.

SGMs can also be adapted to *conditional* generation, where $\mathbf{y}$ represents the conditioning variable. In this case, the score $\nabla_{\mathbf{x}_t} \log p(\mathbf{x}_t, t)$ is replaced by $\nabla_{\mathbf{x}_t} \log p(\mathbf{x}_t, t \mid \mathbf{y})$, which can be decomposed with Bayes' Theorem into $\nabla_{\mathbf{x}_t} \log p(\mathbf{x}_t, t \mid \mathbf{y}) = \nabla_{\mathbf{x}_t} \log p(\mathbf{x}_t, t) + \nabla_{\mathbf{x}_t} \log p(\mathbf{y} \mid \mathbf{x}_t, t)$. While $\nabla_{\mathbf{x}_t} \log p(\mathbf{x}_t, t)$ can be approximated with an already trained $\mathbf{s}_{\boldsymbol{\theta}}$, $\nabla_{\mathbf{x}_t} \log p(\mathbf{y} \mid \mathbf{x}_t, t)$ must be modeled additionally. For $\mathbf{y}$ representing class labels, $p(\mathbf{y} \mid \mathbf{x}_t, t)$ can be approximated with an auxiliary time-dependent classifier $p_{\boldsymbol{\phi}}(\mathbf{y} \mid \mathbf{x}_t, t)$ trained on noisy images $\{\mathbf{x}_t\}_{t \in [0,1]}$. Incorporating $p_{\boldsymbol{\phi}}$ into the sampling process is termed as *classifier guidance* (CG), and can be strengthened (or weakened) with *guidance scale* $s$ through $\nabla_{\mathbf{x}_t} \log p(\mathbf{x}_t, t) + s \cdot \nabla_{\mathbf{x}_t} \log p(\mathbf{y} \mid \mathbf{x}_t, t)$. Therefore, class-conditional sampling in SGMs amounts to additionally maximizing the likelihood $p_{\boldsymbol{\phi}}(y \mid \mathbf{x}_t, t)$ of the classifier throughout the generative process to arrive at images from the data manifold, which resemble (according to $p_{\boldsymbol{\phi}}$) instances of a specific class. We emphasize this fact here for further reference.

**I²SB.** The framework of I²SB extends SGMs to $p_1$ representing an *arbitrary* data distribution. For training, I²SB requires paired data, *e.g.*, in the form of clean and partially masked samples for inpainting, where it learns to infill the missing parts. While SGMs can also be adapted to solve inverse problems like inpainting, I²SB maps these samples *directly* (see Fig. 2 for a comparison of their generative trajectories). Therefore, I²SB follows the same theoretical paradigm, where sampling is achieved by discretizing

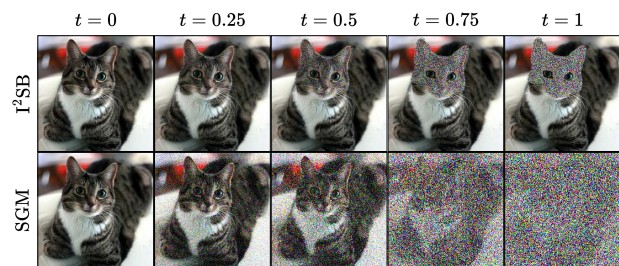

Figure 2: Generative trajectories of I²SB and SGM. Intermediate images of I²SB are much closer to the data manifold.

Eq. (1b) and using a score approximator $\mathbf{s}_{\boldsymbol{\psi}}$, but the generative process begins from a corrupted (*e.g.*, masked) image instead of pure noise. Hence, I²SB can also be adapted to conditional generation in the same manner as SGMs, especially for class-conditioning with an auxiliary classifier. Importantly, a special case of I²SB follows an optimal transport ordinary differential equation (OT-ODE) when $\beta_t \to 0$, eliminating stochasticity beyond the initial sampling step (see Appendix). We utilize the OT-ODE version of I²SB in our implementation.

**SGM-based VCEs.** The initial approach of adapting SGMs to VCE generation, DiME (Jeanneret et al., 2022), obtains the classifier's gradient by mapping the noised image to its clean version at each step through the reverse process. Augustin et al. (2022) (DVCE) incorporate the gradient of a robust classifier and a cone projection scheme. Jeanneret et al. (2023) (ACE) decompose the VCE generation into pre-explanation construction and refinement using RePaint (Lugmayr et al., 2022). Jeanneret et al. (2024) utilize a foundation model, Stable Diffusion (SD, Rombach et al. (2022)), to generate VCEs in a black-box scenario. Farid et al. (2023) (LDCE) and Motzkus et al. (2024) utilize Latent Diffusion Models (LDMs), including SD, in a white-box context. Weng et al. (2024) propose FastDiME to accelerate the generation process in a shortcut learning scenario. Also in black-box context, Sobieski & Biecek (2024) utilize a Diffusion Autoencoder (Preechakul et al., 2022) to find semantic latent directions that globally flip the classifier's decision. Augustin et al. (2024) also make use of SD in various contexts, including classifier disagreement and neuron activation besides VCEs. While ACE and FastDiME also assume constraints to some regions, those are always classifier-dependent. In this work, we consider a more general definition provided in the next section.

## 3 METHOD

In this section, we describe the details of our approach, beginning with the formulation of RVCEs as solutions to conditional inpainting task. Next, we motivate the use of I²SB as an effective prior for synthesizing meaningful RVCEs and follow with a series of steps that better align the gradients of a standard classifier w.r.t corrupted images from its generative trajectory. We conclude with a description of the automated region extraction method, forming the basis of our algorithm.

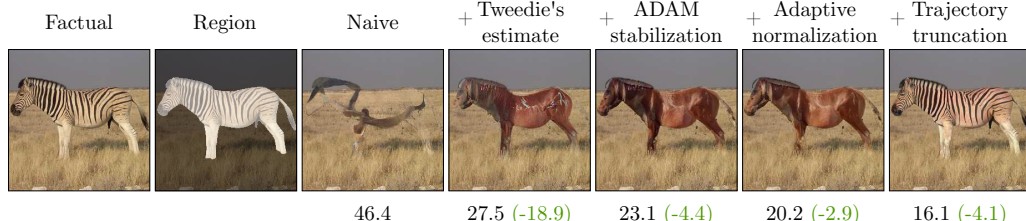

| Factual | Region | Naive | + Tweedie's estimate | + ADAM stabilization | + Adaptive normalization | + Trajectory truncation |
|---------|--------|-------|----------------------|----------------------|--------------------------|-------------------------|
| | | 46.4 | 27.5 (-18.9) | 23.1 (-4.4) | 20.2 (-2.9) | 16.1 (-4.1) |

Figure 3: Series of proposed improvements to better align the gradient's of the classifier of interest with the generative trajectory. Changes to the *factual* image are constrained to the indicated *region*. Subsequent images illustrate the influence of each new adaptation. Numbers below images correspond to FID ($\downarrow$) values obtained in a larger-scale experiment (for details, see Appendix).

**RVCEs through conditional inpainting.** We define the problem of finding RVCEs for the classifier $f$ from a given image $\mathbf{x}^*$, a region $\mathbf{R}$ and *target class* label $y$, where $\arg\max_{y'} f(y' \mid \mathbf{x}^*) \neq y$, as the task of sampling from

$$p(\mathbf{x} \mid \arg\max_{y'} f(y' \mid \mathbf{x}) = y, (\mathbf{1} - \mathbf{R}) \odot \mathbf{x} = (\mathbf{1} - \mathbf{R}) \odot \mathbf{x}^*), \tag{2}$$

where $\mathbf{R}$ is a binary mask with 1 indicating the region which is not restricted to depend on the classifier $f$. Intuitively, sampling from Eq. (2) means obtaining $\mathbf{x}$ with the complement of $\mathbf{R}$ unchanged and the content of $\mathbf{R}$ modified in a way that changes the decision of $f$ to $y$, *i.e.*, performing inpainting with additional condition coming from the classifier $f$.

**Synthesizing meaningful RVCEs.** Looking at Eq. (2), one quickly realizes that obtaining semantically meaningful RVCEs requires maximizing the likelihood $f(y \mid \mathbf{x})$ of the classifier while inpainting $\mathbf{R}$ with content that keeps $\mathbf{x}$ in the data manifold. These conditions greatly resemble the CG scheme in the context of I$^2$SB, since the score estimate $\mathbf{s}_\psi$ serves as an effective prior for generating in-manifold infills, while the likelihood $p_\phi(y \mid \mathbf{x})$ of an auxiliary classifier is maximized to ensure that $p_\phi$ predicts them as instances of $y$. Moreover, I$^2$SB maps masked images *directly* to clean samples, leaving the content outside $\mathbf{R}$ unchanged in the final image.

The above arguments suggest that inserting $f$ in place of $p_\phi$ should function as an effective mechanism for sampling meaningful RVCEs. However, a fundamental drawback of this *naive* approach is that, throughout the generative process, $f$'s gradients originate from evaluating it on images with highly noised infills inside $\mathbf{R}$ (see Fig. 2). Such corrupted images are far from what $f$ observed during training, hence leading to a *misalignment* of its gradients with the correct trajectory and generation of out-of-manifold samples. Similar issue has been identified by previously mentioned SGM-based methods for VCEs, which can be generally unified as attempts to replace the auxiliary classifier $p_\phi$ with $f$ in the CG scheme in SGMs and *correct* $f$'s gradients. Following Fig. 2, one should expect the misalignment in these methods to be of great extent, as the generative trajectory consists of highly noised images, leaving no meaningful content for $f$ to provide accurate gradients. There, as shown in Fig. 2, I$^2$SB provides a crucial advantage, which stems from its generative trajectory being *much closer* to the data manifold. Moreover, by using I$^2$SB, $f$ is able to effectively utilize the readily available context outside $\mathbf{R}$. Hence, in the following, we focus on reducing the misalignment problem caused by the noised content inside $\mathbf{R}$, in the end arriving at a highly effective algorithm for meaningful RVCEs.

**Aligning the gradients.** We propose to adapt the gradients of $f$ to properly align with the generative trajectory of I$^2$SB through a series of incremental steps. To provide the intuition standing behind the introduction of each consecutive improvement, Fig. 3 provides an example RVCE task, where the factual image depicts a zebra correctly predicted by the model (ResNet50 (He et al., 2016)), and the goal is to change the decision to 'sorrel'. We set the region constraint to include the entire animal to make the task challenging enough and verify the improvements quantitatively through a large-scale experiment with around 2000 images. For each step, we compute FID between the RVCEs and original images to assess their realism. For details on the experimental setup, see Appendix.

**Naive.** We first verify that naively plugging $f$ in place of $p_\phi$ does not provide meaningful results. Indeed, as shown in Fig. 3, the method struggles to include the information from $f$. The unrealistic infill also suggests that the classifier's signal negatively influences the score from I$^2$SB.

**Tweedie's formula.** To begin with closing the gap between the data manifold and the generative trajectory, we refer to a classic result of Tweedie's formula (Robbins, 1992; Chung et al., 2022; Weng et al., 2024), which states that a *denoised estimate* of the final image at step $t$ can be achieved by computing the posterior expectation

$$\hat{\mathbf{x}}_0(\mathbf{x}_t) := \mathbb{E}[\mathbf{x}_0 \mid \mathbf{x}_t] = \mathbf{x}_t + \sigma_t^2 \nabla_{\mathbf{x}_t} \log p(\mathbf{x}_t, t), \tag{3}$$

where $\sigma_t^2 = \int_0^t \beta_\tau \mathrm{d}\tau$. For visual differences between $\mathbf{x}_t$ and $\hat{\mathbf{x}}_0(\mathbf{x}_t)$, see Appendix. Crucially, one has access to approximate $\hat{\mathbf{x}}_0(\mathbf{x}_t)$ at every step $t$ by utilizing I$^2$SB as the approximate score. Replacing $\nabla_{\mathbf{x}_t} \log f(y \mid \mathbf{x}_t)$ with $\nabla_{\mathbf{x}_t} \log f(y \mid \hat{\mathbf{x}}_0(\mathbf{x}_t))$ brings the inputs of $f$ much closer to what it expects, improving the conditional inpainting process as indicated by Fig. 3, which now shows a structure resembling a sorrel and a much smaller FID.

**ADAM stabilization.** Despite utilizing the Tweedie's estimate, we observed the norms of $f$'s gradients to have a very noisy tendency throughout the generation process, pointing out a possible cause for visible artifacts and the missing parts of the animal. Hence, we propose to smooth out the gradients by applying the ADAM update rule at each step (Vaeth et al., 2024; Kingma, 2019), to which we simply refer as ADAM stabilization. Figure 3 indicates that this modification allows for filling in the missing parts of the sorrel and further lowering FID.

**Adaptive normalization.** Incorporating ADAM stabilization required greatly lowering the guidance scale to values on the order of $1\mathrm{e}{-2}$, as using standard $s = 1$ led to extreme artifacts. This phenomenon suggested that the step size could also be adjusted throughout the generation process. While we initially experimented with various types of schedulers (see Appendix), using *adaptive normalization* has empirically proven to be the most effective approach. Specifically, at the beginning of the conditional inpainting process, we register the norm of the first encountered gradient of the log-likelihood of $f$. We then use it as a normalizing constant for each subsequent gradient, meaning that the generation begins with gradient of unit norm. This simple modification not only further lowered FID, but also reduced the final visible artifacts and improved color balance (Fig. 3).

**Trajectory truncation.** Up until this point, we relied solely on the ability of I$^2$SB and the classifier's signal to correctly infill the missing regions with semantically meaningful content, with no knowledge of the structure of the missing objects. Since a possible infill of the region is always available from the original image, one can begin the inpainting process from some *intermediate* step instead of the final one. This intervention allows for mixing the available information with the one coming from the classifier, and gives direct control over the preservation of the original content. As our approach does not bias the conditional score with signal from any additional losses (like Learned Perceptual Image Patch Similarity (LPIPS Zhang et al. (2018)) or $l_2$ in other works), we can fully rely on the *conceptual* compression of I$^2$SB, similarly to SGMs (Ho et al., 2020), which decomposes the generation process into initial phases responsible for the overall structure of objects and later ones responsible for small details. Figure 3 showcases the effect of using this *trajectory truncation* ($\tau$) at the $0.4$ level, meaning that the infilling process starts from $t = \tau \cdot T$, where $T$ denotes the final timestep.

Understandably, trajectory truncation greatly lowers the FID score, as much more information is available from the very beginning of the process, and introduces much more subtle changes to the image. We explore the effect of manipulating $\tau$ further in the Appendix, showing that it functions as a very interpretable mechanism for controlling the content preservation.

**Automated region extraction.** While the introduced algorithmical improvements effectively incorporate the classifier's signal into the inpainting process, they do not address the issue of predetermining the region for the resulting explanation. To this end, the optimal strategy would be fully automated and focus on regions that are both important to the classifier's prediction and point to semantically meaningful concepts. This description closely resembles the

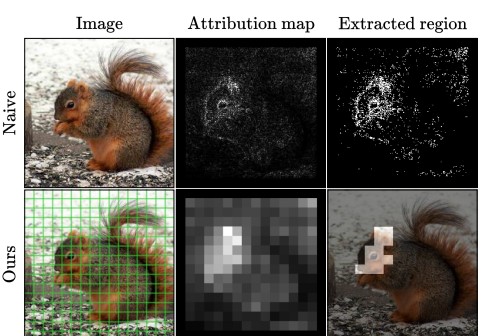

Figure 4: Example region obtained with our automated region extraction. Instead of directly binarizing an attribution map (upper row), we amplify the focus on semantic concepts (bottom row) with a simple approach based on grid cells.

role of visual attribution methods, which assign importance values to pixels based on their relevance to the model's output (Holzinger et al., 2022). Figure 4 shows an example attribution map obtained with Integrated Gradients (IG, Sundararajan et al. (2017)) method for the squirrel prediction of a ResNet50 model. Perceptually, highest attributions are focused around the squirrel's head. To extract a region from such attributions, one can threshold them to cover a specific fraction $a$ of the total image area. However, after binarizing the attributions with $a = 0.05$, we observe that the resulting region is highly scattered, losing focus from semantic concepts. To address this issue, we divide the image into a grid of square cells of size $c \times c$, where each cell receives the value equal to the sum of the absolute pixel attributions inside it. Figure 4 shows that this postprocessing mechanism (here with $c = 16$) greatly amplifies the focus of the resulting map. By thresholding it with $a = 0.05$, we observe the extracted region to focus solely on the squirrel's head. This leads to a fully automated strategy for obtaining regions that are both aligned with semantically meaningful concepts and based on pixels that are important for the classifier.

We term the final version of the algorithm which combines all of the aforementioned improvements with the automated region extraction as RCSB. For the pseudocode of the entire procedure, see Appendix. We include our implementation at https://github.com/sobieskibj/rcsb.

## 4 EXPERIMENTS

Following previous works for VCEs on ImageNet, we base the quantitative evaluation on 3 challenging **main** VCE generation tasks: **Zebra – Sorrel**, **Cheetah – Cougar**, **Egyptian Cat – Persian Cat**, where each task requires creating VCEs for images from both classes and flipping the decision to their counterparts. We treat it as a general benchmark for evaluating the effectiveness of RCSB in various scenarios. We use FID ($\downarrow$) and sFID ($\downarrow$) to assess realism (Heusel et al., 2017), $S^3$ ($\uparrow$) for representation similarity (Chen & He, 2021), COUT $\in [-1, 1]$ ($\uparrow$) (Khorram & Fuxin, 2022) for sparsity and Flip Rate (FR) ($\uparrow$) for efficiency. For qualitative examples, we extend the main tasks with a large array of **other** tasks, which we show throughout the paper and the Appendix, where more details regarding the experimental setup and the metrics description can be found.

**RCSB sets new SOTA for VCEs.** We first verify that synthesizing RVCEs with RCSB leads to new SOTA in VCE generation. Table 1 quantitatively compares RCSB with recent SOTA approaches to VCEs on ImageNet. Our RVCEs are much more realistic (at least $2 - 4\times$ decrease in FID and sFID), stay close to original images (match or exceed best values of $S^3$) and almost always flip the model's decision (FR $\approx 1.0$). RCSB also solves a long-standing challenge of achieving extremely *sparse* explanations on ImageNet, especially on **Zebra – Sorrel** task. While all other methods fail to

| Method | FID | sFID | $S^3$ | COUT | FR |
|---|---|---|---|---|---|
| **Zebra – Sorrel** | | | | | |
| ACE $l_1$ | 84.5 | 122.7 | **0.92** | $-0.45$ | 47.0 |
| ACE $l_2$ | 67.7 | 98.4 | 0.90 | $-0.25$ | 81.0 |
| LDCE-cls | 84.2 | 107.2 | 0.78 | $-0.06$ | 88.0 |
| LDCE-txt | 82.4 | 107.2 | 0.71 | $-0.21$ | 81.0 |
| DVCE | 33.1 | 43.9 | 0.62 | $-0.21$ | 57.8 |
| RCSB$^C$ | 13.0 | 20.4 | 0.82 | 0.70 | **99.7** |
| RCSB$^B$ | 9.51 | 17.4 | 0.86 | 0.72 | 97.4 |
| RCSB$^A$ | **8.0** | **16.2** | 0.88 | **0.74** | 94.7 |
| **Cheetah – Cougar** | | | | | |
| ACE $l_1$ | 70.2 | 100.5 | 0.91 | 0.02 | 77.0 |
| ACE $l_2$ | 74.1 | 102.5 | 0.88 | 0.12 | 95.0 |
| LDCE-cls | 71.0 | 91.8 | 0.62 | 0.51 | 100.0 |
| LDCE-txt | 91.2 | 117.0 | 0.59 | 0.34 | 98.0 |
| DVCE | 46.9 | 54.1 | 0.70 | 0.49 | 99.0 |
| RCSB$^C$ | 30.2 | 39.2 | 0.87 | 0.79 | 100.0 |
| RCSB$^B$ | 23.4 | 32.4 | 0.90 | 0.85 | 99.9 |
| RCSB$^A$ | **17.2** | **26.6** | **0.92** | **0.92** | **100.0** |
| **Egyptian Cat – Persian Cat** | | | | | |
| ACE $l_1$ | 93.6 | 156.7 | 0.85 | 0.25 | 85.0 |
| ACE $l_2$ | 107.3 | 160.4 | 0.78 | 0.34 | 97.0 |
| LDCE-cls | 102.7 | 140.7 | 0.63 | 0.52 | 99.0 |
| LDCE-txt | 121.7 | 162.4 | 0.61 | 0.56 | 99.0 |
| DVCE | 46.6 | 59.2 | 0.59 | 0.60 | 98.5 |
| RCSB$^C$ | 41.1 | 56.3 | 0.79 | 0.82 | 100.0 |
| RCSB$^B$ | 31.3 | 48.1 | 0.84 | 0.87 | 100.0 |
| RCSB$^A$ | **23.0** | **40.0** | **0.87** | **0.92** | **100.0** |

Table 1: Quantitative comparison with SOTA. RCSB outperforms previous methods by a large margin across all metrics. The best results are obtained with $A(a = 0.1, c = 4, s = 3, \tau = 0.6)$, but the superiority is clear for various configurations, including $B(a = 0.2, c = 4, s = 1.5, \tau = 0.6)$, $C(a = 0.3, c = 4, s = 1.5, \tau = 0.6)$.

achieve nonnegative values, RCSB approaches the upper bound of COUT. Our method is clearly the most balanced, as it does not struggle on any specific metric like, *e.g.*, DVCE on $S^3$. In the Appendix, we show that it is also the most computationally efficient.

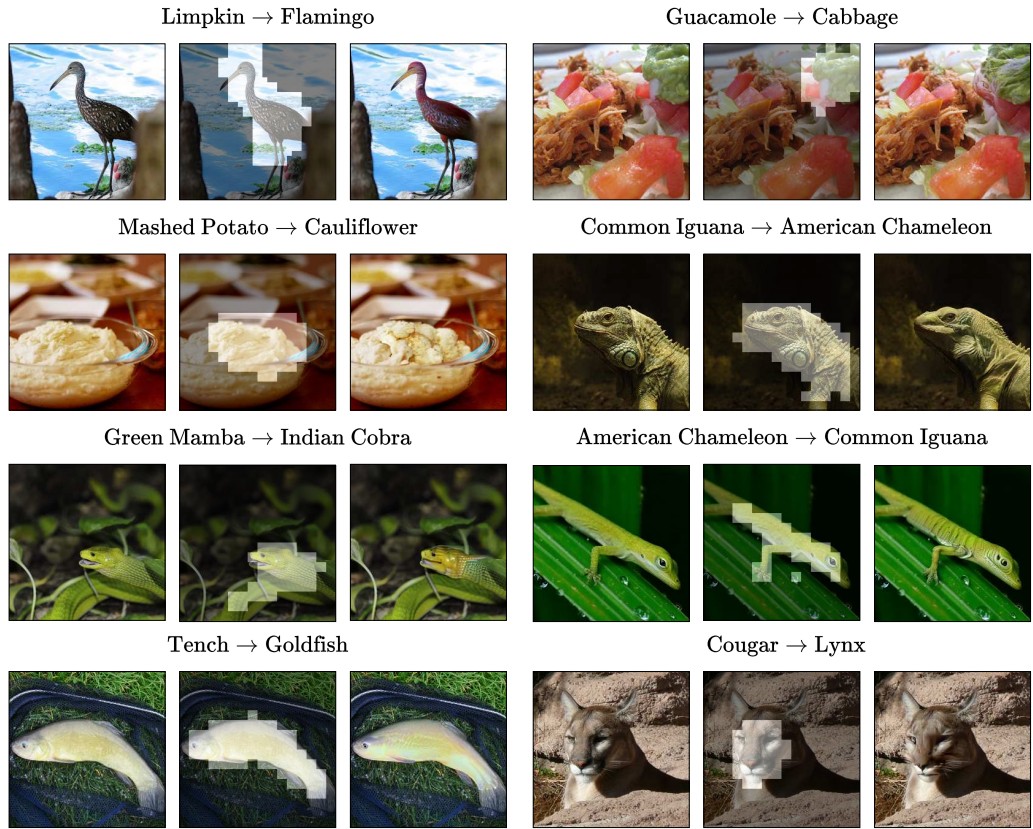

Figure 5: Qualitative examples obtained with RCSB using automated region extraction. Each task of the form *predicted class → target class* shows the factual image, the extracted region and the RVCE obtained with RCSB.

Figure 5 shows example explanations obtained with RCSB, greatly highlighting the importance of synthesizing RVCEs instead of standard VCEs. Our region extraction approach is able to precisely localize semantic concepts responsible for the model's decision. For example, in the Guacamole → Cabbage task, RCSB detects the guacamole bowl in the background and, guided by the classifier, infills it with cabbage while leaving the rest of the image unchanged. RCSB is capable of performing a wide range of editing tasks with various levels of difficulty, beginning with textural and color-based edits (*e.g.*, Tench → Goldfish, Mashed Potato → Cauliflower) to partially changing the object's structure (*e.g.*, Limpkin → Flamingo) to infilling the region with new, realistically looking concepts (*e.g.*, Cougar → Lynx, Green Mamba → Indian Cobra, Cougar → Lynx). Most importantly, thanks to the region constraint, our RVCEs allow for greatly limiting the potential factors that influenced the model's decision, making the explanations much more interpretable.

**RCSB allows for causal inference about the model's reasoning.** Drawing definite conclusions about the model's reasoning from an unconstrained VCE is not possible, as one cannot be certain that modifying potentially irrelevant factors did not in fact influence the prediction. RVCEs overcome this limitation when constrained on the region connected with the sole factor of interest, *e.g.*, the body of an animal in a species prediction task. To adapt RCSB to such scenario, we replace the automated region extraction method with a foundation text-to-object-segmentation model [1]. Using the class name from a given task as the text prompt allows us to obtain highly precise segmentation masks of the relevant objects, enabling the identification of the cause behind the model's prediction change based solely on factors related to the object of interest.

We first quantitatively assess that RCSB is capable of utilizing regions provided by a generic object detector at scale. Table 2(A) shows the results of this evaluation together with the used text prompt. Here, the metrics are computed by first discarding images with a mask that covers area larger than

---

[1]Language Segment Anything (LangSAM) combines Segment Anything Model (Kirillov et al., 2023) with GroundingDINO (Liu et al., 2023b) to allow object segmentation from text prompts.

| Metric | FID | sFID | $S^3$ | COUT | FR | FID | sFID | $S^3$ | COUT | FR | FID | sFID | $S^3$ | COUT | FR |
|---|---|---|---|---|---|---|---|---|---|---|---|---|---|---|---|
| Task | **Zebra – Sorrel** | | | | | **Cheetah – Cougar** | | | | | **Egyptian Cat – Persian Cat** | | | | |
| A | Exact regions obtained with LangSAM and prompts: zebra / horse, cheetah / cougar, cat respectively | | | | | | | | | | | | | | |
| Values | 32.8 | 41.5 | 0.87 | 0.74 | 98.9 | 37.2 | 50.6 | 0.91 | 0.84 | 99.4 | 52.0 | 82.8 | 0.81 | 0.84 | 99.2 |
| B | Regions based on freeform masks with the area in the indicated range | | | | | | | | | | | | | | |
| $10 - 20\%$ | 6.7 | 15.0 | 0.85 | 0.85 | 87.6 | 9.0 | 19.1 | 0.89 | 0.72 | 96.6 | 12.4 | 29.6 | 0.80 | 0.73 | 96.9 |
| $20 - 30\%$ | 7.8 | 15.8 | 0.84 | 0.53 | 92.2 | 11.6 | 21.3 | 0.88 | 0.71 | 99.6 | 17.7 | 34.0 | 0.78 | 0.74 | 99.3 |
| C | Ablation study with adaptations of other inpainting algorithms | | | | | | | | | | | | | | |
| RePaint | 63.8 | 76.0 | 0.55 | 0.77 | 99.3 | 129.3 | 144.2 | 0.50 | 0.77 | 99.0 | 148.7 | 175.2 | 0.38 | 0.76 | 99.5 |
| MCG | 43.2 | 55.6 | 0.73 | 0.45 | 96.0 | 76.6 | 91.4 | 0.74 | 0.64 | 100.0 | 93.7 | 117.5 | 0.62 | 0.65 | 99.9 |
| DDRM | 42.5 | 49.4 | 0.69 | 0.72 | 99.6 | 60.5 | 68.4 | 0.72 | 0.76 | 100.0 | 59.2 | 73.0 | 0.63 | 0.76 | 100.0 |

Table 2: Quantitative results from various experiments. **A**: regions extracted from LangSAM with text prompt connected to the initial class name. **B**: regions based on freeform masks that cover the fraction of the total area from the indicated range. **C**: automatically extracted regions used with adaptations of other inpainting algorithms.

$40\%$. Despite $I^2$SB being trained on masks covering at most $30\%$ of the image area, we observed that it generalizes well beyond this threshold with $40\%$ starting to pose a challenge. Crucially, despite the regions being classifier-agnostic and hence not necessarily focused on the most influential pixels, Table 2(A) indicates that RCSB is versatile enough to maintain most of the performance from the automated approach. The efficiency, sparsity and representation similarity of the obtained RVCEs remain very close to the values achieved by the closest configuration (in terms of hyperparameters) from Table 1, as the region area is often close to or exceeds $30\%$. The slight increase in FID and sFID stems mainly from the regions covering complex objects, whose modification may naturally move RVCEs further from original data at a distribution level, and a lower number of images used for these metrics' computation (as both are sensitive to sample size) due to the rejection of samples from the area constraint.

Regions that contain *exactly* the objects of interest provide novel insights about the model's reasoning. For example, consider the Lemon → Orange task from Fig. 6, where the lemons were correctly identified by the ResNet50 model. One would require the VCE for this task to indicate the sole determining factor of 'why lemons and not oranges'. However, with unconstrained VCEs, this identification process quickly becomes incomprehensible due to small changes added to each object in the image, such as other fruits. By constraining VCEs to the region occupied by the lemons, the reasoning process can be disentangled and simplified, as one can now look for this factor in the modifications of the lemons only. In this case, RCSB is guided by the classifier to change the image in a way that is consistent with human intuition.

RVCEs also allow for clarifying the model's decision-making when its reasoning is not initially understandable. In the Volcano → Seashore task, the image shows both objects, while the model predicts it as the former. Applying RCSB to the exact region of the seashore results in a RVCE that changes the model's decision when the water's color becomes more light blue and structures like stones start to appear. Hence, one is able to better understand what the model *actually* identifies as a seashore. In other examples, the method introduces class-specific characteristics when the changes are constrained *precisely* and *exclusively* to the object of interest, ensuring the receiver about the general cause of the model's decision change. Such cases are also especially relevant when the generative model used to synthesize explanations is prone to systematic errors like, *e.g.*, SGMs struggling with correctly generating hands. In the Night Snake → Kingsnake task, this error can be bypassed with the region constraint by not allowing the generative model to affect anything other than the animal, hence alleviating the evaluation of the classifier on out-of-manifold samples.

**Discovering complex patterns with interactive RVCEs.** Despite the impressive capabilities of deep models in object localization, the receiver of the explanation may be interested in testing the model for highly abstract and complex concepts that cannot be localized automatically and must be provided manually by the user. We begin with verifying the capability of RCSB in generating RVCEs based on user-defined regions by simulating such scenario at scale. Specifically, we randomly match images from the main tasks with regions given by the $10\% - 20\%$ and $20\% - 30\%$ freeform masks from the $I^2$SB training data (Saharia et al., 2022). We argue that this serves as a very

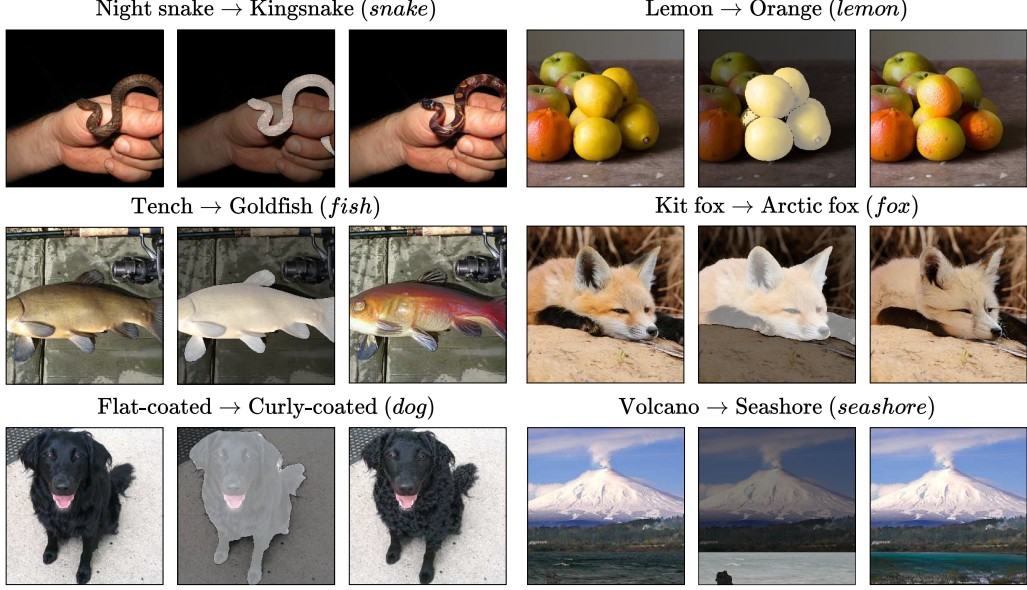

Figure 6: Qualitative examples obtained with RCSB using *exact* regions extracted from LangSAM using text prompt of the predicted class. For each task of the form *predicted class → target class*, a factual image together with the used region and the resulting RVCE are shown. The used text prompts are *emphasized*.

challenging benchmark, since the algorithm's access to the most influential pixels (for the classifier) might often be very restricted.

Despite the task's difficulty, quantitative results from Table 2(B) highlight the versatility of RCSB, which is able to effectively utilize the restricted resources to influence the classifier's prediction. While $S^3$, COUT and FR are not significantly different from previous results, we observe a decrease in FID and sFID, indicating higher realism and closeness to the data distribution. This is largely due to the fact that freeform masks are often not connected to entire complex objects and do not contain the pixels most important to the classifier. Hence, RCSB may often leave large portions of the regions unchanged, which boosts the realism evaluation.

To allow for true interaction of the user with the explanatory process, we implement a simple interface that allows for manual image segmentation using a brush-like cursor. Figure 7 shows example results, where we manually predefine the regions on different images. This exploration gives important insights about the added value provided by RVCEs. In the Cat → Tiger task, we discover that the classifier's decision can be flipped by independently modifying either the cat's paws or snout, in both cases introducing a tiger's coloration. Similarly in the Arctic Fox → Red Fox task, choosing either the ears and muzzle or paws and stomach area allows for changing the model's decision with the features of a red fox. User-defined regions also allow to discover unusual reasoning patterns of the model. In the Cucumber → Zucchini task, the model's decision can be influenced by modifying only one of the cucumbers to zucchini, leaving the other unchanged. This observation connects with recent positions on the topic of contextual and spatial understanding of predictive models (Tomaszewska & Biecek, 2024), providing new rationale in further exploring how image classifiers *actually* reason.

**Ablating RCSB's components.** We empirically verified that combining our novel guidance mechanism with the $I^2$SB prior leads to highly effective RVCEs. To better understand the benefits provided by each component of our framework, we perform an ablation study, where we adapt the proposed improvements to SGM-based inpainters, aiming to assess the influence of the guidance scheme and $I^2$SB in isolation. Specifically, we pick RePaint (Lugmayr et al., 2022), one of the first adaptations of SGMs to inpainting, MCG (Chung et al., 2022) and DDRM (Kawar et al., 2022), two different adaptations of SGMs to linear inverse problems, which also include inpainting. We manually tune our guidance scheme to each method on a small subset of images and repeat the same evaluation protocol with the automated region extraction method (see Appendix for details of each adaptation).

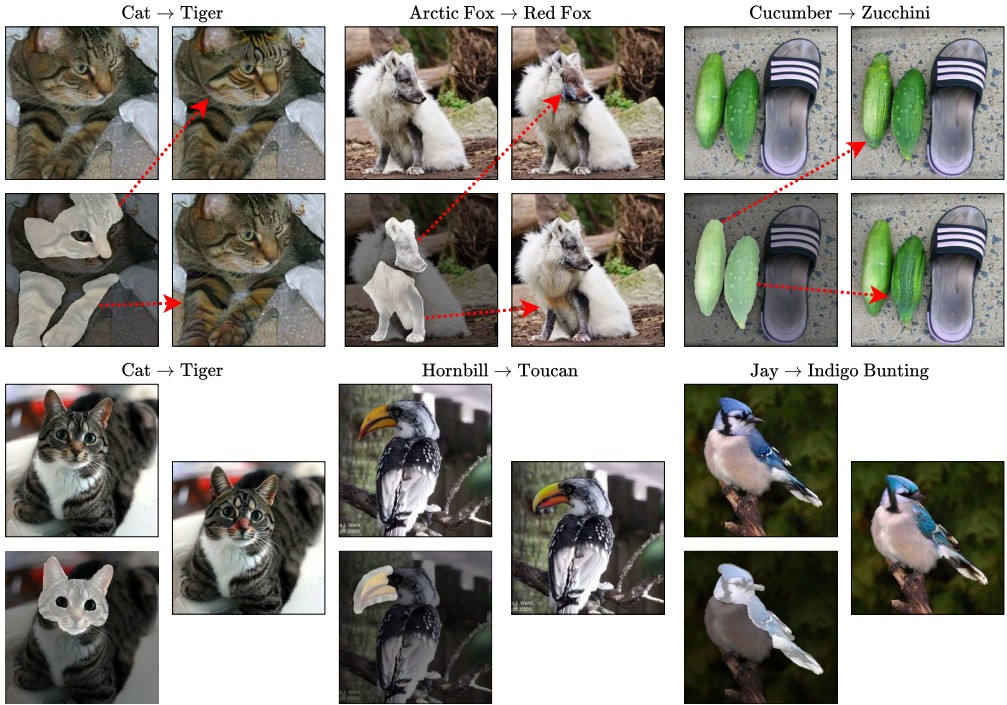

Figure 7: Qualitative examples obtained with RCSB from *user-defined* regions. For each task of the form *predicted class → target class*, a factual image together with the provided regions are shown. Arrows point to RVCEs obtained by modifying only the indicated region.

As these methods are much less compute-efficient, we cap their computational budget on each task to 24 A100 GPU hours.

Table 2(C) shows the results of the ablation study. Despite the fact that the used methods were never explicitly trained for inpainting, combining them with our guidance mechanism and region extraction allows for matching or even exceeding previous SOTA. For example, all adaptations achieve very high sparsity, almost always flip the classifier's decision and keep the explanation close to the original. This indicates the benefits of utilizing only the pixels from the extracted region and a proper utilization of the classifier's gradients without biasing them with additional components like LPIPS or $l_2$ loss. RCSB differentiates itself from the adaptations with a much higher realism of the obtained RVCEs (significantly lower FID and sFID), more balanced results and much smaller computational burden, *e.g.* $24\times$ less NFEs than RePaint. These benefits stem from the $I^2$SB prior, which is trained to map corrupted images directly to clean samples and the resulting trajectory being much closer to the data manifold, allowing the classifier to more effectively influence the inpainting process.

## 5    DISCUSSION & LIMITATIONS

RVCEs offer a new perspective on the concept of VCEs, with RCSB effectively demonstrating their versatility in various scenarios that have not been explored in previous work. In view of this, enforcing a hard region constraint—potentially chosen independently of the predictive model—introduces novel challenges and raises important questions. For instance, the explanations do not reveal changes in the interactions between different objects in the image that influence the model's decision. Furthermore, due to the absence of ground truth, verifying the actual reasoning of a model based on the explanation remains difficult, even if RVCEs appear intuitive. Additionally, the evaluation process of RVCEs may be skewed by the preservation of a large portion of the original pixels (*e.g.*, FID).

We address several of these issues in the Appendix, including two user studies on the general usefulness of RVCEs, the possibility of interacting with the explanatory process, and their informativeness regarding model misclassifications. We also present extended qualitative and quantitative results for other classifiers, datasets, and attribution methods, empirical demonstrations of some of RCSB's capabilities (*e.g.*, shape changes), and other key aspects. We believe that the limitations of RVCEs and RCSB offer valuable directions for future research.

ACKNOWLEDGMENTS

This work was financially supported by the Polish National Centre for Research and Development (NCBiR, xLungs grant no. INFOSTRATEG-I/0022/2021-00). The computational resources were provided by the Laboratory of Bioinformatics and Computational Genomics and the High Performance Computing Center of the Faculty of Mathematics and Information Science, Warsaw University of Technology.

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

CONTENTS

The Appendix is structured as follows. Appendix A shows pseudocode for both I$^2$SB (stochastic and deterministic version) and our RCSB. Appendix B delves deeper into the possible limitations of RVCEs and RCSB. Appendix C includes additional background knowledge connected with I$^2$SB and an extensive literature review regarding topics connected with our work. Appendix D compares our approach to prior methods for VCE generation in detail. Appendix E shows additional figures from the method's description, considerations regarding the incorporation of the classifier's signal into I$^2$SB and more detailed derivation of the OT-ODE version. Appendix F extends our experimental evaluation with details about the setup, additional results regarding, *e.g.*, efficiency and diversity, other datasets and classifiers, and concludes with details about the adaptation of different inpainting algorithms. Appendix G provides details about the conducted user studies. Appendix H provides qualitative examples for 7 additional classifiers, showing the versatility of RCSB, together with more RVCEs for the ResNet50 model.

## A  PSEUDOCODE

---

**Algorithm 1** Standard I$^2$SB Generation

---

1: **Input:** $\mathbf{x}_N \sim p_1(\mathbf{x}_N)$, trained $\mathbf{s}_\psi(\cdot, \cdot)$
2: **for** $n = N$ **to** 1 **do**
3:    Predict $\hat{\mathbf{x}}_0(\mathbf{x}_n)$ using $\mathbf{s}_\psi(\mathbf{x}_n, t_n)$
4:    $\mathbf{x}_{n-1} \sim p(\mathbf{x}_{n-1} \mid \hat{\mathbf{x}}_0, \mathbf{x}_n)$ according to DDPM
5: **end for**
6: **return** $\mathbf{x}_0$

---

---

**Algorithm 2** OT-ODE I$^2$SB Generation

---

1: **Input:** $\mathbf{x}_N \sim p_1(\mathbf{x}_N)$, trained $\mathbf{s}_\psi(\cdot, \cdot)$
2: **for** $n = N$ **to** 1 **do**
3:    Predict $\hat{\mathbf{x}}_0(\mathbf{x}_t)$ using $\mathbf{s}_\psi(\mathbf{x}_n, t_n)$
4:    $\mathbf{x}_{n-1} = \mu_{n-1}\hat{\mathbf{x}}_0 + \bar{\mu}_{n-1}\mathbf{x}_n$
5: **end for**
6: **return** $\mathbf{x}_0$

---

---

**Algorithm 3** RCSB

---

1: **Input:** Number of steps $N$, binary region mask $\mathbf{R}$, trajectory truncation $\tau$, classifier scale $s$, input image $\mathbf{x}^*$, trained $\mathbf{s}_\psi(\cdot, \cdot)$, trained classifier $f(\mathbf{y} \mid \cdot)$, target class $y$
2: $\mathbf{x}_1 = (\mathbf{1} - \mathbf{R}) \odot \mathbf{x}^* + \mathbf{R} \odot \mathbf{z}$, where $\mathbf{z} \sim \mathcal{N}(\mathbf{z}; \mathbf{0}, \mathbf{I})$
3: Discretize truncated timeline $0 = t_0 < t_1 < \cdots < t_N = \tau$
4: $\mathbf{x}_N \sim q(\mathbf{x}_N | \mathbf{x}_0, \mathbf{x}_1)$                # sample from analytic posterior (Eq. (15))
5: **for** $n = N$ **to** 1 **do**
6:    Predict $\hat{\mathbf{x}}_0(\mathbf{x}_n)$ using $\mathbf{s}_\psi(\mathbf{x}_n, t_n)$
7:    $\mathbf{g}_n = \nabla_{\mathbf{x}_n} \log f(y \mid \hat{\mathbf{x}}_0)$
8:    $\bar{\mathbf{g}}_n = \text{ADAM}(\mathbf{g}_n)$
9:    **if** $n == N$ **then**
       $g = \|\bar{\mathbf{g}}_N\|_2$                # register norm of the first gradient
10:   **end if**
11:   $\bar{\mathbf{x}}_n = \mathbf{x}_n + s\frac{\bar{\mathbf{g}}_n}{g}$
12:   $\mathbf{x}_{n-1} = \mu_{n-1}\hat{\mathbf{x}}_0 + \bar{\mu}_{n-1}\bar{\mathbf{x}}_n$
13: **end for**
14: **return** $\mathbf{x}_0$

---

---

**Algorithm 4** ADAM Update Rule

---

1: **Input:** Gradient at step $n$ $\mathbf{g}_n$, hyperparameters $\alpha, \epsilon, \beta_1, \beta_2$ (set to PyTorch (Paszke et al., 2019) defaults)
2: $\mathbf{m}_n = \beta_1\mathbf{m}_{n-1} + (1 - \beta_1)\mathbf{g}_n$                # update biased first moment estimate
3: $\mathbf{v}_n = \beta_2\mathbf{v}_{n-1} + (1 - \beta_2)\mathbf{g}_n^2$                # update biased second moment estimate
4: $\hat{\mathbf{m}}_n = \mathbf{m}_n/(1 - \beta_1^n)$                # compute bias-corrected first moment
5: $\hat{\mathbf{v}}_n = \mathbf{v}_n/(1 - \beta_2^n)$                # compute bias-corrected second moment
6: $\bar{\mathbf{g}}_n = \alpha\hat{\mathbf{m}}_n/(\sqrt{\hat{\mathbf{v}}_n} + \epsilon)$                # update gradient
7: **return** $\bar{\mathbf{g}}_n$                # return updated gradient

---

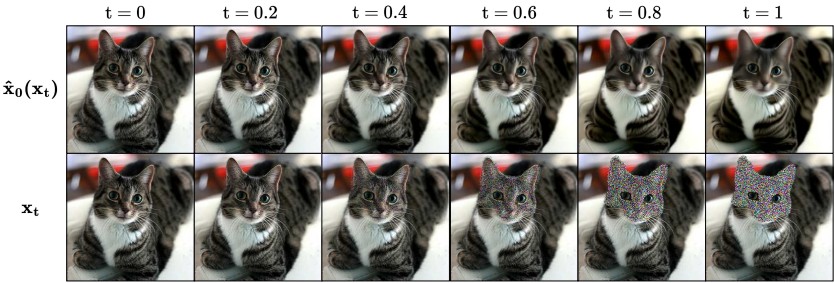

Figure 8: Visual difference between $\mathbf{x}_t$ and its corresponding Tweedie's estimate $\hat{\mathbf{x}}_0(\mathbf{x}_t)$ across different timesteps.

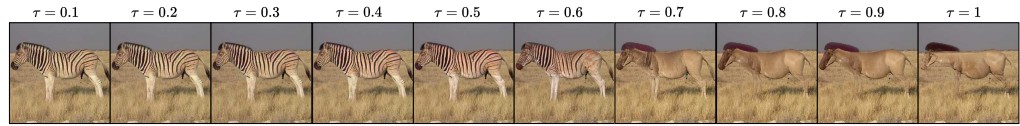

Figure 9: Influence of manipulating $\tau$ on the final RVCE obtained with the region shown in Fig. 3.

## B  LIMITATIONS

Despite setting new quantitative records, our approach comes with natural limitations that must be mentioned. From the user perspective, RVCEs generated with RCSB, especially when the region is visually appealing and extracted independently from the classifier, may lead to overconfidence about the model's decision-making process. As an example, consider the bird's image from Fig. 1 and the RVCE based on the region of its head. Because our approach allows for changing the model's decision by using solely the area of the head, one could interpret that the new features are explicitly and exclusively responsible for the new prediction. However, the situation may be much more complex. For example, the appearance of some specific features of the head, *e.g.*, a red eye, may influence the model *conditionally* on some other features which were already present in the image. Since no ground truth to a counterfactual explanation exists (at least in non-synthetic scenarios), the exact relationships between different aspects of the image remain unknown. Hence, one must be careful when interpreting any kind of counterfactual explanation, especially RVCEs, to not draw incorrect and misleading conclusions.

Moreover, modifying a given image to a counterfactual explanation, in particular to an RVCE, does not mean that the model is not relying on some unintuitive shortcuts or spurious correlations which were not modified in the process. Hence, one has to remember that VCEs aim at identifying the minimal semantic change required to change the model's decision, and are not guaranteed to modify every feature that the model relies on. These aspects highlight the importance and need of principled evaluation measures for this kind of sample-based explanations. While attribution methods have been heavily addressed in this context in recent years, the evaluation of VCEs remains a very difficult challenge.

In terms of practical limitations, our experiments are based on an I2SB model trained for an inpainting task based on freeform masks with coverage of 20%-30% of the total area. While the model generalizes well to larger fractions of the total image area on the order of 40%-45%, its performance deteriorates above this threshold. From a theoretical point of view, the I2SB algorithm is not limited to any particular upper bound on the total area, but demonstrating that it is possible to obtain good performance over a total area of the order of, *e.g.*, 90%, remains an open research question that has important implications for our work. We believe this to be an interesting direction for future research with many possible extensions.

## C  EXTENDED BACKGROUND & RELATED WORK

**Visual counterfactual explanations.** In recent years, increasing attention is being paid to synthesizing VCEs for image classifiers (Goyal et al., 2019; Schut et al., 2021). These explanations aim at elucidating the model's reasoning by modifying the input image in a semantically minimal and meaningful way while flipping its prediction. Utilizing generative models for this task has historically proven to be very effective (Chang et al., 2019; Singla et al., 2020; Lang et al., 2021). Non-SGM-based methods include works like Thiagarajan et al. (2021) which builds on the concept of deep inversion, OCTET from Zemni et al. (2023) focusing on VCEs for complex scenes and more examples built on top of generative models (Rodríguez et al., 2021; Jacob et al., 2022; Shih et al., 2021; Zhao et al., 2018; Van Looveren & Klaise, 2021).

While offering impressive results (Farid et al., 2023; Jeanneret et al., 2024; Augustin et al., 2024; Motzkus et al., 2024), we argue that utilizing general foundation models like SD in the VCE generation task may cause misleading conclusions, since the explained classifiers are trained on much smaller datasets than the generative model. For example, about 1 million images from ImageNet (Deng et al., 2009) are used to train the classifier, while SD is trained on 5 billion images from LAION-5B (Schuhmann et al., 2022). This discrepancy may naturally lead to SD synthesizing realistically looking variations of a given image that flip the classifier's decision but simultaneously include semantic attributes never observed by the predictive model during training. Therefore, one may question the *counterfactual* nature of the explanation, as the classifier should not be expected to correctly treat attributes that it was never close to observing. Hence, in this work, we focus on generative models trained on the same data as the classifier of interest. This way, we can study the behavior of predictive models in a *faithfull* manner, which is an open challenge for XAI community Biecek & Samek (2024).

**Inverse problems.** Inverse problems (Kirsch, 2011) are defined as the task of recovering an unknown signal $\mathbf{x}$ based on a measurement $\mathbf{y}$ related via a measurement model $\mathbf{H}$ through $\mathbf{y} = \mathbf{H}(\mathbf{x})$, where $\mathbf{H}$ is not necessarily required to be linear or bijective. Hence, for a given measurement $\mathbf{y}$, there may exist a probability distribution over possible solutions $p(\mathbf{x} \mid \mathbf{y} = \mathbf{H}(\mathbf{x}))$. One special case of an inverse problem is image inpainting, where the missing area of an image, indicated by the mask $\mathbf{M}$, must be infilled using the available context. The measurement model is then defined as $\mathbf{H}(\mathbf{x}) = \mathbf{M} \odot \mathbf{x}$, where $\odot$ denotes an element-wise product and $\mathbf{M}$ is a binary mask.

In recent years, deep learning methods have proven to be very effective at solving various kinds of inverse problems (Scarlett et al., 2022). Recently, utilizing generative methods, especially SGMs, established itself as the new SOTA approach in the image domain. One way of adapting SGMs to inverse problems is through conditional generation, where the conditional score can be derived with the measurement model. Many additional techniques, such as data consistency (Chung et al., 2023b), manifold constraint (Chung et al., 2022; 2023a) and others (Kawar et al., 2022), are further utilized to improve this adaptation.

**Image-to-Image Schrödinger Bridges.** A much harder but possibly also much more effective approach is to learn *direct* mappings between the distribution of signals $\mathbf{x} \sim p_0$ and measurements $\mathbf{y} \sim p_1$ instead of adapting pretrained models. In this line of research, Liu et al. (2023a) propose to learn such mappings by constructing a tractable subclass of Schrödinger bridges (SBs, Schrödinger (1932)), termed Image-to-Image Schrödinger Bridges ($\text{I}^2\text{SBs}$). The SB is an entropy-regularized optimal transport model, which, resembling the framework of SGMs, considers the following forward and backward SDEs:

$$\mathrm{d}\mathbf{x}_t = (\mathbf{F}_t(\mathbf{x_t}) + \beta_t \nabla_{\mathbf{x}_t} \log \mathbf{\Psi}(\mathbf{x}_t, t))\mathrm{d}t + \sqrt{\beta_t}\mathrm{d}\mathbf{w}, \tag{4a}$$

$$\mathrm{d}\mathbf{x}_t = (\mathbf{F}_t(\mathbf{x_t}) - \beta_t \nabla_{\mathbf{x}_t} \log \widehat{\mathbf{\Psi}}(\mathbf{x}_t, t))\mathrm{d}t + \sqrt{\beta_t}\mathrm{d}\bar{\mathbf{w}}, \tag{4b}$$

Similarly to SGMs, the marginal densities of Eqs. (4a) and (4b) are equivalent. The functions $\mathbf{\Psi}, \widehat{\mathbf{\Psi}} \in C^{2,1}(\mathbb{R}^d, [0,1])$ represent time-varying energy potentials and are additionally constrained to solve the following partial differential equations

$$\begin{cases} \frac{\mathbf{\Psi}(\mathbf{x}_t, t)}{\partial t} = -\nabla\mathbf{\Psi}^\top \mathbf{F} - \frac{1}{2}\beta\Delta\mathbf{\Psi} \\ \frac{\widehat{\mathbf{\Psi}}(\mathbf{x}_t, t)}{\partial t} = -\nabla \cdot (\widehat{\mathbf{\Psi}}\mathbf{F}) + \frac{1}{2}\beta\Delta\widehat{\mathbf{\Psi}} \end{cases} \tag{5a}$$

$$\text{s.t. } \mathbf{\Psi}(\mathbf{x}_0, 0)\widehat{\mathbf{\Psi}}(\mathbf{x}, 0) = p_0(\mathbf{x}), \mathbf{\Psi}(\mathbf{x}, 1)\widehat{\mathbf{\Psi}}(\mathbf{x}, 1) = p_1(\mathbf{x}) \tag{5b}$$

In general, numerically solving Eqs. (4a) and (4b) is much more difficult compared to SGMs due to nonlinear terms $\boldsymbol{\Psi}, \widehat{\boldsymbol{\Psi}}$ being coupled via Eq. (5b). However, with **Theorem 3.1**, Liu et al. (2023a) show an important connection between the frameworks of SBs and SGMs. We repeat it here explicitly for later reference.

**Theorem 1 (Reformulating SB drifts as score functions (Liu et al., 2023a))** *If $\widehat{\boldsymbol{\Psi}}, \boldsymbol{\Psi}$ fulfill the constraints given by Eq. (5), then $\nabla_{\mathbf{x}_t} \log \widehat{\boldsymbol{\Psi}}(\mathbf{x}_t, t), \nabla_{\mathbf{x}_t} \log \boldsymbol{\Psi}(\mathbf{x}_t, t)$ are the score functions of the following linear SDEs, respectively*

$$\mathrm{d}\mathbf{x}_t = \mathbf{F}_t(\mathbf{x_t})\mathrm{d}t + \sqrt{\beta_t}\mathrm{d}\mathbf{w}, \quad \mathbf{x}_0 \sim \widehat{\boldsymbol{\Psi}}(\cdot, 0), \tag{6}$$

$$\mathrm{d}\mathbf{x}_t = \mathbf{F}_t(\mathbf{x_t})\mathrm{d}t + \sqrt{\beta_t}\mathrm{d}\bar{\mathbf{w}}, \quad \mathbf{x}_1 \sim \boldsymbol{\Psi}(\cdot, 1). \tag{7}$$

Crucially, Theorem 1 states that, while $\widehat{\boldsymbol{\Psi}}, \boldsymbol{\Psi}$ are not in general assumed to be valid probability distributions, it is true that $\nabla \log \widehat{\boldsymbol{\Psi}} = \nabla \log p^6$ and $\nabla \log \boldsymbol{\Psi} = \nabla \log p^7$ for $p^6, p^7$ representing the densities of the respective SDEs. Following this theoretical result, Liu et al. (2023a) also show a principled approach for approximating $\nabla_{\mathbf{x}_t} \log \widehat{\boldsymbol{\Psi}}(\mathbf{x}_t, t)$ with a neural network $\mathbf{s}_\psi$. In essence, these results allow to train *direct* inverse problem solvers with the use of paired data, where $p_0$ represents a clean data distribution and $p_1$ the distribution of its corrupted measurements. Additionally, Liu et al. (2023a) show how I²SB connects with flow-based optimal transport (OT) (Peyré & Cuturi, 2019; McCann, 1997), where assuming that $\beta_t \to 0$ leads to an ordinary differential equation (ODE) $\mathrm{d}\mathbf{x}_t = \mathbf{v}_t(\mathbf{x}_t \mid \mathbf{x}_0)\mathrm{d}t$ that provides a deterministic mapping with the use of $\mathbf{s}_\psi$ estimate. In practive, this is achieved by eliminating the noise from the intermediate sampling steps (see Algorithm 2).

**Visual attribution methods.** The very first works in the current era of Explainable Artificial Intelligence (XAI) were concerned with providing explanations of the model's decision through visual heatmaps, which highlighted pixels considered important to the its prediction. One of the first approaches by Simonyan et al. (2014) proposed simple backpropagation of the model's output w.r.t. the input, indicating the direction of its greatest ascent in the pixel space, often termed as Saliency. More sophisticated approaches emerged in the following years, where techniques like Layer-wise Relevance Propagation (LRP, Bach et al. (2015)), Integrated Gradients (IG, Sundararajan et al. (2017)), DeepLift and Input × Gradient (Shrikumar et al., 2017), Guided Backpropagation (GuidedBackprop, Springenberg et al. (2015)), GradCAM (Selvaraju et al., 2020), Deconvolution (Zeiler & Fergus, 2014) and others that utilize the gradient of the neural network promised to indicate more semantically meaningful concepts in a less 'noisy' manner. Concurrent line of research about perturbation-based methods assumed a more general black-box scenario, where explanations could be provided for a broader class of models. There, methods like Occlusion (Zeiler & Fergus, 2014), Local Interpretable Model-agnostic Explanations (LIME, Ribeiro et al. (2016)), SHapley Additive exPlanations (SHAP, Lundberg & Lee (2017)) and its variations quickly advanced the state-of-the-art. In this work, we utilize their unified implementations provided by the Captum package (Kokhlikyan et al., 2020).

## D  COMPARISON TO PREVIOUS WORKS

Due to space limits, we only briefly mention the previous diffusion-based approaches to VCE generation in Section 2. In the following, we provide more details about their theoretical formulations and how they relate to our approach.

Most of the previously published diffusion-based approaches to VCEs rely on the conditional reverse process obtained by replacing the unconditional score $\nabla_{\mathbf{x}_t} \log p(\mathbf{x}_t, t)$ in Eq. (1b) with the conditional variant of the form $\nabla_{\mathbf{x}_t} \log p(\mathbf{x}_t, t) + \nabla_{\mathbf{x}_t} \log p(\mathbf{y} \mid \mathbf{x}_t, t)$. This formulation allows one to incorporate the classifier-based conditioning in various forms. Jeanneret et al. (2022) (DiME) propose to approximate the likelihood score $\nabla_{\mathbf{x}_t} \log p(\mathbf{y} \mid \mathbf{x}_t, t)$ by fully denoising the image at step $t$ and using it to obtain the classifier's gradient leading to quadratic complexity with respect to the total number of timesteps. The work of Augustin et al. (2022) (DVCE) instead propose to regularize the gradients of the explained classifier with the one coming from a robustly trained classification network. Specifically, the gradient of the former is projected onto a cone around the direction indicated by the gradient of the latter with some predetermined angle. The follow-up work of Jeanneret et al.

(2023) (ACE) proposes to split the explanation generation into two distinct phases, with the first one responsible for generating the *pre-explanation*, and the second one performing *post-processing*. The former combines standard diffusion denoising with a PGD attack performed with the use of the classifier on $\mathbf{x}_t$ for each $t$. Then, the latter phase computes the absolute difference between the original image and the pre-explanation, and extracts a binary mask by thresholding this difference. Next, it uses the RePaint (Lugmayr et al., 2022) algorithm to unconditionally inpaint the masked region beginning from some intermediate timestep. To address the low efficiency of DiME, Weng et al. (2024) (FastDiME) propose to improve the conditioning process by utilizing at each timestep $t$ the gradient of the classifier with respect to the Tweedie's estimate. By default, their approach performs dynamic masking throughout the generation process which indicates the region modified at each timestep. Moreover, the authors propose two 2-step extensions of their approach, namely FastDiME-2 and FastDiME-2+, that first perform either standard FastDiME or FastDiME without dynamic masking, then extract the most differing regions and utilize the binary mask resulting from the largest changes to conditionally inpaint it with the use of the classifier.

Importantly, ACE and FastDiME bear resemblence to our approach in terms of performing some variant of conditional inpainting. Specifically, ACE first combines the classifier with diffusion denoising to extract a region important to the predictive model, but inpaints it unconditionally. Fast-DiME (and its variants) also find a region that is regarded as important to the classifier, but inpaint it conditionally. Our approach solves a more general problem by synthesizing RVCEs which do not assume any dependence of the predetermined region to the classifier of interest, which we show through a large array of experiments with regions coming from sources like automated segmentation or user interaction. Moreover, while all of the previously mentioned works are concerned with adapting standard SGMs to either guided generation or inptainting, we show how a more general class of models (tractable Schrödinger Bridges) can be adapted to the problem of conditional inpainting with the use of a classifier. Additionally, we propose a series of improvements that better align its gradients with the generative trajectory that were not previously present in this line of research connected to XAI, such as ADAM stabilization or adaptive normalization. Finally, to the best of our knowledge, our approach is the first one to show that guidance can be performed with the signal coming solely from the classifier of interest, omitting the usage of additional proxy measures (like $l_2$ loss or LPIPS) that maintain similarity to the original image. It it also important to higlight that, while very simple, our automated region extraction approach was also not present in this line of research and, through experimental evaluation, was shown to provide highly interpretable and intuitive regions.

## E EXTENDED METHOD

### E.1 ADDITIONAL FIGURES

We provide illustrative examples for the visual differences between $\mathbf{x}_t$ and the Tweedie's estimate $\hat{\mathbf{x}}_0(\mathbf{x}_t)$ in Fig. 8. For the effect of manipulating the $\tau$ hyperparameter, see Fig. 9

### E.2 INCORPORATING THE CLASSIFIER'S SIGNAL

In the following, we explicitly define the DDPM (Ho et al., 2020) sampler mentioned in Algorithm 1 and elaborate on the exact way of incorporating the classifier's gradients into the generation process of I²SB.

Denote by $\{t_i\}_{i \in \{0,...,N\}}$ the discrete sequence of timesteps of length N such that $0 = t_0 < t_1 < \cdots < t_N = 1$. By $\sigma_n^2 = \int_0^{t_n} \beta_\tau d\tau$ and $\bar{\sigma}_n^2 = \int_{t_n}^1 \beta_\tau d\tau$, we denote the variances accumulated from each side. Additionally, let $\alpha_{n-1}^2 = \int_{t_{n-1}}^{t_n} \beta_\tau d\tau$ be the variance accumulated between two consecutive timesteps. For ease of notation, we define $\mu_{n-1}$ and $\bar{\mu}_{n-1}$ as

$$\mu_{n-1} = \frac{\alpha_{n-1}^2}{\alpha_{n-1}^2 + \sigma_{n-1}^2}, \tag{8}$$

$$\bar{\mu}_{n-1} = \frac{\sigma_{n-1}^2}{\alpha_{n-1}^2 + \sigma_{n-1}^2}. \tag{9}$$

With that, we can define the DDPM posterior sampler as

$$\mathbf{x}_{n-1} \sim p(\mathbf{x}_{n-1}|\hat{\mathbf{x}}_0, \mathbf{x}_n), \tag{10}$$

$$\mathbf{x}_{n-1} \sim \mathcal{N}\left(\mu_{n-1}\hat{\mathbf{x}}_0 + \bar{\mu}_{n-1}\mathbf{x}_n, \frac{\alpha_{n-1}^2 \sigma_{n-1}^2}{\alpha_{n-1}^2 + \sigma_{n-1}^2}I\right), \tag{11}$$

where $\hat{\mathbf{x}}_0 = \hat{\mathbf{x}}_0(\mathbf{x}_n)$ denotes the Tweedie's estimate obtained with $\mathbf{s}_\psi$, *i.e.*, a trained I$^2$SB.

When using the OT-ODE version of I$^2$SB, we replace sampling from the posterior with a deterministic version by following the mean, which yields the update rule

$$\mathbf{x}_{n-1} = \mu_{n-1}\hat{\mathbf{x}}_0 + \bar{\mu}_{n-1}\mathbf{x}_n. \tag{12}$$

By converting the Tweedie's estimate to the conditional score using Eq. (3) and applying Bayes' Theorem, we are left with

$$\begin{aligned}
\mathbf{x}_{n-1} &= \mu_{n-1}\left(\mathbf{x}_n + \sigma_n^2 \nabla_{\mathbf{x}_n} \log p(\mathbf{x}_n, n \mid \mathbf{y})\right) + \bar{\mu}_{n-1}\mathbf{x}_n \\
&= \mu_{n-1}\left(\mathbf{x}_n + \sigma_n^2 \nabla_{\mathbf{x}_n} \log p(\mathbf{x}_n, n) + \sigma_n^2 \nabla_{\mathbf{x}_n} \log p(\mathbf{y} \mid \mathbf{x}_n, n)\right) + \bar{\mu}_{n-1}\mathbf{x}_n,
\end{aligned} \tag{13}$$

where $\nabla_{\mathbf{x}_n} \log p(\mathbf{x}_n, n)$ can be approximated by a standard I$^2$SB network trained on the task of inpainting. By manipulating Eq. (13) further, one can arrive at the following update rule

$$\begin{aligned}
\mathbf{x}_{n-1} &= \mu_{n-1}\left(\mathbf{x}_n + \sigma_n^2 \nabla_{\mathbf{x}_n} \log p(\mathbf{x}_n, n)\right) + \mu_{n-1}\sigma_n^2 \nabla_{\mathbf{x}_n} \log p(\mathbf{y} \mid \mathbf{x}_n, n) + \bar{\mu}_{n-1}\mathbf{x}_n \\
&= \mu_{n-1}\left(\mathbf{x}_n + \sigma_n^2 \nabla_{\mathbf{x}_n} \log p(\mathbf{x}_n, n)\right) + \bar{\mu}_{n-1}\left(\frac{\mu_{n-1}\sigma_n^2}{\bar{\mu}_{n-1}} \nabla_{\mathbf{x}_n} \log p(\mathbf{y} \mid \mathbf{x}_n, n) + \mathbf{x}_n\right) \\
&= \mu_{n-1}\left(\mathbf{x}_n + \sigma_n^2 \nabla_{\mathbf{x}_n} \log p(\mathbf{x}_n, n)\right) + \bar{\mu}_{n-1}\left(c_n \nabla_{\mathbf{x}_n} \log p(\mathbf{y} \mid \mathbf{x}_n, n) + \mathbf{x}_n\right).
\end{aligned} \tag{14}$$

Here, we explicitly define the time-dependent coefficient $c_n$. While plugging $\nabla_{\mathbf{x}_n} \log f(\mathbf{y} \mid \mathbf{x}_n)$ in place of $\nabla_{\mathbf{x}_n} \log p(\mathbf{y} \mid \mathbf{x}_n, n)$ in Eq. (13) is the most intuitive, we empirically verified that replacing $c_n \nabla_{\mathbf{x}_n} \log p(\mathbf{y} \mid \mathbf{x}_n, n)$ with $\nabla_{\mathbf{x}_n} \log f(\mathbf{y} \mid \mathbf{x}_n)$ leads to more semantically meaningful results. Practically, this can be explained by $\mu_{n-1}$ achieving its highest values at the end of the generation process, effectively incorporating the classifier's signal to the highest extent in the final steps of the generation. Since we are interested in influencing the generative trajectory with the classifier $f$ along the entire process (and possibly decreasing its influence to the greatest possible extent in the final timesteps to avoid adversarial changes), it seems intuitive that incorporating $f$ into Eq. (14) allows for obtaining more meaningful RVCEs. This is due to $\bar{\mu}_n = 1 - \mu_n$, meaning that the classifier's signal is amplified in the beginning of the generation and decreased in the final steps. This intervention also explains the effectiveness of the introduced improvements, as they break the independence of the classifier's signal between consecutive steps, practically incorporating the time-dependent coefficient $c_n$ into the gradient alignment.

### E.3 ANALYTIC POSTERIOR AND OT-ODE

Following the original work of Liu et al. (2023a) (I$^2$SB), the analytic posterior from the forward stochastic process, which governs the mapping between a given boundary pair $(\mathbf{x}_0, \mathbf{x}_1)$, is defined as

$$q(\mathbf{x}_t|\mathbf{x}_0, \mathbf{x}_1) = \mathcal{N}\left(\boldsymbol{\mu}(\mathbf{x_0}, \mathbf{x_1}, t) = \mathbf{x}_0 + t(\mathbf{x}_1 - \mathbf{x}_0), \boldsymbol{\Sigma}_t = \alpha t(1-t)\mathbf{I}\right), \tag{15}$$

where by default $\alpha = 1$. To arrive at the OT-ODE version of I$^2$SB, one must use $\alpha \to 0$, effectively reducing $q$ to a Dirac delta distribution centered at $\boldsymbol{\mu}(\mathbf{x_0}, \mathbf{x_1}, t)$.

## F  EXTENDED EXPERIMENTS

We follow the evaluation protocol from previous works for VCEs on Imagenet, which, for a given task, uses all images from the training subset correctly predicted by the evaluated model. For ResNet50, this results in around 2000 images per task. We extract the results of other methods from the work of Farid et al. (2023), except the DVCE method (Augustin et al., 2022), which evalutes with a protocol that we were not able to fully reproduce. Hence, to ensure fair comparison, we adapted the implementation of DVCE to our evaluation. Specifically, we utilize the multiple-norm

robust ResNet50 from the work of Boreiko et al. (2022), which the authors of DVCE propose as default, to achieve VCEs for the ResNet50 model. In terms of hyperparameters, we fine-tuned them with grid search on a subset of Zebra–Sorrel task and used $s = 18.0$ as the guidance scale for the non-robust ResNet50, since it performed the best.

For I$^2$SB, we utilize the original checkpoint from Liu et al. (2023a) trained on $20 - 30\%$ freeform masks from Saharia et al. (2022). While the checkpoint trained on the $10 - 20\%$ variant is also available and verified to work within our framework, we discovered that the former generalizes well to smaller area values. Hence, for the sake of simplicity, we utilize the $20 - 30\%$ version only. By default, we use NFE=100, which we explored the most, but lower NFE regimes provided promising initial results. For the automated region extraction, we use IG by default, but evaluate 10 other attribution methods in Appendix F.5.

## F.1 DETAILS OF INDIVIDUAL EXPERIMENTS

Fig. 3: Each improvement, together with the naive approach, is evaluated on the zebra-sorrel task with around 2000 images from ImageNet training set (following the protocol from the main experimental evaluation). Each image is initially predicted as either zebra or sorrel by the ResNet50 (He et al., 2016) model and the decision must be flipped to the opposite class. FID is computed between the obtained explanations and original images. The hyperparameter values used for all improvements are: $a = 0.3$, $s = 1.0$ (except ADAM stabilization with $s = 1e - 2$), $c = 16$, $\tau = 1.0$ (except trajectory truncation, where $\tau = 0.6$).

Table 2: **A**: For all tasks, we use $s = 1.5$ and $\tau = 0.4$ (to better preserve the original content). As we cannot control the area of masks provided by LangSAM, hyperparameters $a$ and $c$ are not applicable in this scenario. Images with masks covering area greater than $40\%$ of the total image are discarded from the evaluation to ensure that we only use meaningful RVCEs. **B**: Across all tasks, the $10\% - 20\%$ experiment uses configuration $B$ from Table 1, while the $20\% - 30\%$ experiment uses configuration $C$. Hyperparameters $a$ and $c$ are not applicable, since masks are provided automatically from the mentioned dataset. **C**: Each inpainting algorithm is given a 24 A100 GPU hours time budget, resulting in around 2000 images for DDRM, 800 images for MCG and 400 images for RePaint on each task. Details of their adaptations are provided separately in Appendix F.3.

## F.2 METRICS DESCRIPTION

In the following, we provide detailed description of each metric used in the quantitative evaluation.

**FID and sFID (realism).** Following works on image synthesis, measuring the realism of the obtained explanations at a distribution level is often done with FID and sFID (Heusel et al., 2017). Specifically, FID compares a set of real ($r$) and generated ($g$, in this case, the explanations) images by first extracting their corresponding features from the InceptionV3 network (Szegedy et al., 2016) and then computing

$$\text{FID} = ||\boldsymbol{\mu}_r - \boldsymbol{\mu}_g||^2 + \text{Tr}\left(\Sigma_r + \Sigma_g - 2\left(\boldsymbol{\Sigma}_r \boldsymbol{\Sigma}_g\right)^{1/2}\right), \tag{16}$$

where $\boldsymbol{\mu}_r, \boldsymbol{\mu}_g$ are the mean vectors and $\boldsymbol{\Sigma}_r, \boldsymbol{\Sigma}_g$ are the covariance matrices of the respective distributions in the feature space. As comparing original images with their edited versions (*e.g.*, explanations) may bias the metric with original pixels mostly unchanged, artificially boosting the realism evaluation, sFID first divides the sets into folds and averages FID over the independent counterparts.

**S$^3$ (representation similarity).**

Explanations should also resemble original images from a representation respective. Here, following the work of Jeanneret et al. (2023), we compute average SimSiam Similarity (S$^3$) over a set of original images and the resulting counterfactuals. Specifically, S$^3$ utilizes a SimSiam network (Chen & He, 2021), which encodes both the factual and counterfactual images into their respective representations $\mathbf{r}_f$, $\mathbf{r}_{cf}$ and computes the cosine similarity as

$$\text{S}^3 = \frac{\mathbf{r}_f \cdot \mathbf{r}_{cf}}{||\mathbf{r}_f||_2 \cdot ||\mathbf{r}_{cf}||_2}. \tag{17}$$

**COUT (sparsity).** In the context of VCEs, sparsity is understood as perturbing a minimal number of pixels to flip the model's decision. To quantify this criterion, the COUnterfactual Transition (COUT)

metric computes

$$\text{COUT} = \text{AUPC}_y - \text{AUPC}_{y^*},$$
$$\text{AUPC}_k = \frac{1}{M} \sum_{m=0}^{M-1} \frac{1}{2} (f(k \mid \mathbf{x}^{(m)}) - f(k \mid \mathbf{x}^{(m+1)})) \tag{18}$$

where $\mathbf{x}^{(0)}$ is the factual image, $\mathbf{x}^{(M)}$ the resulting VCE, and $y^*, y$ are the class labels predicted by $f$ for $\mathbf{x}^{(0)}, \mathbf{x}^{(M)}$ respectively. In practice, COUT measures how fast the classifier's decision changes when interpolating between the original and the explanation, but the interpolation is defined as inserting pixels to the original image according to the extent (absolute value) of change observed in the VCE through $M$ steps. COUT is typically reported as an average over a set of samples.

**Flip Rate (efficiency).** A major criterion for a VCE method is its efficiency, understood as the ability to effectively flip the model's decision. For a set of triplets $\{\mathbf{x}_i^*, \mathbf{x}_i, y_i\}_{i=1}^{I}$, where $\mathbf{x}_i^*$ is the original image and $\mathbf{x}_i$ is the resulting VCE targeted to flip $f$'s decision to $y_i$, Flip Rate (FR) is defined as the fraction of cases which correctly flipped the decision to the target class, *i.e.*,

$$\text{FR} = \frac{1}{I} \sum_{i=1}^{I} \mathbf{1}(\arg\max_{y'} f(y' \mid \mathbf{x}_i) = y_i). \tag{19}$$

### F.3 Adaptation of other inpainting algorithms

In this subsection, we describe the adaptation of each inpainting algorithm from our ablation study. For each method, we follow the notation from its corresponding original work to make the description easier to follow.

#### F.3.1 Manifold Constrained Gradient (MCG, Chung et al. (2022))

MCG iteratively denoises (inpaints) the missing parts with the following two-step update (Equations 14 and 15, Chung et al. (2022)):

$$\mathbf{x}'_{t-1} = \mathbf{f}(\mathbf{x_i}, \mathbf{s}_\theta) - \frac{\partial}{\partial \mathbf{x_t}} \|\mathbf{W}(\mathbf{y} - \mathbf{H}\hat{\mathbf{x}}_\mathbf{0}(\mathbf{x_t}))\|_2^2 + g(\mathbf{x_t})\mathbf{z}, \quad \mathbf{z} \sim \mathcal{N}(0, \mathbf{I}) \tag{20}$$
$$\mathbf{x_{i-t}} = \mathbf{A}\mathbf{x}'_{t-1} + \mathbf{b} \tag{21}$$

where Eq. (20) is a manifold constraint update and Eq. (21) is a data consistency step. As described by the authors, both steps are crucial to ensure that the gradient of the measurement term stays on the manifold and to deal with the potential deviation from the measurement consistency.

Since $\mathbf{f}(\mathbf{x}_t, \mathbf{s}_\theta)$ implicitly predicts the mean $\mu_t$ and variance $\sigma_t$ at each step $t$, related to the underlying SDE dynamics, we apply our guidance scheme by modifying the original

$$\mathbf{f}(\mathbf{x_i}, \mathbf{s}_\theta) = \mu_t + \sigma_t \mathbf{z}, \quad \mathbf{z} \sim \mathcal{N}(0, \mathbf{I}) \tag{22}$$

by adding properly scaled (according to the relationship between the likelihood score and mean) conditioning:

$$\mathbf{f}'(\mathbf{x_i}, \mathbf{s}_\theta) = \mu_t + \sigma_t \mathbf{z} + s \cdot \sigma_t^2 \frac{\overline{\mathbf{g}}_n}{g}, \quad \mathbf{z} \sim \mathcal{N}(0, \mathbf{I}) \tag{23}$$

where $\mathbf{g}_n$ and $g$ are obtained as described in Algorithm 3.

#### F.3.2 Denosing Diffusion Restoration Models (DDRM, Kawar et al. (2022))

DDRM considers the SVD decomposition of the measurement model matrix $\mathbf{H}$ in the linear noisy inverse problem

$$\mathbf{y} = \mathbf{H}\mathbf{x} + \sigma_\mathbf{y} \mathbf{z}, \mathbf{z} \sim \mathcal{N}(0, \mathbf{I}), \tag{24}$$

where $\sigma_{\boldsymbol{y}}$ is the standard divination of the measurement noise. For the task of inpaiting, the $\mathbf{H}$ matrix is a diagonal matrix with either $0$ or $1$ on the diagonal indicating available and missing pixels. Hence, its SVD decomposition simplifies to using identity matrices in place of $\mathbf{U}$ and $\mathbf{V}$.

The main contribution of DDRM is that it provides a way to include the information from that decomposition and observation $\mathbf{y}$ into the generative process, which the authors summarize in Equations 7 and 8 in the original work. The method uses a trained denoising network to obtain a prediction of $\mathbf{x}_0$ at timestep $t$, denoted as $\mathbf{x}_{\theta,t}$. In order to include the additional information from the classifier into the DDRM framework, we modify that prediction to include the model's gradients by replacing the update rule

$$\bar{\mathbf{x}}_{\theta,t} = V^T \mathbf{x}_{\theta,t} \tag{25}$$

with

$$\bar{\mathbf{x}}_{\theta,t} = V^T (\mathbf{x}_{\theta,t} + s \frac{\bar{g}}{g}), \tag{26}$$

where $\bar{g}$ and $g$ are obtained as described in Algorithm 3.

### F.3.3 RePaint (Lugmayr et al., 2022)

RePaint performs the task of inpainting by modifying the standard denoising process, where, at each timestep $t$, the network's input is composed of noised pixels known from the original input sampled directly from $q$ and the unknown noisy pixels predicted by the network in the previous timestep. Additionally, to harmonize the two parts of the image, RePaint samples $x_t$ directly from $q(x_t|x_{t-1})$ and repeats the forward procedure. By default, this resampling scheme is repeated 20 times for each of the standard diffusion steps.

In order to incorporate the information from the classifier, we modify the unconditional mean of the posterior $p_\theta$ in the denosing step to conditional one, effectively replacing the mean predictor

$$\varepsilon_{\boldsymbol{\theta}}(X_t, t) \tag{27}$$

shown in the 7-th step of Algorithm 1 from the original work (Lugmayr et al., 2022) with

$$\varepsilon_{\boldsymbol{\theta}}(X_t, t) + s \cdot \sigma_t^2 \frac{\bar{g}}{g} \tag{28}$$

where $\bar{g}$ and $g$ are obtained as described in Algorithm 3.

### F.4 Schedulers for guidance scale

Figure 10 visualizes example schedulers used throughout the development of our method. Our adaptive normalization technique empirically outperformed all tested schedulers.

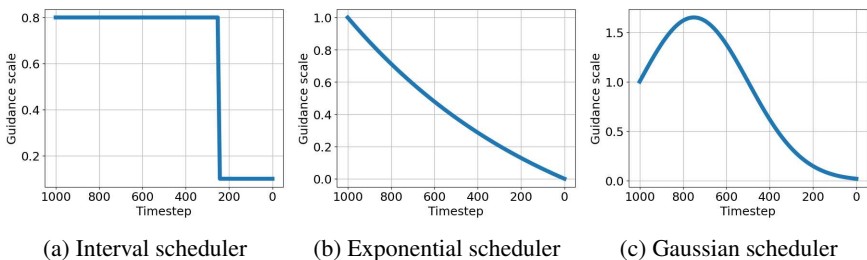

(a) Interval scheduler  (b) Exponential scheduler  (c) Gaussian scheduler

Figure 10: Visualization of more complex schedulers used throughout the development of our method.

### F.5 Quantitative evaluation of other attribution methods

To pick a default attribution method for RCSB, we evaluated it on the Zebra–Sorrel using the $\text{RCSB}^B$ hyperparameter configuration for 11 different attribution methods shown in Table 3. Based on these results, we chose Integrated Gradients (Sundararajan et al., 2017) as the default, since it provides the most balanced performance.

| Zebra – Sorrel | | | | | |
|---|---|---|---|---|---|
| Attribution method | FID | sFID | $S^3$ | COUT | FR |
| LRP | **7.5** | **15.5** | **0.87** | 0.62 | 93.6 |
| InputXGradient | 9.0 | 16.8 | 0.87 | 0.73 | 97.8 |
| DeepLift | 9.2 | 17.0 | 0.87 | 0.73 | 97.9 |
| Integrated Gradients | 9.5 | 17.4 | 0.86 | 0.72 | 97.4 |
| GradientShap | 10.5 | 18.5 | 0.87 | 0.74 | 97.4 |
| LIME | 12.9 | 20.7 | 0.85 | 0.55 | 88.4 |
| GuidedBackprop | 13.8 | 21.49 | 0.86 | 0.72 | 96.5 |
| Occlusion | 13.9 | 21.7 | 0.86 | 0.50 | 86.0 |
| GradCAM | 14.1 | 22.15 | 0.85 | 0.52 | 87.1 |
| GuidedGradCAM | 15.1 | 22.5 | 0.86 | 0.71 | 96.1 |
| Saliency | 15.2 | 23.0 | 0.86 | **0.75** | **98.4** |

Table 3: Quantitative evaluation of 11 attribution methods (described in Appendix C) on the Zebra–Sorrel task following our evaluation protocol.

## F.6 DIVERSITY ASSESSMENT

RCSB utilizes the OT-ODE version of the I$^2$SB, which provides a deterministic mapping between the noisy image and the resulting RVCE. The source of randomness comes from the Gaussian noise inserted into the image in the place of missing pixels at the beginning of the inpainting process. In order to examine the diversity of the generated RVCEs, we followed the evaluation procedure from the work of Jeanneret et al. (2023). In essence, we compute the mean pair-wise LPIPS metric between two runs with different seeds (used in generation of the Gaussian noise) for our three main configurations of hyperparameters RCSB$^A$, RCSB$^B$ and RCSB$^C$. For each run, 256 RVCEs were generated. Results are shown in Table 4. Naturally, decreasing the area hyperparameter limits the extent of possible changes, leading to a decrease in diversity. Picking $a = 0.3$ results in diversity comparable to values reported by previous works, *e.g.*, Jeanneret et al. (2023).

| | Zebra – Sorrel | Cheetah – Cougar | Egyptian Cat – Persian Cat |
|---|---|---|---|
| RCSB$^A$ | 0.067 | 0.060 | 0.065 |
| RCSB$^B$ | 0.092 | 0.096 | 0.095 |
| RCSB$^C$ | 0.129 | 0.140 | 0.137 |

Table 4: Diversity evaluation using 256 images for each task from our experimental protocol.

## F.7 COMPUTATIONAL EFFICIENCY ASSESSMENT

What connects previous SOTA SGM-based methods with our work is the use of large U-Net (Ronneberger et al., 2015) checkpoints for the denoising network with the number of hyperparameters far exceeding (*e.g.*, $10\times$) the size of the utilized classifier, effectively dominating the computational burden. Hence, to ensure fair comparison, Table 5 shows the number of Neural Function Evaluations (NFEs) used by each method to produce a single explanation, divided into a. model (*U-Net*, *classifier* and *other*) b. and *forward / backward* passes, where the backward pass is around $2\times$ more computationally demanding than the forward pass. Importantly, this comparison eliminates the differences stemming from the utilized hardware and optimality of the implementation, and is fair as each method consumes virtually the same amount of GPU memory. One exception to the use of the standard U-Net model is the LDCE method of Farid et al. (2023), which applies it in the latent space of an autoencoder. However, as each latent U-Net step also requires decoding the image with the decoder, the computational demand stays similar to the standard approach.

As indicated by Table 5, RCSB is the most efficient approach, both in terms of balancing the use of the U-Net and the classifier, and the number of forward/backward passes. Importantly, the *other* category shows non-zero numbers only for the DVCE method, which additionally uses the gradients of a robust classifier in the generation process. The high number of forward/backward passes through the classifiers in DVCE stems from applying them to a set of 16 augmented versions of $\mathbf{x}_t$ at each timestep.

| Inpainting method | NFE | | | | | |
| --- | --- | --- | --- | --- | --- | --- |
| | U-Net | | Classifier | | Other | |
| | forward | backward | forward | backward | forward | backward |
| RCSB | 100 | 100 | 100 | 100 | 0 | 0 |
| LDCE | 191 | 191 | 191 | 191 | 0 | 0 |
| DVCE | 200 | 200 | 1600 | 1600 | 1600 | 1600 |
| ACE | 520 | 500 | 25 | 25 | 0 | 0 |
| DDRM | 200 | 200 | 200 | 200 | 0 | 0 |
| MCG | 1000 | 1000 | 1000 | 1000 | 0 | 0 |
| RePaint | 2410 | 2410 | 2410 | 2410 | 0 | 0 |

Table 5: Number of NFEs for each respective method with details about the model type and forward/backward passes.

### F.8 ADDITIONAL QUANTITATIVE RESULTS

For most visually appealing results, we found $a \approx 0.1 - 0.15$, $c \approx 8 - 16$, $\tau \approx 0.3 - 0.6$ and $s \approx 2 - 3$ to perform the best. These hyperparameters were used to create RVCEs for Fig. 5. To assess that the performance of these configurations does not deviate from best configurations of Table 1, we followed the same evaluation protocol on the most challenging Zebra–Sorrel task and include the results in Table 6. Crucially, the performance stays virtually the same when comparing with Table 1.

| Area $a$ | Cell size $c$ | Guidance scale $s$ | Trajectory truncation $\tau$ | FID | sFID | $S^3$ | COUT | FR |
| --- | --- | --- | --- | --- | --- | --- | --- | --- |
| 0.1 | 16 | 3.0 | 0.2 | 10.1 | 18.4 | 0.92 | 0.79 | 95.8 |
| | | | 0.3 | 10.7 | 19.0 | 0.91 | 0.76 | 95.0 |
| | | | 0.4 | 10.8 | 18.9 | 0.90 | 0.74 | 94.3 |
| | | 3.5 | 0.2 | 11.0 | 19.2 | 0.91 | 0.81 | 97.0 |
| | | | 0.4 | 11.4 | 19.4 | 0.89 | 0.77 | 96.0 |
| | | | 0.3 | 11.6 | 19.7 | 0.90 | 0.79 | 96.2 |
| | | 4.0 | 0.2 | 11.7 | 19.8 | 0.91 | 0.83 | 97.2 |
| | | | 0.4 | 12.3 | 20.2 | 0.88 | 0.79 | 96.7 |
| | | | 0.3 | 12.4 | 20.4 | 0.89 | 0.80 | 96.2 |
| 0.15 | 16 | 2.0 | 0.4 | 11.2 | 19.2 | 0.89 | 0.77 | 97.7 |
| | | | 0.5 | 12.1 | 20.0 | 0.88 | 0.74 | 96.5 |
| | | | 0.6 | 12.7 | 20.4 | 0.86 | 0.70 | 94.8 |
| | | 3.0 | 0.4 | 13.5 | 21.3 | 0.87 | 0.82 | 99.5 |
| | | | 0.5 | 13.9 | 21.7 | 0.86 | 0.80 | 99.2 |
| | | | 0.6 | 14.2 | 21.7 | 0.85 | 0.78 | 98.6 |
| | | 4.0 | 0.4 | 15.3 | 22.9 | 0.86 | 0.84 | 99.6 |
| | | | 0.5 | 15.5 | 23.1 | 0.85 | 0.82 | 99.7 |
| | | | 0.6 | 15.8 | 23.5 | 0.83 | 0.81 | 99.4 |

Table 6: Quantitative results for hyperparameters that provide the most visually appealing results.

### F.9 FREEFORM MASKS

Figure 11 presents example RVCEs for the *sorrel → zebra* task from the experiments based on freeform masks with quantitative results in Table 2(B). We observe that RCSB focuses on modifying the intersection of the randomly assigned mask with features that should intuitively be important to the classifier, while leaving the unimportant parts, like background and sky, mostly unchanged. This provides additional justification for high performance of RCSB despite not exclusively focusing on the most important regions.

### F.10 UNINTUITIVE CLASSES

Figure 12 shows RVCEs obtained for unintuitive class pairings. Interestingly, RCSB is able to largely preserve the realism of the explanations, while providing unusual compositions of objects, *e.g.* placing a maltese dog in place of cauliflower.

## F.11 SHAPE MODIFICATION

While VCEs, and RVCEs in particular, are focused on providing the minimal semantic change that modifies the classifier's decision, one may be interested in obtaining new content that deviates further from the original. Figure 13 shows example RVCEs focused on modifying the original image to a larger extent, leading to shape and contour changes. By increasing the area ($a$) of the region constraint and the trajectory truncation ($\tau$), RCSB allows for weaker preservation of the original content, leading to more visible changes. Modifying the shape of the original objects is possible in various scenarios. For example, in the *white stork $\to$ black stork* task, despite the fact that the bird's color is the dominant differentiating feature, guiding RCSB with the classifier of interest can also lead to large shape changes. This is also visible in the *vulture $\to$ flamingo* task, where the bird's legs appear thinner and longer, while its modified head points to an opposite direction. Moreover, ImageNet contains task that are mostly characterized by shape differences rather than color or texture changes. Figure 13 shows that the primary characteristic of a pretzel can be easily modified to change the model's decision to *bagel*. The same can be seen when changing the decision from *hatchet* to *hammer*, where the latter's back part becomes longer than the original. Moreover, objects like paperknife and spoon can be easily modified with RCSB to be predicted as spoon and ladle respectively through realistic shape changes.

To assess the effectiveness of RCSB in tasks, where the classifier's decision should mostly depend on the objects shape rather than texture or color, we evaluate it on another three tasks: *pretzel$\to$bagel*, *hatchet$\to$hammer* and *paperknife$\to$wooden spoon*. To compare with previous SOTA on ImageNet, we evaluate DVCE in the same scenario. The hyperparameters of both methods and experimental details follow those from the main evaluation protocol in Table 1. Results from Table 7 show that, despite the different nature of the considered tasks, RCSB preserves its performance and advantage over DVCE from Table 1.

| | Pretzel $\to$ Bagel | | | | | Hatchet $\to$ Hammer | | | | | Paperknife $\to$ Wooden spoon | | | | |
|---|---|---|---|---|---|---|---|---|---|---|---|---|---|---|---|
| Method | FID | sFID | $S^3$ | COUT | FR | FID | sFID | $S^3$ | COUT | FR | FID | sFID | $S^3$ | COUT | FR |
| DVCE | 34.3 | 43.9 | 0.59 | 0.37 | 77.4 | 31.2 | 39.8 | 0.66 | 0.43 | 92.8 | 29.1 | 35.4 | 0.69 | 0.41 | 88.2 |
| RCSB$^A$ | 11.4 | 22.9 | 0.86 | 0.84 | 97.2 | 9.8 | 15.4 | 0.91 | 0.89 | 97.8 | 9.9 | 18.2 | 0.86 | 0.88 | 98.9 |

Table 7: Quantitative results for tasks focused on characteristics connected to shape instead of texture or color. RCSB is compared to DVCE, which is regarded as current SOTA on ImageNet.

We also address the topic of shape modification further in Appendix F.14, where RCSB is evaluated on the MNIST dataset. In grayscale handwritten digits, individual classes are primarily identified based on the shape of samples, hence serving as a proper proof-of-concept benchmark for evaluating the understanding of shapes by the method.

## F.12 LOWER-LEVEL ATTRIBUTIONS

To verify whether RCSB is able to effectively synthesize RVCEs based on pixels that are considered less important to the classifier, we extract the regions using our automated approach by first zeroing out the absolute attributions above some quantile $q$ before converting it to a binary mask with the approach mentioned at the end of Section 3.

Table 9 shows the results of this experiment performed on the three main tasks from ImageNet, where we follow the default protocol from the main part of this paper and use the RCSB$^A$ hyperparameter configuration from Table 1. We evaluate the RVCEs resulting from $q \in \{0.3, 0.4, 0.5, 0.6, 0.7, 0.8, 0.9\}$, where the extracted region covers 10% of total image area. Crucially, the presented results showcase that RCSB's is largely preserved despite using pixels which should influence the classifier much more weakly. FID and sFID both decrease monotonically with respect to $q$, which indicates that an increasing number of pixels from the background gets modified, resulting in a smaller number of changed data characteristics. $S^3$ remains mostly unchanged, meaning that the representational similarity is not influenced by varying $q$. Intuitively, both COUT and FR also decrease when picking smaller $q$. This is because RCSB is not able to utilize the most influential pixels for the classifier, which makes the task harder, thus lowering both the sparsity (changing most influential pixels) and efficiency of flipping the model's decision.

| $q$ | | Zebra – Sorrel | | | | | Cheetah – Cougar | | | | | Egyptian Cat – Persian Cat | | | |
|---|---|---|---|---|---|---|---|---|---|---|---|---|---|---|---|
| Metric | FID | sFID | $S^3$ | COUT | FR | FID | sFID | $S^3$ | COUT | FR | FID | sFID | $S^3$ | COUT | FR |
| 0.9 | 5.2 | 13.8 | 0.87 | 0.62 | 91.6 | 6.1 | 17.1 | 0.92 | 0.88 | 99.8 | 13.5 | 31.4 | 0.86 | 0.89 | 99.9 |
| 0.8 | 4.6 | 13.3 | 0.87 | 0.53 | 88.0 | 4.4 | 15.7 | 0.91 | 0.80 | 99.1 | 11.0 | 28.6 | 0.85 | 0.84 | 99.3 |
| 0.7 | 3.9 | 12.7 | 0.88 | 0.48 | 85.7 | 3.5 | 15.1 | 0.91 | 0.74 | 97.4 | 9.2 | 26.9 | 0.85 | 0.79 | 98.5 |
| 0.6 | 3.6 | 12.4 | 0.89 | 0.45 | 82.4 | 2.9 | 14.6 | 0.92 | 0.68 | 94.3 | 7.8 | 25.5 | 0.85 | 0.73 | 96.1 |
| 0.5 | 3.6 | 12.4 | 0.89 | 0.40 | 78.5 | 2.7 | 14.5 | 0.92 | 0.65 | 93.1 | 6.9 | 24.7 | 0.86 | 0.69 | 94.6 |
| 0.4 | 3.7 | 12.3 | 0.89 | 0.38 | 77.7 | 2.8 | 14.6 | 0.92 | 0.65 | 93.4 | 6.6 | 24.5 | 0.87 | 0.65 | 93.1 |
| 0.3 | 3.9 | 12.6 | 0.89 | 0.40 | 79.2 | 3.1 | 14.9 | 0.92 | 0.69 | 95.7 | 7.5 | 25.3 | 0.86 | 0.67 | 94.2 |

Table 8: Quantitative results for lower-level attributions on three main tasks from ImageNet. Here, $q$ denotes the value of the quantile above which absolute attributions are zeroed out before extracting a region with 10% area coverage.

### F.13  OTHER CLASSIFIERS

In Table 9, we provide quantitative results for other classifiers mentioned at the end of the experimental evaluation. The experimental setting follows the same protocol as the one from Table 1 and we use the RCSB$^A$ configuration. Note that the results are consistent with those for ResNet50 in Table 1 for all classifiers except the robust Madry ResNet50. Since we use a single hyperparameter configuration, this is probably due to the different nature of the model, and the results could be easily improved by tuning a specific configuration for it.

| | Zebra – Sorrel | | | | |
|---|---|---|---|---|---|
| Classifier | FID | sFID | $S^3$ | COUT | FR |
| ClipZeroShot | 4.13 | 12.76 | 0.90 | 0.93 | 100.0 |
| ConvNeXtBase | 15.69 | 23.55 | 0.82 | 0.84 | 99.8 |
| MadryResNet50 | 47.49 | 55.22 | 0.65 | −0.19 | 36.2 |
| RBDeiT | 10.00 | 17.76 | 0.83 | 0.70 | 94.0 |
| RBXCiT | 16.04 | 23.45 | 0.79 | 0.46 | 83.6 |
| SwinB | 3.20 | 12.19 | 0.94 | 0.50 | 88.0 |
| VGG16 | 7.29 | 15.39 | 0.88 | 0.84 | 98.0 |
| VGG16_BN | 5.44 | 13.53 | 0.91 | 0.87 | 99.9 |
| ViTB16 | 8.60 | 16.84 | 0.86 | 0.80 | 98.9 |

Table 9: Quantitative results for other classifiers evaluated on the most challenging (out of the three considered) *zebra – sorrel* task from ImageNet.

### F.14  OTHER BENCHMARKS

We extend the evaluation of RCSB with three additional datasets: CelebA-HQ (Karras et al., 2018) with 30 000 samples of $256 \times 256$ resolution face images, CelebA (Liu et al., 2015) with around 200 000 samples of $128 \times 128$ resolution face images, and MNIST (Deng, 2012) with 70 000 samples of $32 \times 32$ resolution images of handwritten digits. The first two datasets are chosen to compare with all previously published diffusion-based approaches that did not evaluate on ImageNet, effectively complementing our experimental results. While both of these datasets contain face images, they pose unique challenges, since CelebA-HQ contains much less samples ($\sim 6\times$) which are also of higher resolution, making the predictive tasks very different. The MNIST dataset was chosen to provide additional proof of the versatility of RCSB. While this dataset is no longer in active use as an evaluation benchmark, we include it here to show that even on the data of much different nature and resolution, RCSB is still able to provide meaningful and informative RVCEs that must focus on modifying the shape of a digit, since there is no notion of 'texture' present in the data.

For both CelebA and CelebA-HQ, we follow previous works and provide explanations for the DenseNet121 (Huang et al., 2017) model trained in a multilabel scenario consisting of 40 distinct attributes. We consider two tasks evaluated in prior works, i.e., flipping the *smile* and *age* classes to the opposite prediction. For MNIST, we train LeNet (Lecun et al., 1998) from scratch using the default training and validation splits. We note that, for each considered dataset, I2SB must be trained independently for the task of inpainting (on $20\% - 30\%$ freeform masks). We start its training using a pretrained diffusion checkpoint a. from Lugmayr et al. (2022) for CelebA-HQ with default training hyperparameters from Liu et al. (2023a) and 40 000 iterations, b. from Jeanneret et al. (2022)

for CelebA with default training hyperparameters from Liu et al. (2023a) and 100 000 iterations and c. from scratch on MNIST.

To provide a more comprehensive comparison, we also adapt the implementation of DiME (Jean-neret et al., 2022) to ImageNet and FastDiME to both CelebA-HQ and ImageNet. We first tune the hyperparameters of both methods with a large grid search on a small subset of images, and then evaluate the best configuration using the standard protocol for ImageNet (i.e., the same as our method) and 2048 samples from CelebA-HQ. Both DiME and FastDiME are implemented with the use of the same checkpoints that the training of I2SB starts from, i.e. from the work of Lugmayr et al. (2022) for CelebA-HQ and Dhariwal & Nichol (2021) for ImageNet.

| Method | FID | sFID | $S^3$ | COUT | FR |
|---|---|---|---|---|---|
| **Zebra – Sorrel** | | | | | |
| DiME | 222.85 | 243.16 | 0.19 | $-0.31$ | 0.0 |
| FastDiME | 96.48 | 103.45 | 0.22 | $-0.44$ | 14.0 |
| RCSB$^A$ | 8.0 | 16.2 | 0.88 | 0.74 | 94.7 |
| **Cheetah – Cougar** | | | | | |
| DiME | 268.22 | 291.99 | 0.11 | $-0.16$ | 0.0 |
| FastDiME | 133.01 | 141.12 | 0.12 | $-0.11$ | 18.0 |
| RCSB$^A$ | 17.2 | 26.6 | 0.92 | 0.92 | 100.0 |
| **Egyptian Cat – Persian Cat** | | | | | |
| DiME | 322.79 | 352.08 | 0.44 | $-0.05$ | 0.0 |
| FastDiME | 193.63 | 207.12 | 0.10 | 0.01 | 20.0 |
| RCSB$^A$ | 23.0 | 40.0 | 0.87 | 0.92 | 100.0 |

Table 10: Quantitative results on the ImageNet dataset for DiME and FastDiME.

In terms of evaluation measures, we follow previous works (Jeanneret et al., 2022; 2023; Weng et al., 2024) and utilize FVA, FS (Cao et al., 2018), MNAC (Rodríguez et al., 2021) and CD (Jeanneret et al., 2022) in addition to metrics used on ImageNet.

Tables 11 and 12 show the quantitative results achieved by all considered methods on CelebA and CelebA-HQ. Importantly, RCSB is able to outperform the current SOTA in many cases. For example, it provides new records for COUT, MNAC and FR on all considered (dataset, class) pairs. While high COUT and FR are expected based on the method's performance on ImageNet, where it is able to efficiently generate very sparse RVCEs (with respect to the classifier), low MNAC additionaly shows that RCSB focuses on a small subset of face attributes and leaves others mostly unmodified. Our approach is also able to obtain very low FID and sFID values, often performing worse than ACE only, which indicates that the obtained explanations preserve the realism of the original samples. We include example RVCEs obtained with RCSB on CelebA-HQ in Fig. 14.

| | Smile | | | | | | | | Age | | | | | | | |
|---|---|---|---|---|---|---|---|---|---|---|---|---|---|---|---|---|
| Method | FID | sFID | FVA | FS | MNAC | CD | COUT | FR | FID | sFID | FVA | FS | MNAC | CD | COUT | FR |
| DiVE | 29.4 | - | 97.3 | - | - | - | - | - | 33.8 | - | 98.1 | - | 4.58 | - | - | - |
| DiVE$^{100}$ | 36.8 | - | 73.4 | - | 4.63 | 2.34 | - | - | 39.9 | - | 52.2 | - | 4.27 | - | - | - |
| STEEX | 10.2 | - | 96.9 | - | 4.11 | - | - | - | 11.8 | - | 97.5 | - | 3.44 | - | - | - |
| ACE $\ell_1$ | 1.27 | 3.97 | 99.9 | 0.87 | 2.94 | 1.73 | 0.78 | 97.6 | 1.45 | 4.12 | 99.6 | 0.78 | 3.20 | 2.94 | 0.72 | 96.2 |
| ACE $\ell_2$ | 1.90 | 4.56 | 99.9 | 0.87 | 2.77 | 1.56 | 0.62 | 84.3 | 2.08 | 4.62 | 99.6 | 0.80 | 2.94 | 2.82 | 0.56 | 77.5 |
| DiME | 3.17 | 4.89 | 98.3 | 0.73 | 3.72 | 2.30 | 0.53 | 97.0 | 4.15 | 5.89 | 95.3 | 0.67 | 3.13 | 3.27 | 0.44 | 99.0 |
| FastDiME | 4.18 | 6.13 | 99.8 | 0.76 | 3.12 | 1.91 | 0.44 | 99.0 | 4.82 | 6.76 | 99.2 | 0.74 | 2.65 | 3.80 | 0.36 | 98.6 |
| FastDiME-2 | 3.33 | 5.49 | 99.9 | 0.77 | 3.06 | 1.89 | 0.44 | 99.4 | 4.04 | 6.01 | 99.6 | 0.75 | 2.63 | 3.80 | 0.37 | 99.3 |
| FastDiME-2+ | 3.24 | 5.23 | 99.9 | 0.79 | 2.91 | 2.02 | 0.41 | 98.9 | 3.60 | 5.59 | 99.7 | 0.77 | 2.44 | 3.76 | 0.32 | 98.7 |
| RCSB | 2.98 | 4.79 | 100.0 | 0.91 | 2.24 | 2.78 | 0.87 | 99.8 | 2.94 | 4.94 | 99.9 | 0.88 | 2.14 | 3.63 | 0.81 | 99.3 |

Table 11: Quantitative results on the CelebA dataset. We extract the results of other methods (Rodríguez et al., 2021; Jacob et al., 2022; Jeanneret et al., 2022; 2023) from the work of Weng et al. (2024).

Regarding the MNIST dataset, we provide example RVCEs obtained with RCSB using the auto-mated region extraction for various tasks in Fig. 15. Crucially, the presented samples show that, despite the complexity of the detected regions, RCSB is able to properly modify the shape of the initial digit to change the classifier's decision. As can be observed, this is performed through a hybrid

| | | | Smile | | | | | | | | Age | | | | | |
|---|---|---|---|---|---|---|---|---|---|---|---|---|---|---|---|---|
| Method | FID | sFID | FVA | FS | MNAC | CD | COUT | FR | FID | sFID | FVA | FS | MNAC | CD | COUT | FR |
| DiVE | 107.0 | - | 35.7 | - | 7.41 | - | - | - | 107.5 | - | 32.3 | - | 6.76 | - | - | - |
| STEEX | 21.9 | - | 97.6 | - | 5.27 | - | - | - | 26.8 | - | 96.0 | - | 5.63 | - | - | - |
| DiME | 18.1 | 27.7 | 96.7 | 0.67 | 2.63 | 1.82 | 0.65 | 97.0 | 18.7 | 27.8 | 95.0 | 0.66 | 2.10 | 4.29 | 0.56 | 97.0 |
| ACE $\ell_1$ | 3.21 | 20.2 | 100.0 | 0.89 | 1.56 | 2.61 | 0.55 | 95.0 | 5.31 | 21.7 | 99.6 | 0.81 | 1.53 | 5.4 | 0.40 | 95.0 |
| ACE $\ell_2$ | 6.93 | 22.0 | 100.0 | 0.84 | 1.87 | 2.21 | 0.60 | 95.0 | 16.4 | 28.2 | 99.6 | 0.77 | 1.92 | 4.21 | 0.53 | 95.0 |
| LDCE | 13.6 | 25.8 | 99.1 | 0.76 | 2.44 | 1.68 | 0.34 | - | 14.2 | 25.6 | 98.0 | 0.73 | 2.12 | 4.02 | 0.33 | - |
| FastDiME-2+ | 16.51 | 31.4 | 99.9 | 0.87 | 1.43 | 4.16 | 0.28 | 87.1 | 26.0 | 40.3 | 99.6 | 0.81 | 3.15 | 4.36 | 0.31 | 92.6 |
| RCSB | 3.04 | 20.0 | 100.0 | 0.93 | 1.22 | 3.22 | 0.83 | 98.9 | 4.92 | 27.3 | 100.0 | 0.96 | 1.47 | 5.16 | 0.80 | 99.4 |

Table 12: Quantitative results on the CelebA-HQ dataset. We extract the results of other methods from the work of Farid et al. (2023) and Jeanneret et al. (2023) except FastDiME which we implement and evaluate ourselves.

approach, where some parts of the initial digit remain the same, while new parts appear to either combine the existing elements of the digit or create entirely new ones.

# G  USER STUDIES

The main goal of VCEs, and RVCEs in particular, is to explain the model's reasoning to humans. This capability can be evaluated from various perspectives. In the following, we provide a detailed analysis of two independently conducted user studies, with the first one focused on the general usefulness of RVCEs in understanding the model's decision-making and the potential benefits stemming from a possible interaction of humans with the explanation creation process, and the second one concerned with a specific use-case, where RVCEs are used to inform the user about the causes of model's misclassification and what must be changed in a given image for it to predict correctly. For both studies, the model of interest is the ResNet50 used in main experimental evaluation and the samples are extracted from ImageNet.

## G.1  I: USEFULNESS AND INTERACTION

In this study, 15 participants with background knowledge in machine learning (at the level of MSc studies, not aware of the research conducted for this paper) were presented with comparisons of VCEs and RVCEs for the same factual images, together with absolute differences between the original and the explanation. An example of such comparison is included in Fig. 16. The participants were asked about which type of explanation is more useful in understanding the model's decision-making. Here, 86.6% answered in favor of RVCEs. Moreover, each user was asked to provide the reasons for their judgement. The answers generally focused on the semantic change being more localized, easier to interpret and better aligned with human intuition.

The second part of this study focused on evaluating the added value provided by the possibility of interacting with the explanation creation process through manual region specification. First, the participants were shown a default interaction with VCEs, i.e., the original image and its VCE were presented to users with no possibility of interacting with it. Then, the participants were presented with the process of manual region specification for which an RVCE was generated, and offered the possibility to provide the region themselves. After that, each participant was asked whether the interactive process may be helpful and more useful in obtaining a better understanding of the model's reasoning than the standard scenario. There, 93.3% of participants answered in favor of the interactive process. They were also asked to justify their choice, and the answers generally focused on the possibility of verifying regions that align with human understanding and incorporating domain experts that would be able to more thoroughly analyze the model.

In both parts, RVCEs were extracted from the figures in the main part of the manuscript and DVCE was used to generate standard VCEs, since this method provided the best quantitative results prior to our work.

## G.2  II: UNDERSTANDING MODEL'S FAILURES

In this study, a different group of 11 participants with background knowledge in machine learning at a similar level took part in evaluating whether RVCEs help in identifying the reasons for model's

misclassifications. The study consisted of a general introduction to the problem of explaining deep classifiers with VCEs, the concept of RVCEs and the two-fold goal of the study: to understand why a given model misclassifies the image and to identify the minimal semantic change required to correct the prediction. Then, the participants took part in 5 variations of the same experiment, where in each case a different misclassification was shown. The experiment began with a presentation of the factual image, the initially predicted class, the correct class and two sets of images, each representing instances of one of the two classes taken randomly from the web (see Fig. 17(left) for an example introduction to the experiment). Then, each participant observed the original image, the region constraint from our automated approach and the resulting series of RVCEs (see Fig. 17(right) for an example). After each of the 5 variants of this experiment, the participants were asked whether they were able to identify the semantic features that the model lacked in its initial prediction and that lead to correcting the decision once they appeared on the image. The response was, on average, positive in 80.02% of the cases with 14.56% standard deviation. In each case, they were also asked to describe these features. Here, the answers almost always aligned with what the RVCEs were introducing to the image. After all experiments, the participants were asked if RVCEs are able to indicate the semantic features that were missing in the beginning for the model to predict correctly (90.9% positive answers), whether they better understood the initial misclassification (90.9% positive answers) and if they judge RVCE as a useful tool in explaining the model's decision-making (100% positive answers).

The conducted user studies highlight that the concept of RVCEs and the application of RCSB to their generation is preferred by the users in comparison to standard VCEs. Our explanations are found to be more useful and helpful in explaining the model's decision-making. The possibility of interacting with the explanation creation is also enjoyed and recognized to improve the explanatory process by almost all participants. Moreover, our method helped the users in obtaining a better understanding of the classifier's failure cases and its potential causes.

## H    QUALITATIVE EXAMPLES

We provide additional qualitative examples for different scenarios. Results for other classifiers, obtained with the automated region extraction, are depicted in Fig. 21 (VGG16, Simonyan & Zisserman (2015)), Fig. 22 (VGG16 with Batch Normalization (BN), Simonyan & Zisserman (2015)), Fig. 23 (ConvNeXt Base, Liu et al. (2022)), Fig. 24 (ViT-B/16, Dosovitskiy et al. (2021)), Fig. 25 (SwinB, Liu et al. (2021)) , Fig. 26 (robust Madry ResNet50, Engstrom et al. (2019)), Fig. 27 (robust Tian DeiT, Tian et al. (2022)), Fig. 28 (zero-shot CLIP classifier, Radford et al. (2021)). We used $a = 0.2, \tau = 0.7, c = 16, s = 1.5$ universally across all additional classifiers, showcasing the versatility of RCSB. Additional examples for ResNet50 are shown in Figs. 29 to 32 (automated region extraction with different hyperparameters), Fig. 33 (exact regions from LangSAM) and Fig. 34 (user-defined regions).

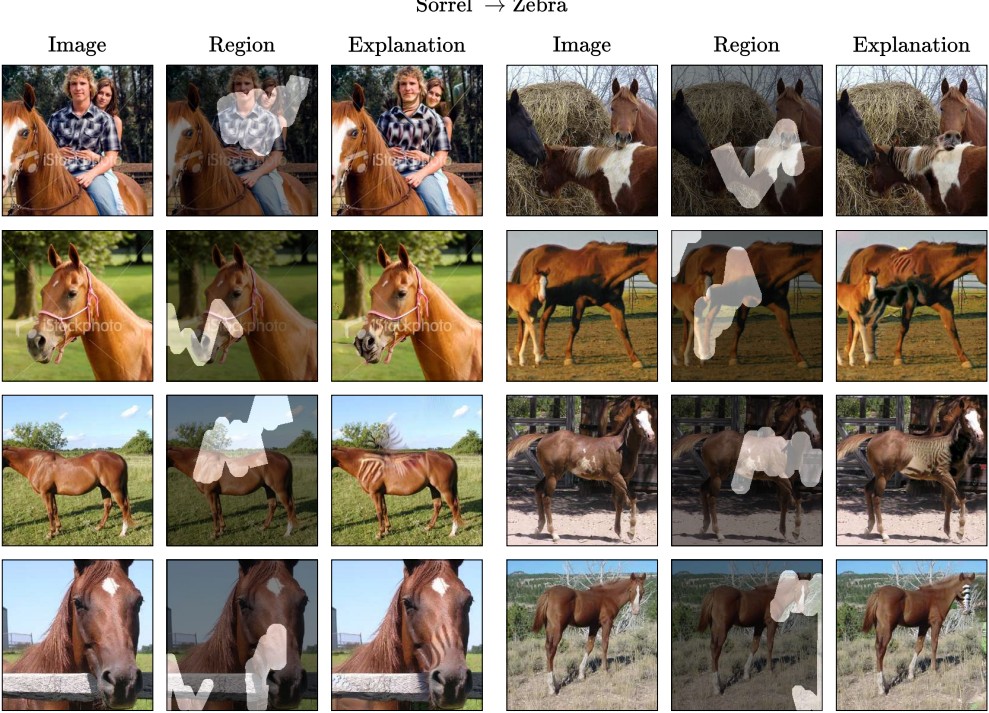

Figure 11: Example RVCEs obtained in the *sorrel → zebra* task from the experiments based on freeform masks with quantitative results in Table 2(B). The columns show the factual image, the region to which changes are constrained and the resulting RVCE.

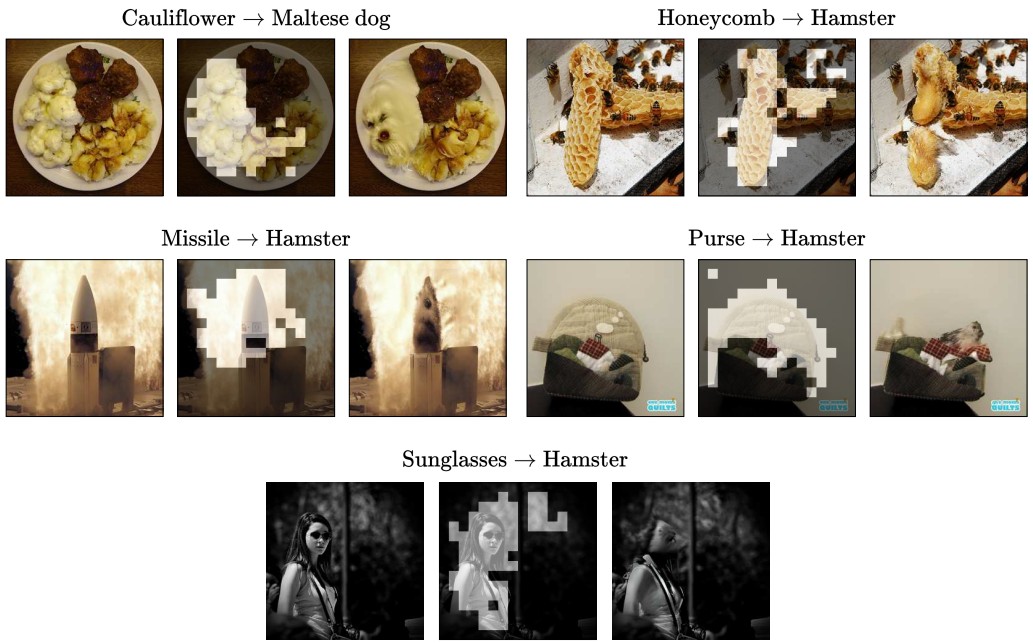

Figure 12: RVCEs for unusual class pairings. Each task of the form *initial class → target class* depicts the factual image, the region constraint and the resulting explanation respectively.

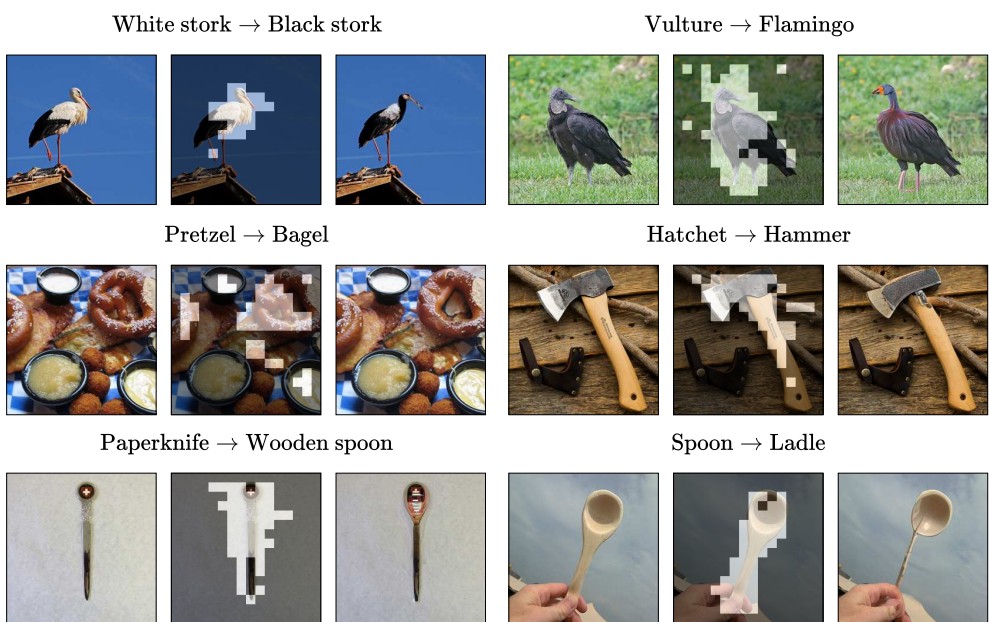

Figure 13: RVCEs focused on modifying the shape of objects within the region constraint obtained with the automated extraction approach. By picking larger area ($a$) and trajectory truncation ($\tau$), the preservation of the original content can be reduced, leading to more diverse infills that change the classifier's decision.

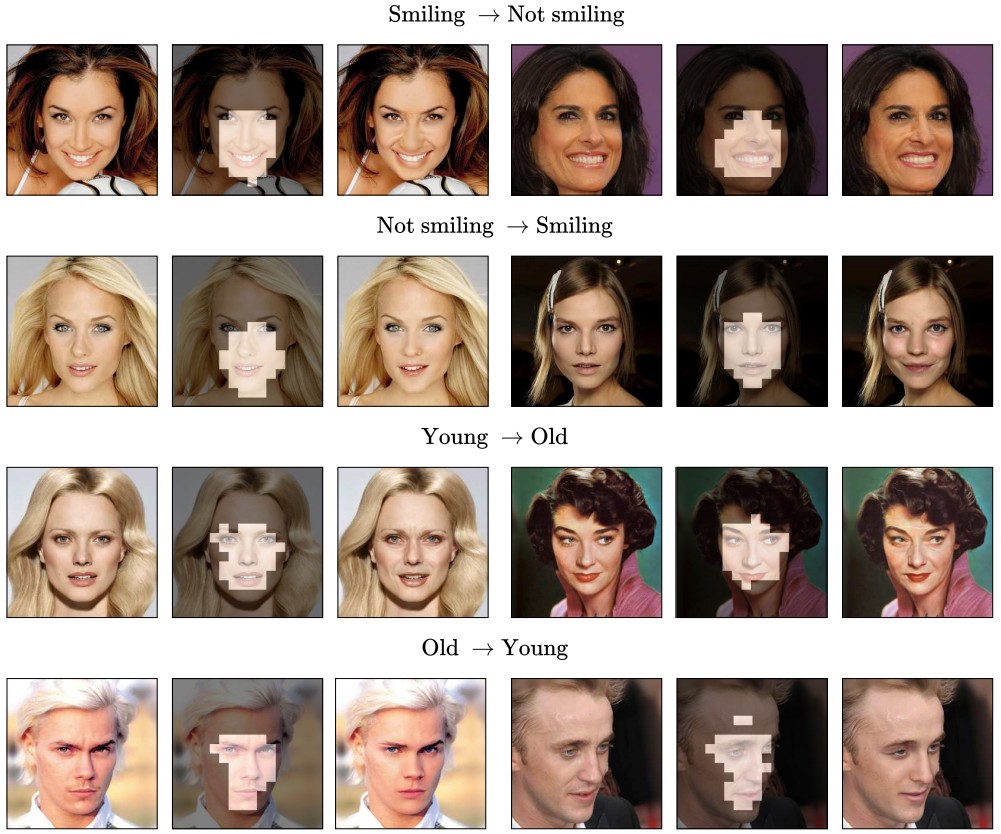

Figure 14: Qualitative examples of RVCEs obtained with RCSB in *smiling↔not smiling* and *young→old* tasks on CelebA-HQ.

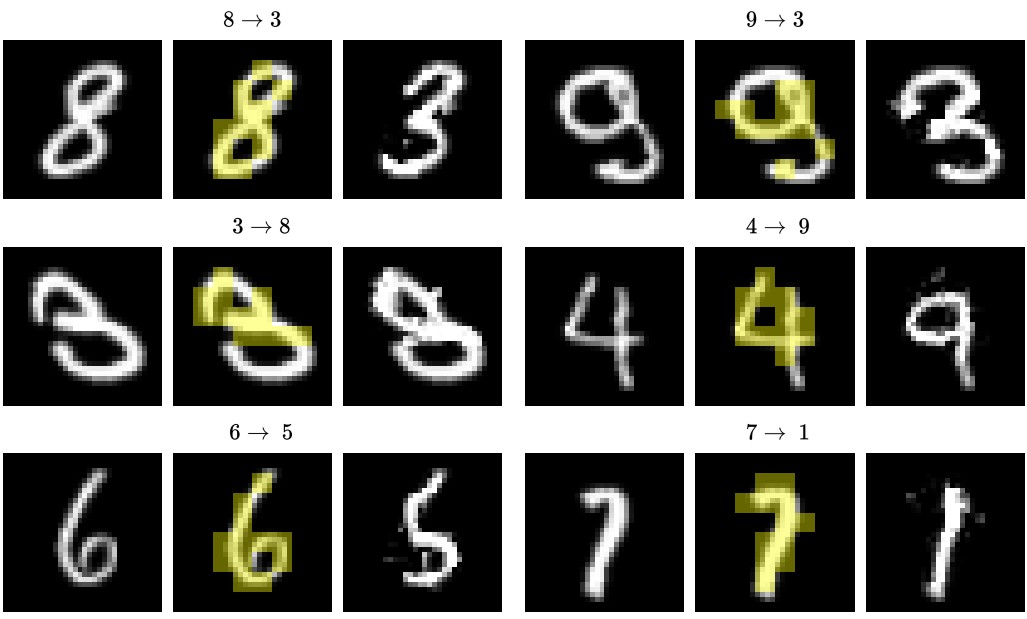

Figure 15: Qualitative examples of RVCEs obtained with RCSB in various digit-flipping tasksk on the MNIST dataset.

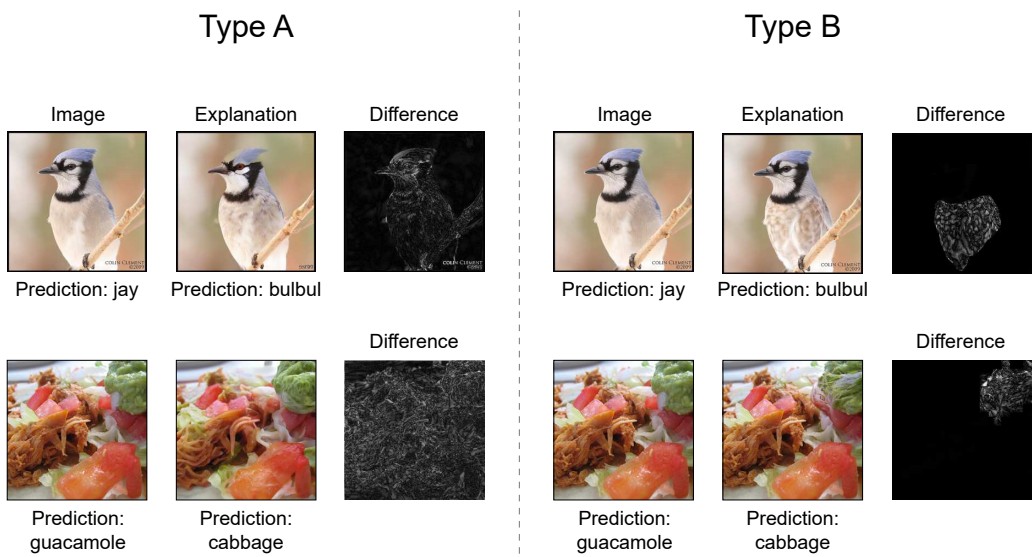

Figure 16: An example comparison of VCEs and RVCEs presented to participants of the I user study.

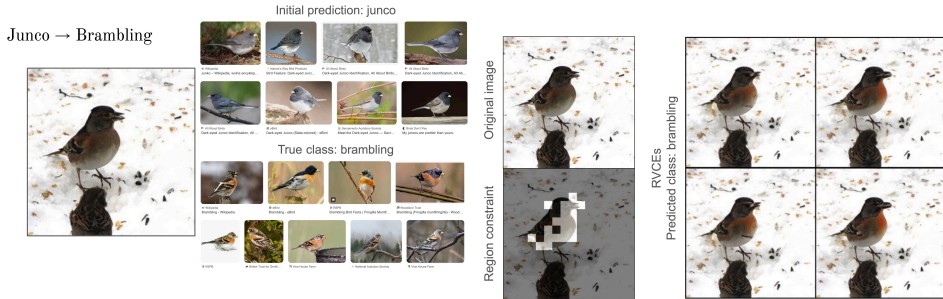

Figure 17: An example introduction (left) and evaluation (right) from one of the experiments conducted in the II user study.

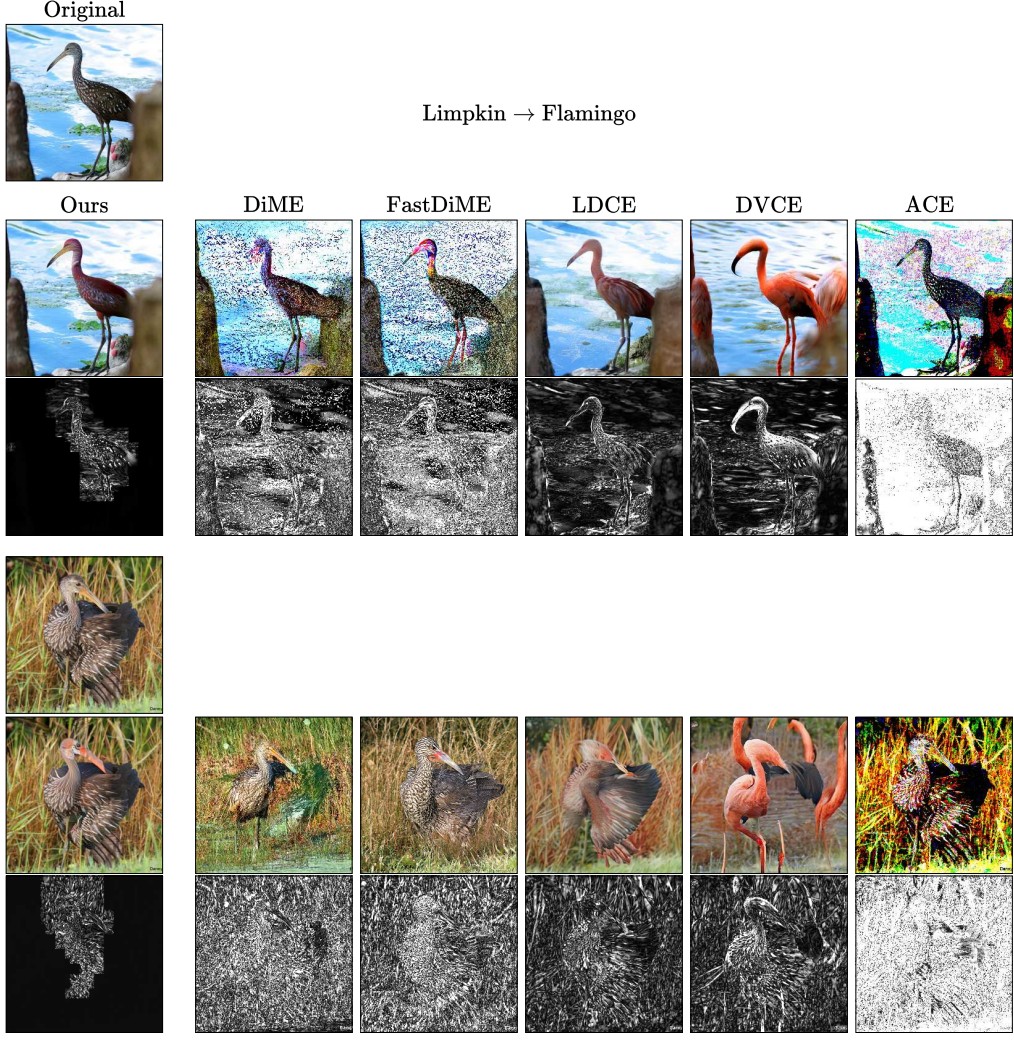

Figure 18: Qualitative comparison to other VCE generation methods: DiME (Jeanneret et al., 2022), FastDiME (Weng et al., 2024), LDCE (Farid et al., 2023), DVCE (Augustin et al., 2022) and ACE (Jeanneret et al., 2023). For each explanation, the absolute difference between the factual image is additionaly provided.

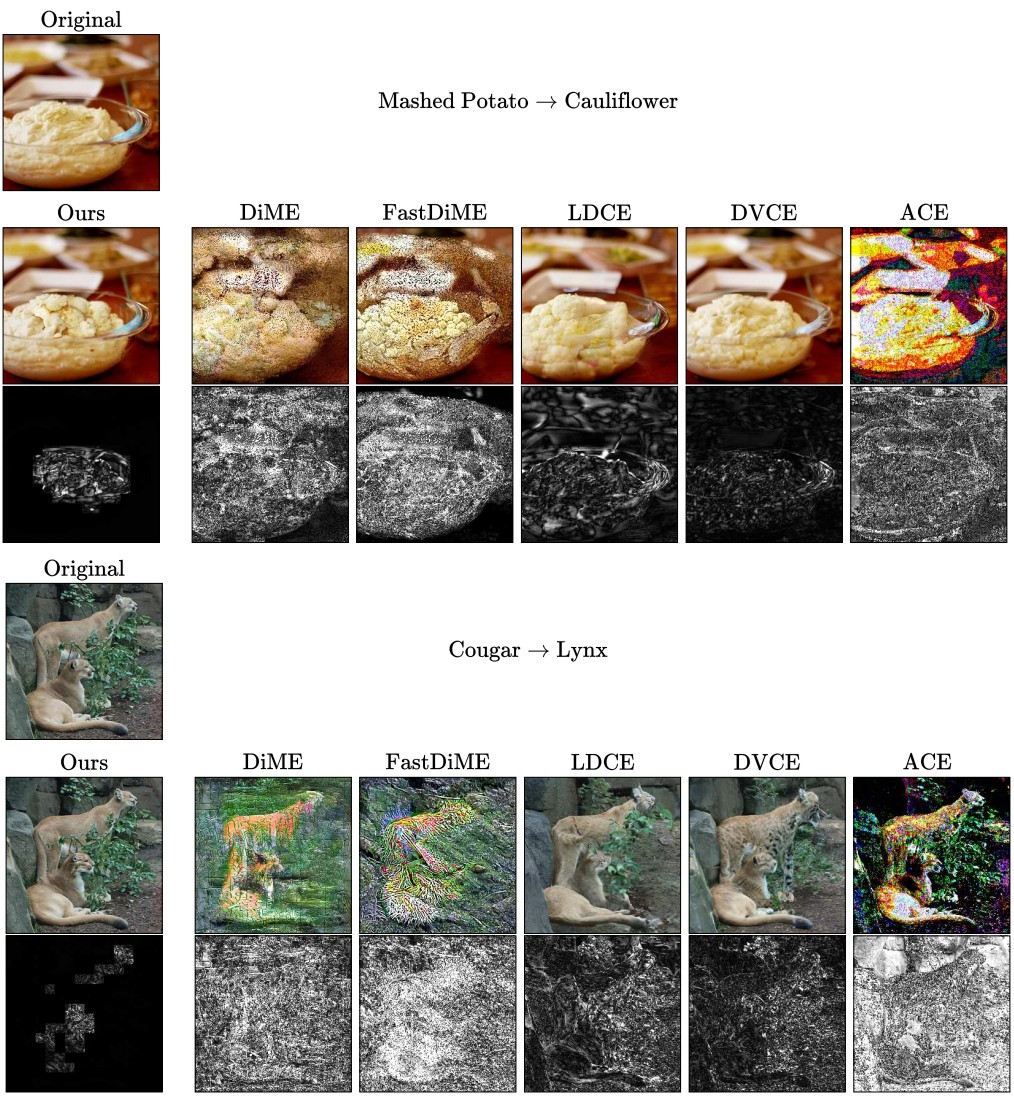

Figure 19: Qualitative comparison to other VCE generation methods: DiME (Jeanneret et al., 2022), FastDiME (Weng et al., 2024), LDCE (Farid et al., 2023), DVCE (Augustin et al., 2022) and ACE (Jeanneret et al., 2023). For each explanation, the absolute difference between the factual image is additionaly provided.

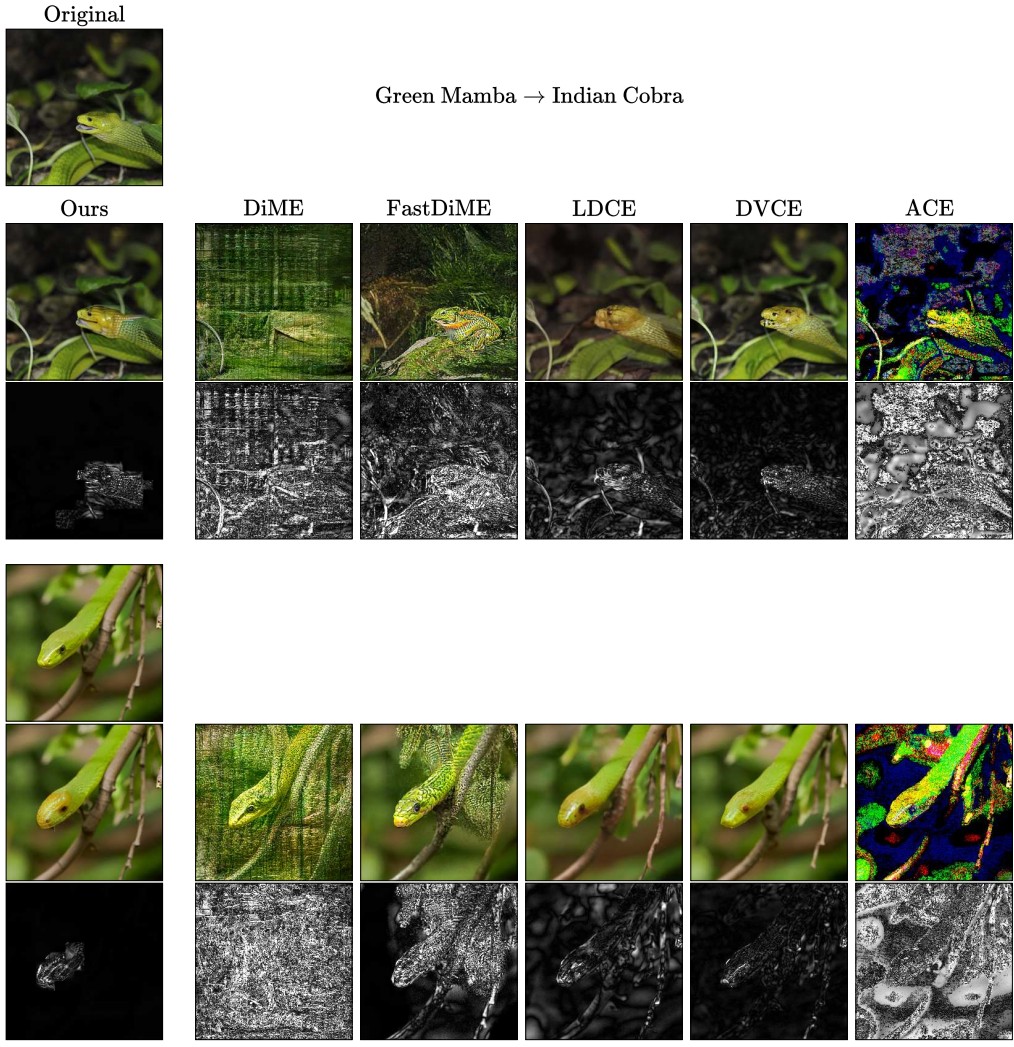

Figure 20: Qualitative comparison to other VCE generation methods: DiME (Jeanneret et al., 2022), FastDiME (Weng et al., 2024), LDCE (Farid et al., 2023), DVCE (Augustin et al., 2022) and ACE (Jeanneret et al., 2023). For each explanation, the absolute difference between the factual image is additionaly provided.

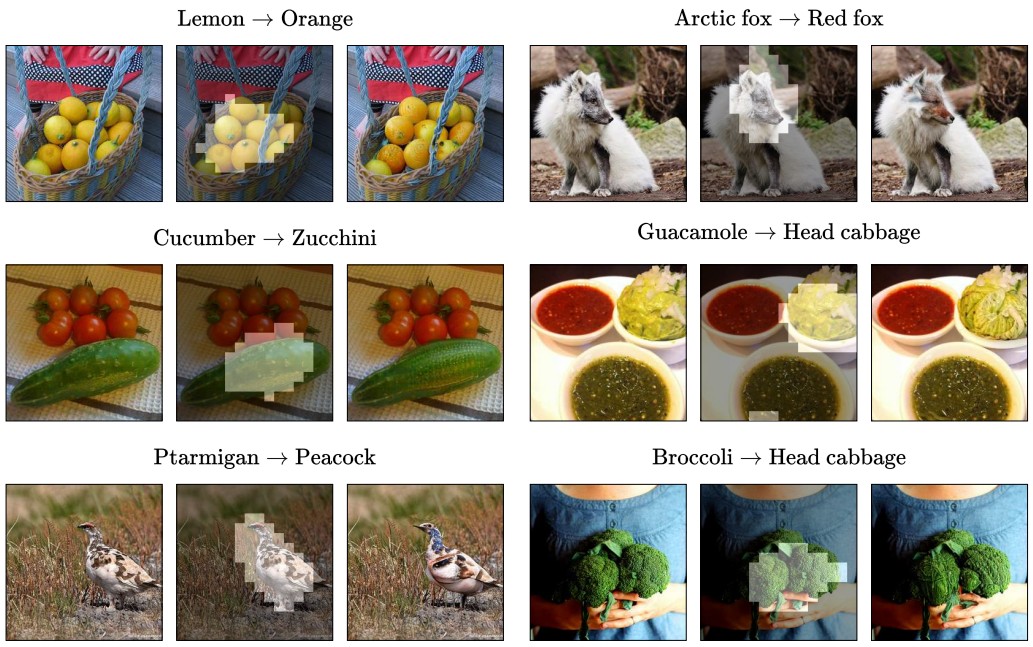

Figure 21: Extended qualitative evaluation of automated region extraction for VGG16 (Simonyan & Zisserman, 2015) classifier. For each task, factual image is shown on the left with the used region in the middle and the generated RVCE on the right.

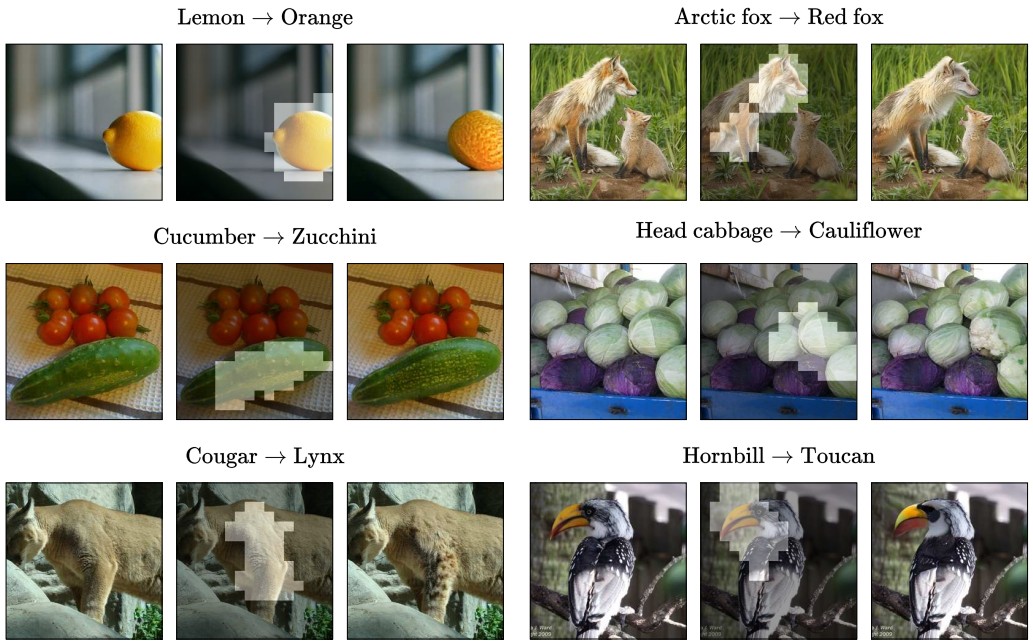

Figure 22: Extended qualitative evaluation of automated region extraction for VGG16BN (Simonyan & Zisserman, 2015) classifier. For each task, factual image is shown on the left with the used region in the middle and the generated RVCE on the right.

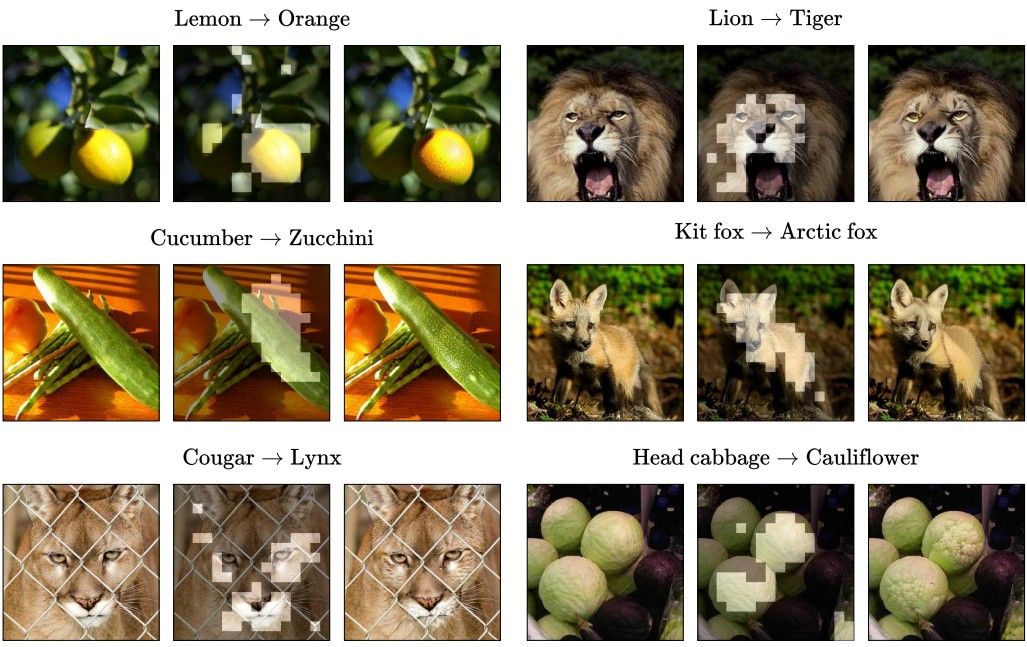

Figure 23: Extended qualitative evaluation of automated region extraction for ConvNeXt Base (Liu et al., 2022) classifier. For each task, factual image is shown on the left with the used region in the middle and the generated RVCE on the right.

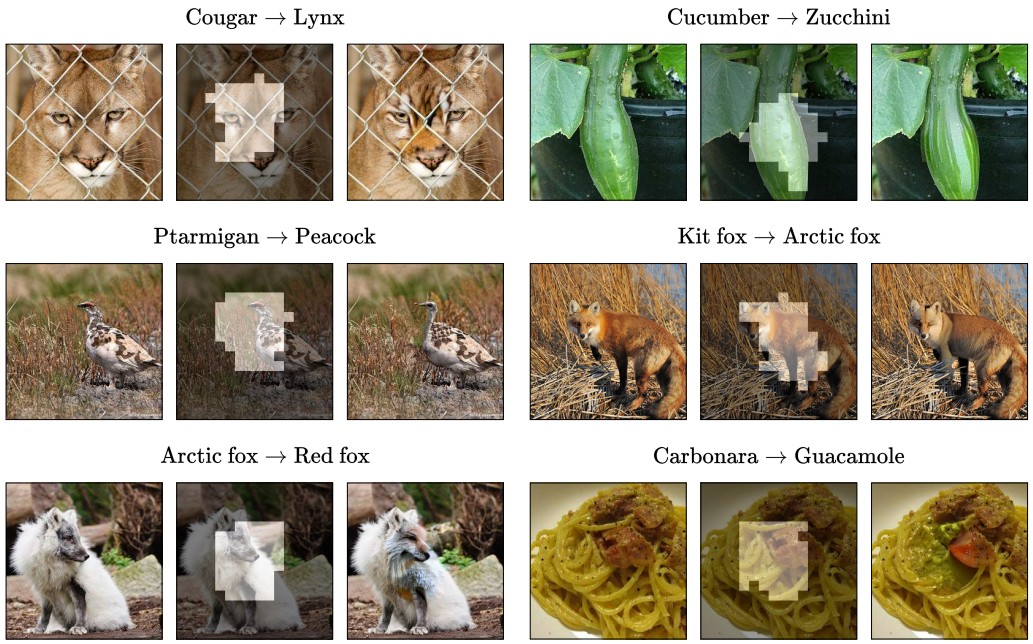

Figure 24: Extended qualitative evaluation of automated region extraction for ViTB16 (Dosovitskiy et al., 2021) classifier. For each task, factual image is shown on the left with the used region in the middle and the generated RVCE on the right.

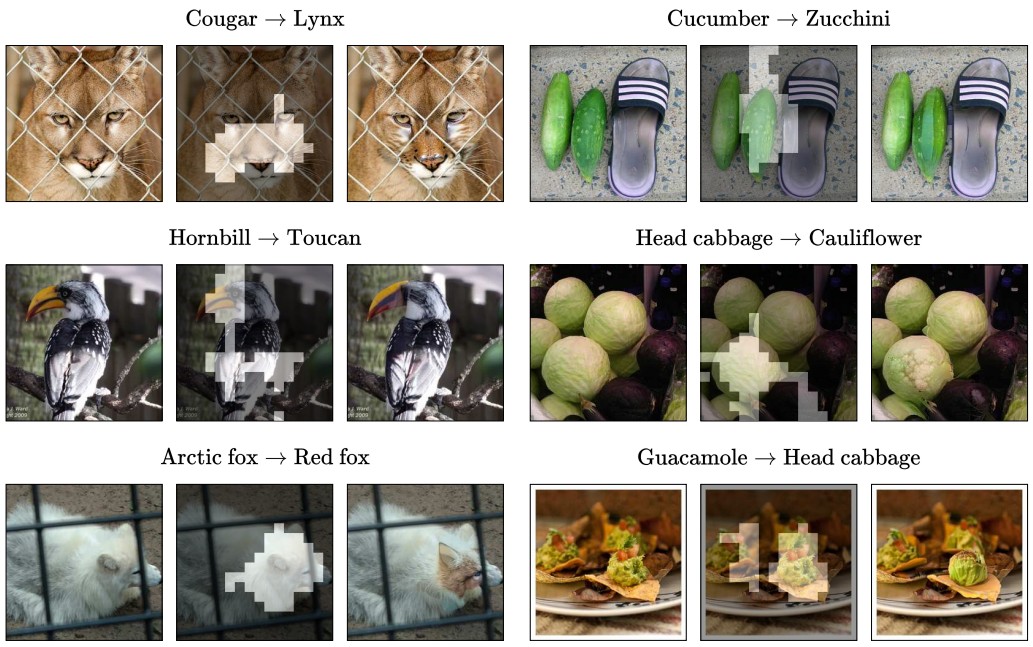

Figure 25: Extended qualitative evaluation of automated region extraction for SwinB (Liu et al., 2021) classifier. For each task, factual image is shown on the left with the used region in the middle and the generated RVCE on the right.

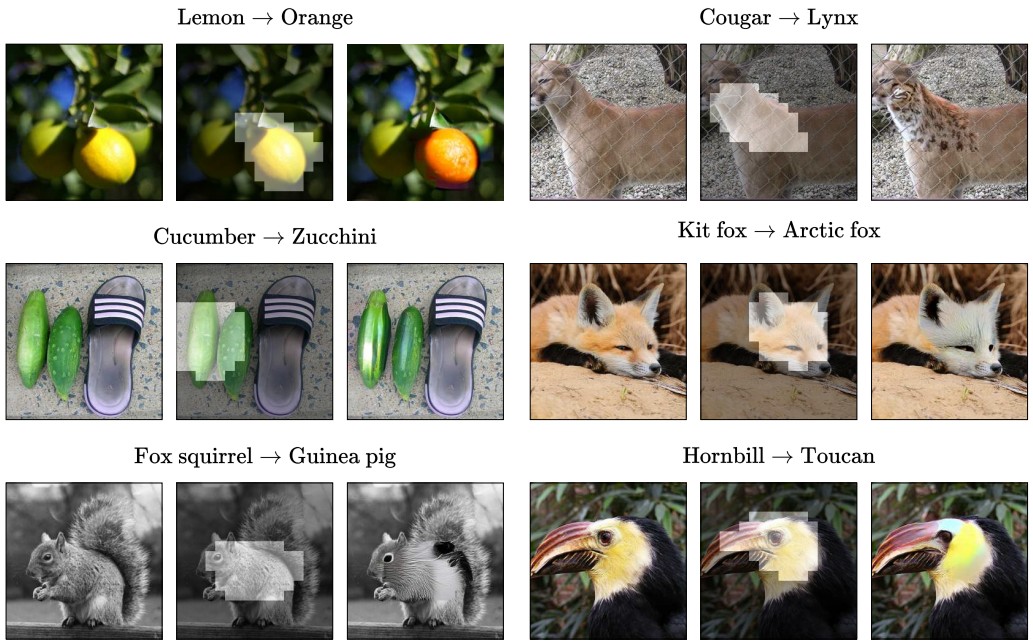

Figure 26: Extended qualitative evaluation of automated region extraction for Madry ResNet50 (Engstrom et al., 2019) $l_2$-norm robust classifier. For each task, factual image is shown on the left with the used region in the middle and the generated RVCE on the right.

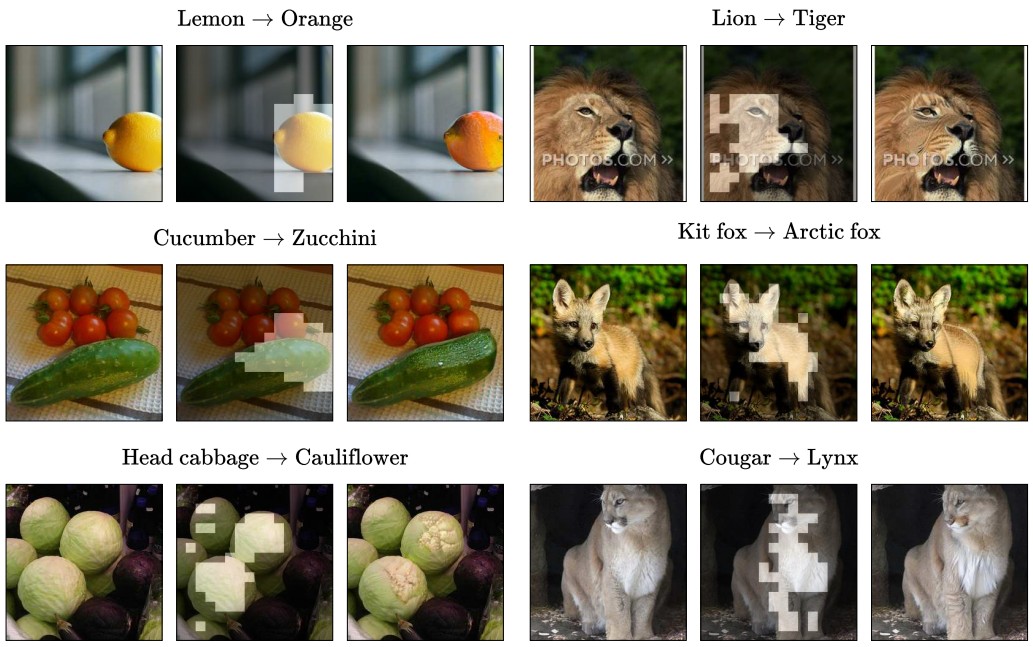

Figure 27: Extended qualitative evaluation of automated region extraction for Tian DeiT (Tian et al., 2022) corruption robust classifier. For each task, factual image is shown on the left with the used region in the middle and the generated RVCE(s) on the right.

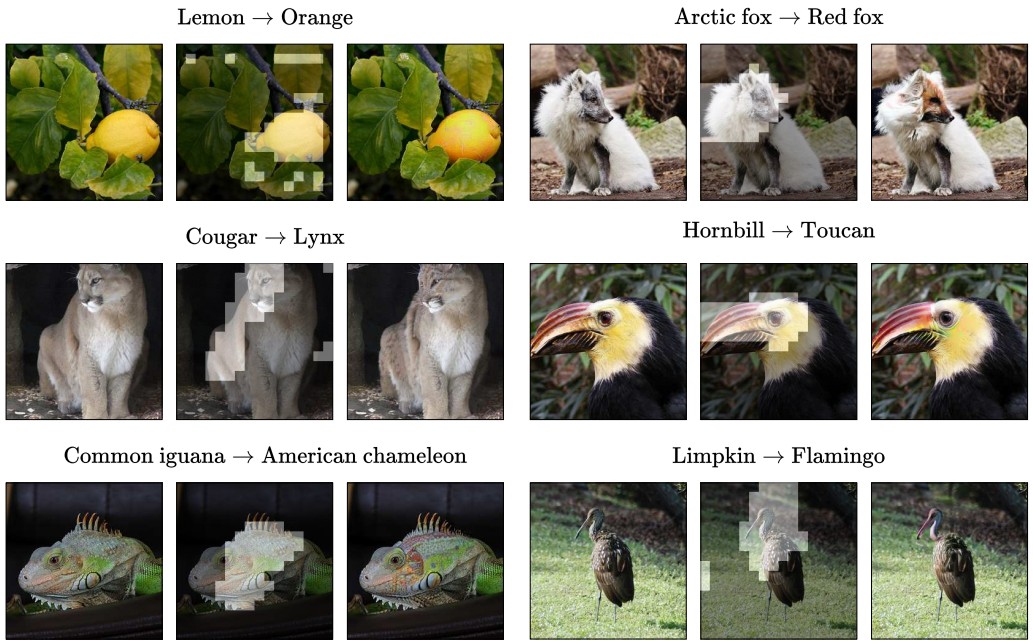

Figure 28: Extended qualitative evaluation of automated region extraction for CLIP ViT-B/32 (Radford et al., 2021) zero-shot classifier. For each task, factual image is shown on the left with the used region in the middle and the generated RVCE on the right.

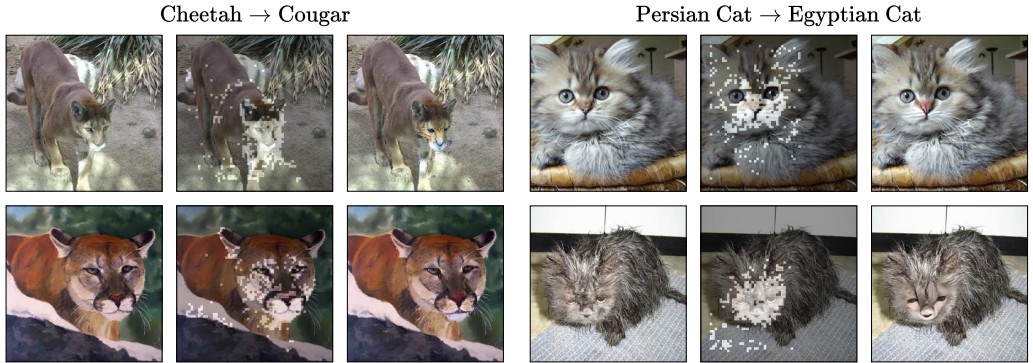

Figure 29: Extended qualitative evaluation of automated region extraction with $c = 4$, $a = 0.1$ for the ResNet50 classifier. For each task, factual image is shown on the left with the used region in the middle and the generated RVCE on the right.

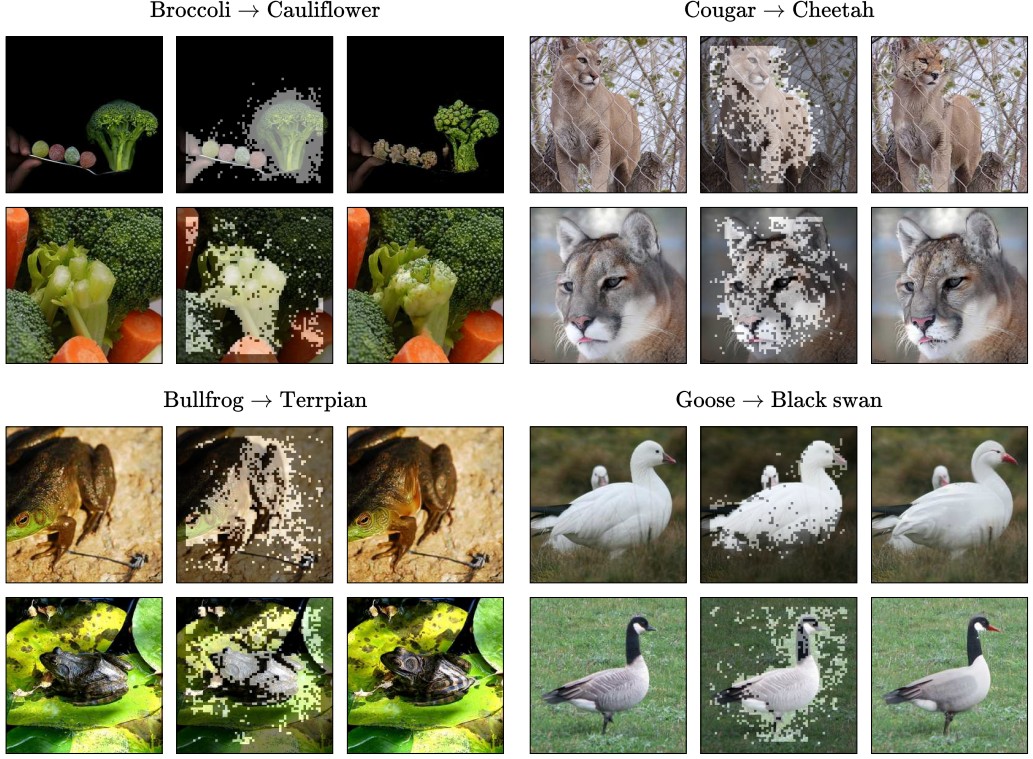

Figure 30: Extended qualitative evaluation of automated region extraction with $c = 4$, $a = 0.3$ for the ResNet50 classifier. For each task, factual image is shown on the left with the used region in the middle and the generated RVCE on the right.

Cheetah → Cougar            Goose → Black swan

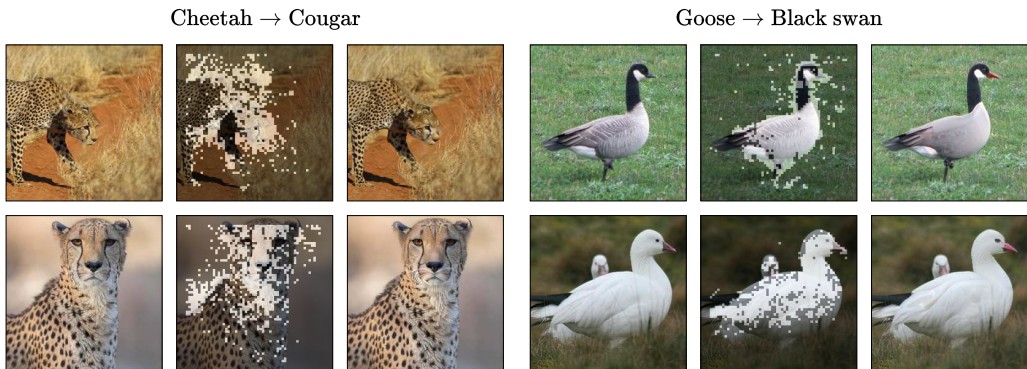

Figure 31: Extended qualitative evaluation of automated region extraction with $c = 4$, $a = 0.2$ for the ResNet50 classifier. For each task, factual image is shown on the left with the used region in the middle and the generated RVCE on the right.

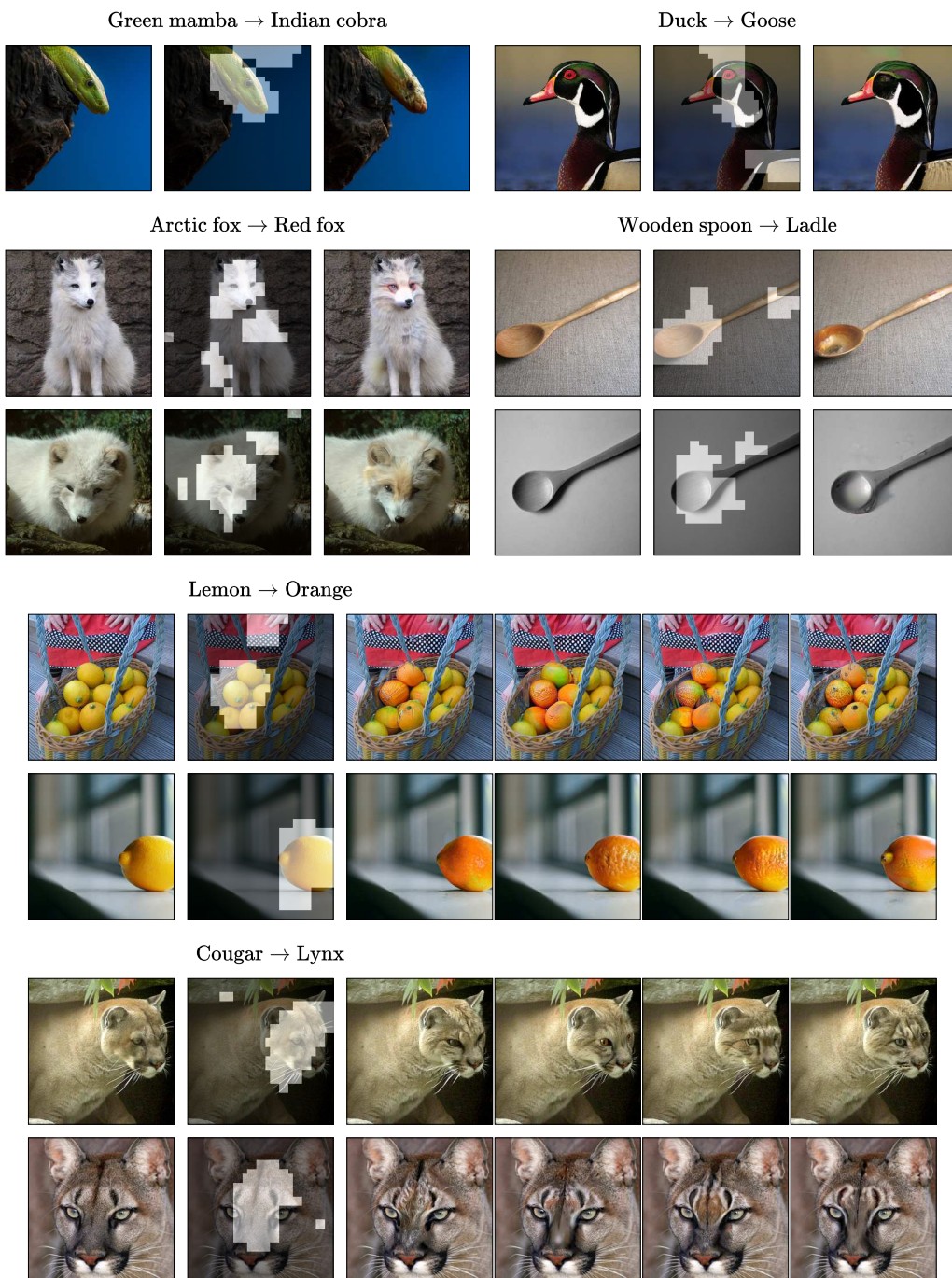

Figure 32: Extended qualitative evaluation of automated region extraction with $c = 8$, $a = 0.2$ for the ResNet50 classifier. For each task, factual image is shown on the left with the used region in the middle and the generated RVCE(s) on the right.

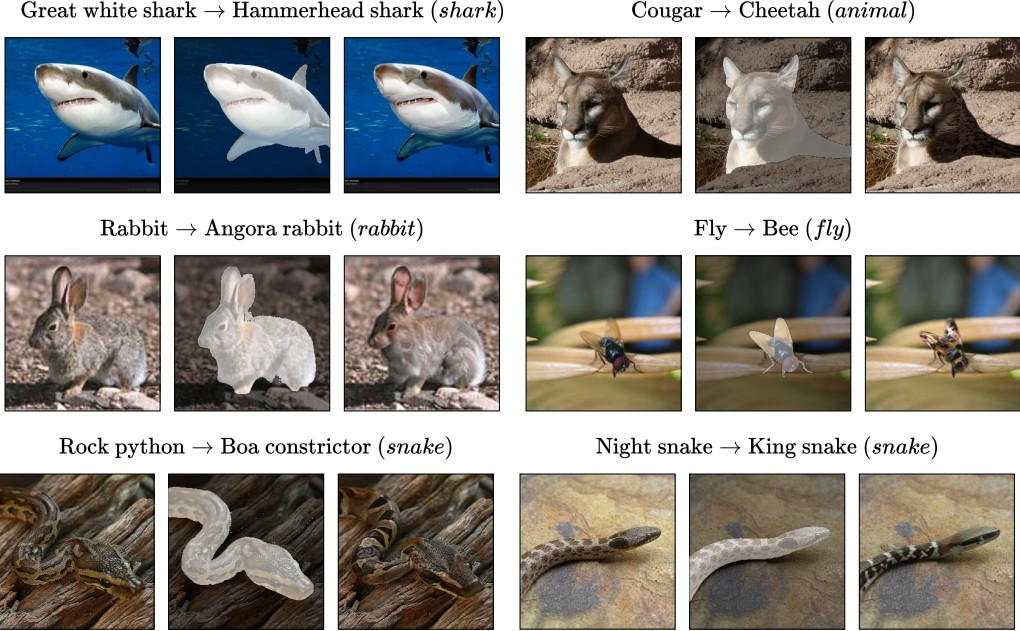

Figure 33: Extended qualitative evaluation of exact regions obtained with LangSAM for the ResNet50 classifier. For each task, factual image is shown on the left with the used region in the middle and the generated RVCE(s) on the right.

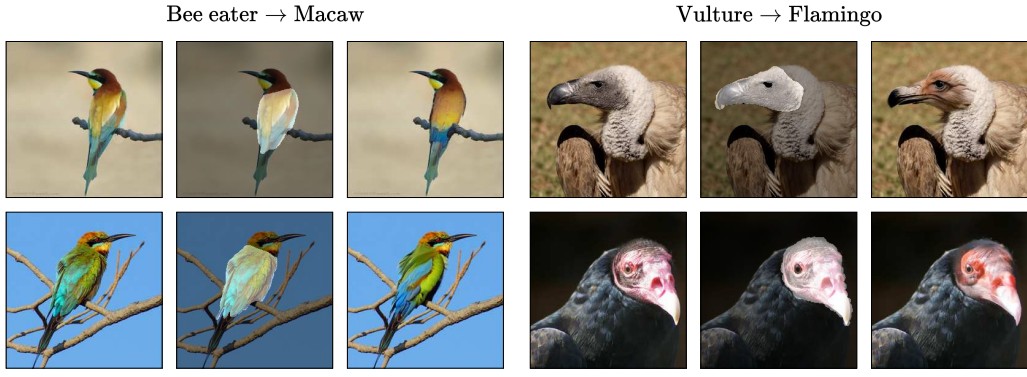

Figure 34: Extended qualitative evaluation of user-defined regions for the ResNet50 classifier. For each task, factual image is shown on the left with the used region in the middle and the generated RVCE(s) on the right.

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
