# OpenReview forum: "Rethinking Visual Counterfactual Explanations Through Region Constraint"
_ICLR.cc/2025/Conference — ICLR 2025 Poster_

### Official Review · Reviewer_TMN9 · 2024-10-25

**Soundness:** 3
**Presentation:** 3
**Contribution:** 2
**Rating:** 6
**Confidence:** 3

**Summary:**

The authors propose a method for visual counterfactual explanation which is based upon methods like latent diffusion counterfactual explanation by Farid et al. and modified the previous approaches by adding a spatial (region) constraint, which allows more precise target generation of new images containing features of the target class. Thereby, also an external classifier is used as guidance for the generation process, e.g. for a stable diffusion model. Furthermore, the authors added optimization techniques to overcome limitation with respect to generated artefacts, color balance and realism (measured via FID score). Furthermore, the authors introduced also an approach for generating the regions automatically to enhance the usability of the method. In the evaluation, the authors show improved realism via FID score, effectiveness via flip rate, similarity and sparsity in comparison to current state-of-the-art methods (e.g. ACE or LDCE). In addition, qualitative results are presented showing the applicability of the method for e.g. discovering complex patterns. The authors claim finally that they could improve on existing methods by a large margin.

**Strengths:**

- Well written text that is good to follow as a reader
- Despite there a different attempt to condition the generation process, the idea is novel enough when including the optimization steps of the method bringing XAI methods
- The theoretic description of the method is well formulated with respect to quality and clarity
- The structure of the evaluation has a high quality

**Weaknesses:**

- Only tested in a limited dataset, the authors should consider applying the method also on different datasets or domains
- From the results in the main paper, it could be confirmed that the method works on different generative models and external classifiers but claimed that results are available in the appendix. Therefore, I encourage the authors to include at least the numeric results in the main paper, so that a reader can obtain insights on the performance for different domains and models.
- The use of FID score is misleading the comparison to other methods, that do not limit the generation process by regions, since the limitation to certain spatial regions automatically reduced the FID score significantly. The authors should at least consider this limitation and think of better comparison, e.g. maybe a region-based FID score.
- In the paper the authors state that the method helps gaining more clear visual explanations where even a user can interact with the method to optimize the results. This is true with respect to how the method works, but no study with real users was performed to evaluate if such statement holds true. Either this claim should be removed or validated by a (small) user study.
- The other SOTA methods, e.g. ACE and DVCE are not really introduced, nor cited and not compared from a theoretical point of view. At least they must be cited and introduced.

**Questions:**

Apart from the recommendations in the weaknesses section I have the following questions and suggestions:
- Highlight the possible use cases of the method (e.g. class confusion) more so that the value of the method can be better understood
- Avoid using emphasizing phrases, e.g. by a large margin, and not bringing a quantitative results or at least a reference
- To many details and efforts with respect to the evalution are in the appendix weakening the main paper. I would suggest to include more details in the main paper from the additional evaluation results on different generative models and classifiers and reducing some of the visuals in the main paper + some very detailed analysis of the shown images and come up with a more condensed evaluation but including all the insights.
- What are the limitations of the method?
- It was not clear to me If there are not many possible image modifications for one sample yielding into a class change. How is that restricted to find the “best” modification and what are the optimal criteria?

---

> ### Author Response · Authors · 2024-11-24
> **Response I, part 1/2**
>
> We sincerely thank the reviewer for the feedback and appreciation of the mentioned aspects of our paper. We address the reviewer's concerns below.
>
> **Weaknesses**
>
> **W1. Only tested in a limited dataset (...).** While ImageNet is considered as a very challenging benchmark for VCE generation ([Augustin et al. (2022)](https://arxiv.org/abs/2210.11841), [Augustin et al. (2024)](https://arxiv.org/abs/2311.17833)), we agree that the evaluation could be extended. To address that, **we include three additional datasets** to ensure a fairer comparison. Specifically, following recent works ([Jeanneret et al. (2022)](https://arxiv.org/abs/2203.15636), [Jeanneret et al. (2023)](https://openaccess.thecvf.com/content/CVPR2023/papers/Jeanneret_Adversarial_Counterfactual_Visual_Explanations_CVPR_2023_paper.pdf), [Weng et al. (2024)](https://link.springer.com/chapter/10.1007/978-3-031-73016-0_20)), we evaluate RCSB on CelebA and CelebA-HQ, and provide an additional experiment on MNIST to showcase that our method works across the entire spectrum of resolutions and data complexities. These experiments include training I2SB for the inpainting task on each dataset prior to evaluation. Moreover, we adapt the implementation of DiME ([Jeanneret et al. (2022)](https://arxiv.org/abs/2203.15636), [Jeanneret et al. (2023)](https://openaccess.thecvf.com/content/CVPR2023/papers/Jeanneret_Adversarial_Counterfactual_Visual_Explanations_CVPR_2023_paper.pdf), [Weng et al. (2024)](https://link.springer.com/chapter/10.1007/978-3-031-73016-0_20)) to ImageNet and FastDiME ([Weng et al. (2024)](https://link.springer.com/chapter/10.1007/978-3-031-73016-0_20)) to both CelebA-HQ and ImageNet, to provide a broader comparison. For details of these extended experiments, please see the newly added section **Other benchmarks** in the Appendix. Overall, our method sets new SOTA in either 5 or 6 (depending on the case) out of 8 considered metrics. We also provide qualitative examples presenting the method's ability to create realistic RVCEs on face images. Regarding the MNIST dataset, we utilize it as additional proof that our method is able to handle shape modifications in addition to textural and color-based changes. For these results, we refer to the **Shape modification** section in the Appendix. We hope that this extension of our evaluation protocol effectively addresses the reviewer's concerns.
>
> **W2. From the results in the main paper, (...).** We fully agree that including the mentioned numeric results in the main paper would benefit the reader. However, including additional tables in the main manuscript is not currently possible due to space constraints. Therefore, we include the additional results from **W1.** in the new sections mentioned above. Moreover, to support future research, we will shortly update the Appendix with quantitative results for all of the classifiers mentioned at the end of the experimental section. [Update: the results are now included in the Appendix].
>
> **W3. The use of FID score is misleading (...).** We agree with the reviewer's concern that using FID as evaluation measure in this context might be misleading. We use it mainly to ensure a fair comparison with previous works, e.g. [Jeanneret et al. (2023)](https://openaccess.thecvf.com/content/CVPR2023/papers/Jeanneret_Adversarial_Counterfactual_Visual_Explanations_CVPR_2023_paper.pdf), which also identified this issue as FID may be biased by the unmodified original pixels. To alleviate it, also following prior works, our quantitative evaluation uses the sFID metric, which divides the original and generated data into independent splits. Next, FID is computed on splits where the synthetic images are independent from original samples, and averaged. This ensures that the evaluation is not biased by the unmodified original pixels. For a more detailed description of each metric, we refer all readers to subsection **Metrics description** in the Appendix.
>
> **W4. In the paper the authors state that the method (...).** We agree that this claim lacked proper support in the initial version of the paper. To address that, we perform an independent user study focused on a. understanding which type of the explanation the users prefer (standard VCE vs. our RVCE) and b. verifying whether the users treat the possible interaction with the method as a useful and helpful addition. We briefly describe the results below. For more details, see the newly added **User studies** section in the Appendix, where an additional user study focused on improving the understanding of model's failure cases is also mentioned.

---

> > ### Author Response · Authors · 2024-11-24
> > **Response I, part 2/2**
> >
> > The study involved 15 participants with MSc-level knowledge in machine learning, who were not aware of the research conducted in our paper. They compared VCEs and RVCEs for the same factual images, along with the absolute differences from the originals. Participants were asked which explanation type they found more helpful in understanding the model’s decision-making. 86.6% preferred RVCEs, with common reasons including that the semantic changes were more localized, easier to interpret, and better aligned with human intuition.
> >
> > The second part of the study assessed the benefits of interacting with the explanation creation process by specifying regions manually. Initially, participants were presented with standard VCEs, where only the original image and its VCE were shown, with no possible interaction. Next, participants observed the process of manual region specification to generate RVCEs and were given the opportunity to specify the regions themselves. Afterward, they were asked if the interactive process was more helpful for understanding the model’s reasoning than the standard static approach. 93.3% of participants supported the interactive process, citing the ability to validate regions that align with human understanding and incorporate input from domain experts for more in-depth analysis of the model.
> >
> > Overall, the participants' feedback highlighted that RVCEs provided clearer, more intuitive insights into the model's decisions and failure points, further supporting their usefulness in practical scenarios.
> >
> > **W5. The other SOTA methods (...).** Due to space constraints, we were only able to briefly introduce and cite the previous SOTA methods in **Background & Related Work** section of the main paper. To improve on that, we extend the Appendix with an additional section **Comparison to previous works** that describes prior approaches (including ACE and DVCE) from a theoretical point of view and compares them to our method.
> >
> > **Questions**
> >
> > Thank you for the provided suggestions. With regards to possible use cases, we note that besides standard applications of VCEs ([Samek et al. (2019)](https://link.springer.com/book/10.1007/978-3-030-28954-6)), our approach allows to include human interaction in the explanation creation process, offering important applications in domains like medical imaging, where the radiologist points to regions of lesions. Moreover, we also want to highlight the second user study conducted to evaluate how our method may help in understanding the misclassifications made by the model (details in the Appendix and the response to reviewer Rswz). In terms of our method's limitations, this aspect was also addressed by other reviewers, and we extend the Appendix with a section on this topic (**Limitations**).
> >
> > Regarding the last point, it is indeed possible for our method to provide many explanations for a single image and region pair due to stochasticity induced by the initial Gaussian noise inserted into the masked area. However, in terms of optimality, the evaluation criteria for VCEs stand as an open and active research question. This is mainly due to the definition of a VCE requiring that the introduced change is minimal in a *semantically meaningful* way. Since semanticity is a largely subjective measure, there are no direct approaches to evaluate it. One typically resorts to proxy measures of various aspects of such explanations, e.g., FID for realism, COUT for sparsity with regards to the classifier, LPIPS for semantic similarity. One of the novelties introduced by our work is that the conditioning signal comes only from the classifier of interest and does not require using any proxy measures for keeping similarity to the original image. Despite that, our method finds the "best" modification in terms of the proxy metrics used to evaluate VCEs, which are included in Tables 1. and 2., after the initial hyperparameter grid search on a small subset of images. In practice, during evaluation, we limit the generation to provide only a single RVCE for each original image, hence alleviating the issue of choosing a better one.

---

### Official Review · Reviewer_Rswz · 2024-10-29

**Soundness:** 3
**Presentation:** 3
**Contribution:** 2
**Rating:** 5
**Confidence:** 4

**Summary:**

This paper presents a method for creating visual counterfactual explanations for classification tasks such as ImageNet classification. These explanations illustrate how an image could be modified to change the classification output. The approach is based on a combination of existing methods which select the most relevant parts of the image to change the decision and then mask and inpaint these regions to change the classification decision. A major contribution of the proposed method is limiting the masking/inpainting to a small connected spatial region in the image rather than allowing changes everywhere. The method outperforms other approaches on a set of metrics which aim to balance various factors desirable for visual counterfactuals (namely, changing the model’s prediction while keeping the output image similar to the input, keeping changes sparse, and ensuring the output is realistic-looking).

**Strengths:**

- The proposed method outperforms similar approaches on the proposed metrics, at least on a very small evaluation dataset (3 class pairs).
- The writing is very clear, and the results are very well organised and presented, with well-designed figures to illustrate the model's performance.
- The paper shows some interesting possible applications, such as allowing users to target specific regions of an image to see those regions would need to change to change the model's decision.

**Weaknesses:**

- The main contribution of the paper (limiting the inpainting to a specific region) also seems like a weakness, since the model cannot capture cases where the differences between two classes are changes in shape. (For example, this is shown the American chameleon <-> common iguana examples -- the main difference between the two types of lizard is the overall shape/size, but the model can only show how the texture/color differs between these classes.)
- The intro says that the goal of generating counterfactuals is to explain the decision-making process of an image classifier to a user, but it's not clear whether the proposed model is better than any other model at achieving that result. This approach performs best on the metrics, but simply maximizing performance across metrics seems like it would result in a "counterfactual" image that flips the classifier's decision with almost no visible changes to the image, which wouldn't be a useful counterfactual for human users (as they wouldn't be able to tell what changed).

**Questions:**

- The comparison models in the main experiment (Table 1) are not defined anywhere in the paper -- the abbreviations used in the table are never used in the text, and the table has no citations to tell you what the abbreviations might mean. I think I was able to work out what these models are through Googling but they should be named in the paper (though it's worth noting that there is another XAI method called "ACE").
- Since the evaluation dataset seems to be restricted to class pairs that favor the proposed method (cases where the two classes have the same shape but different surface color/texture), I'd be curious to know how the proposed model compares on a wider set of counterfactual pairs. Does the proposed method outperform ACE, etc. on arbitrary counterfactual pairings, where the key differences between the classes might include shape changes?
- Can this method be helpful for users to understand adversarial attacks (if the input image is a adversarial image, can the counterfactuals help explain why the model saw the image as the wrong category)?

---

> ### Author Response · Authors · 2024-11-24
> **Response I, part 1/2**
>
> We thank the reviewer for the provided feedback and appreciation of the mentioned aspects of our paper. Regarding the reviewer's concerns, we detail our clarifications below.
>
> **Strengths**
>
> **S1. (...) at least on a very small evaluation dataset (3 class pairs).** Since a common part of the reviews was the aspect of an insufficiently comprehensive benchmark, we implemented our approach for additional three datasets: CelebA and CelebA-HQ to provide a fair comparison with all of the previous diffusion-based methods, and MNIST as a sanity check application, with which we also wanted partially to address the reviewer's concern about our method not being able to modify shapes. Specific details of these experiments are included in **Other benchmarks** and **Shape modification** in the Appendix. Regarding the MNIST experiment, we hope that the qualitative samples provide some proof that our method is able to modify shapes in addition to textures. We further address this aspect below.
>
> **Weaknesses**
>
> **W1. The main contribution of the paper (...).** While we fully acknowledge the reviewer's point, we must argue that the classification model does not have to rely on the described changes in overall shape/size, since it may be biased by how these classes (*American chameleon* $\leftrightarrow$ *Common iguana*) are represented within the training data. Following the definition of VCEs, one aims at obtaining a *minimal semantic* change that changes the model's decision, and the change in texture/color seems to simply be enough for the model to change the prediction.
>
> We must also mention that both the region area and the trajectory truncation feature allow for controlling how the original content is preserved. Hence, setting the area to contain almost only the object of interest and picking low trajectory truncation will typically result in textural changes only. To address this concern, we provide new qualitative samples in the **Shape modification** section in the Appendix, which show that these parameters can be properly modified to result in changing the overall shape and size. Moreover, the samples are supported with a quantitative comparison to previous SOTA on three new tasks, which should mostly depend on modifying the object's shape instead of texture and/or color: pretzel $\rightarrow$ bagel, hatchet $\rightarrow$ hammer and paperknife $\rightarrow$ wooden spoon. We repeat the results from the mentioned section in the table below. Clearly, the performance of our method and its advantage over previous SOTA is largely preserved when compared with Table 1. from our paper. We hope that these extended results, together with qualitative examples, provide sufficient proof of our method's ability to modify entire shapes of objects.
>
> **Pretzel → Bagel**
>
> | Method       | FID  | sFID | S³   | COUT | FR   |
> |--------------|------|------|------|------|------|
> | DVCE         | 34.3 | 43.9 | 0.59 | 0.37 | 77.4 |
> | RCSB $^A$ (ours) | 11.4 | 22.9 | 0.86 | 0.84 | 97.2 |
>
> **Hatchet → Hammer**
>
> | Method       | FID  | sFID | S³   | COUT | FR   |
> |--------------|------|------|------|------|------|
> | DVCE         | 31.2 | 39.8 | 0.66 | 0.43 | 92.8 |
> | RCSB $^A$ (ours) | 9.8  | 15.4 | 0.91 | 0.89 | 97.8 |
>
> **Paperknife → Wooden Spoon**
>
> | Method       | FID  | sFID | S³   | COUT | FR   |
> |--------------|------|------|------|------|------|
> | DVCE         | 29.1 | 35.4 | 0.69 | 0.41 | 88.2 |
> | RCSB $^A$ (ours) | 9.9  | 18.2 | 0.86 | 0.88 | 98.9 |
>
> **W2. The intro says that the goal of (...).** We agree with the reviewer's point that the aspect of explaining the decision-making process of the classifier to the user was not backed in the original version. To address that, we perform two independent user studies, providing a comprehensive analysis of how our RVCEs are more helpful in understanding the model's reasoning, how the interactive explanation creation serves as added value to the explanatory process and whether our explanations help in understanding the misclassifications of the model.
>
> The first study assessed the effectiveness of RVCEs compared to standard VCEs and the advantages of incorporating an interactive process. The results indicated that 86.6% of participants considered RVCEs more effective than VCEs for understanding and explaining the model's decisions. Key reasons included the ability to make localized semantic changes, enhanced interpretability, and stronger alignment with human intuition. Furthermore, 93.3% expressed a preference for our method over standard static interaction with VCEs when given the option to manually specify regions for RVCEs. This preference was largely driven by the ability to validate regions consistent with human understanding and incorporate input from domain experts.

---

> > ### Author Response · Authors · 2024-11-24
> > **Response I, part 2/2**
> >
> > In the second study, RVCEs were utilized to help another group analyze model failure cases and pinpoint the minimal semantic changes needed for correction. We conducted five iterations of the same type of experiment, which began with showing the participants the wrong initial prediction of the model on a given image, the correct class and the simple characteristics of the predicted and true class. Then, the participants were presented with the region constraint given by our method together with a series of RVCEs showing the semantic change that flips the model's decision to the correct one. In this study, over 80% of participants found RVCEs to be an effective tool for addressing the issue of understanding why the model `saw` the image as the wrong category. Additionally, 90.9% were able to identify the missing semantic features necessary for accurate predictions and reported an improved understanding of the failure cases after examining RVCEs. Every participant agreed that RVCEs were a valuable resource for explaining the model's decisions. For more details, please refer to the newly added *User studies* section in the Appendix.
> >
> > **Questions**
> >
> > **Q1.** We thank the reviewer for pointing out that the reader might get lost in the tables based on the method's abbreviations only. To fix that, we provide them in the **Background & Related work** section. Moreover, since this was also mentioned by another reviewer, we provide a broader description of each of the competing methods and how they relate to our approach in **Comparison to previous works** the Appendix .
> >
> > **Q2.** We address this question above in the comment related to **W2.**.
> >
> > **Q3.** We address this point with an independent user study also mentioned in **W2.**.
> >
> > We hope that the provided additional results consitute sufficient empirical proof that our method is not limited to textural/color changes, that it provides added value to the users and that it achieves new SOTA across all typically considered benchmarks. We are open to engage in further discussion if any issues remain and encourage the reviewer to consider increasing our score based on the additional evidence of the method's effectiveness.

---

> > > ### Author Response · Authors · 2024-12-01
> > >
> > > Dear reviewer,
> > > as the discussion period comes to an end, we would greatly appreciate the feedback on our responses to your comments. We are open to further discussion if any issues remain unclear or additional concerns arise. We would also like to highlight a large series of experiments conducted during the rebuttal to address the comments of other reviewers, which provide additional supporting evidence of the capabilities of our approach.

---

> > > > ### Author Response · Authors · 2024-12-01
> > > >
> > > > Dear reviewer,
> > > > following up on our previous comment, we would be extremely grateful if you could refer to whether the responses provided above adequately address your concerns and possibly engage further in the discussion.

---

### Official Review · Reviewer_p98R · 2024-11-03

**Soundness:** 3
**Presentation:** 3
**Contribution:** 2
**Rating:** 6
**Confidence:** 4

**Summary:**

This paper targets at the task of VCEs and defines more refined task with region constraints to gain more precise explanations. Further, it also proposes the method RCSB for the newly defined task, which is built upon the I^2SB and solves the problem as a conditional inpainting task. The idea is simple and the experiments demonstrates its effectiveness. Nevertheless, the technical novelty is relatively limited.

**Strengths:**

1. The newly formulated RVCEs is reasonable.
2. The presentation is good and the paper is easy to follow.
3. The experiments demonstrates the effectiveness of the proposed method.

**Weaknesses:**

1. The technical novelty is relatively limited. It formulates the task as a conditional inpainting problem and solves it by applying $I^2SB$.

**Questions:**

1. The paper claimed that the definition of VCEs has potential confirmation bias. However, it is not deeply investigated. Besides, the teaser example in Figure 1 also cannot indicate such confirmation bias.
2. There is no direct qualitative comparison between different methods, in Figure 5 and 6.

---

> ### Author Response · Authors · 2024-11-24
> **Response I, part 1/2**
>
> We thank the reviewer for the insightful feedback and appreciating our paper's clarity. Below, we refer to the mentioned concerns.
>
> **Weaknesses**
>
> **W1. The technical novelty is relatively limited. (...).** To elaborate on the aspect of limited novelty, we must point to specific contributions of our work. First, RCSB is the first method to show that a more general class of generative models than SGMs (namely tractable Schrödinger Bridges) can be adapted to the problem of counterfactual explanation generation. Second, contrary to all previous diffusion-based approaches to VCEs, our approach builds from the ground up the practical approach to effective utilization of the context available from the outside of the region. There, the novelties stem from treating the entire process as an optimization procedure, which can be improved by ADAM stabilization and adaptive normalization - two crucial factors in highly increasing our method's effectiveness. Third, the proposed general definition of RVCEs combined with high effectiveness of RCSB allows for obtaining explanations where the region of interest was chosen independently from the classifier, e.g., through automated segmentation or even human interaction, which was not addressed by any of the previous diffusion-based approaches. We believe that these contributions constitute great technical novelty.
>
> Moreover, the quantitative results highlight that our approach outperforms previous SOTA by a large margin. For example, we achieve up to 4 times better FID, up to 3 times better sFID, up to 2 times higher COUT and mach or exceed the best previous results in terms of representational similarity and efficiency. As part of the rebuttal, we extended our quantitative evaluation with two benchmarks, CelebA and CelebA-HQ, used by many previous diffusion-based approaches. There, we also provide new SOTA records, outperforming the previous one in 5 or 6 (depending on the case) out of 8 metrics. We believe that these results highlight the novelty brought by the effectiveness of our method.
>
> **Questions**
>
> **Q1. The paper claimed that the (...).** We agree that the aspect of confirmation bias could be more thoroughly analyzed in the paper. While we treat it rather as a starting point for motivating our method, here we describe it more deeply. Beginning with Fig. 1, the regular VCE may bias the user to think that the model's decision changed due to the most visible and the most intuitive changes of the bird, i.e. the modified feathers and head. However, since the VCE also modifies the branch, the background and the copyright caption, there is a risk that these changes influence the decision change more strongly, but are not recognized by users that do not investigate the explanation carefully. On the other hand, RVCEs alleviate this problem by not allowing modifications anywhere except a predefined region.
>
> As part of the rebuttal process, we conduct two independent user studies, which we believe to deeply investigate many practical aspects of RVCEs, in particular the notion of confirmation bias and its presence in VCEs. The first study evaluated the usefulness of RVCEs compared to standard VCEs and the benefits of the interactive process. Results showed that 86.6% of participants found RVCEs more helpful than VCEs for understanding and explaining the model’s decisions. Participants cited reasons such as more localized semantic changes, improved interpretability, and better alignment with human intuition. Additionally, 93.3% preferred our method when comparing standard static interaction with VCEs to manually specifying regions for RVCEs. This preference was primarily attributed to the ability to validate regions that align with human understanding and to involve domain expertise.
>
> In the second study, RVCEs were used to assist a different group in analyzing model failure cases and identifying the minimal semantic changes required for correction. Over 80% of participants found RVCEs effective for addressing the problem. Furthermore, 90.9% successfully identified the missing semantic features needed for the model to make accurate predictions and reported a better understanding of failure cases after reviewing RVCEs. All participants agreed that RVCEs were a valuable tool for explaining the model's decisions. Additional details can be found in the newly added **User studies** section of the Appendix.

---

> > ### Author Response · Authors · 2024-11-24
> > **Response I, part 2/2**
> >
> > **Q2. There is no direct qualitative (...).** We thank the reviewer for pointing out the absence of a qualitative comparison. To address that, we extend the Appendix with qualitative samples from other methods for example images from Fig. 5 of our paper. Notably, these results highlight the effectiveness of our approach in preserving the original image content while introducing semantic changes that effectively flip the model's decision, which clearly stand out when compared with other approaches in terms of minimality and realism.
> >
> > In addition to the above, we would like to highlight the extent of our additional contributions throughout the rebuttal and discussion phase. Namely, the introduction of two independent user studies, three additional benchmarks with new SOTA records (for CelebA and CelebA-HQ, with MNIST as an additional proof-of-concept example) and additional experiments verifying various aspects of our approach, e.g., the ability to modify entire shapes, helpfulness for the users in clearly understanding model's misclassifications and many more.

---

> ### Comment · Reviewer_p98R · 2024-11-29
>
> Thanks for the detailed response. After comprehensive consideration of both the reply and the reviews by other reviewers, I maintain my rating score.

---

> > ### Author Response · Authors · 2024-12-01
> >
> > We would like to thank the reviewer for actively engaging in the discussion period. We are open to address any additional concerns or questions regarding our approach if needed.

---

### Official Review · Reviewer_PMZS · 2024-11-03

**Soundness:** 3
**Presentation:** 3
**Contribution:** 2
**Rating:** 5
**Confidence:** 3

**Summary:**

This manuscript studies the problem of visual counterfactual explanations (VCEs) and argues that region is important. It is an extension of the visual counterfactual explanation by adding a hard region constraint. The method is based on a recently developed diffusion-based image-to-image translation method, named Image-to-Image Schrodinger Bridge ($I^2$SB). The authors discover a good way to inject the gradient from the classifier into the generation process of $I^2$SB and make the $I^2$SB generate more plausible images that change the prediction of the classifier. It gives us a more flexible way to understand the model prediction process.

**Strengths:**

- Bring the region constraint to the counterfactual example generation somehow makes sense to better align with human perception.

- The authors provide a way to better inject the guidance from the gradient of the classifier f into the generation process of the inpainting model. From the visual examples, we can indeed observe a much higher quality.

**Weaknesses:**

- As the regions are the core input of the proposed method, how to define/find the regions is crucial. The method uses either the attribution map generation or the grounding method. Will these methods introduce additional bias in the understanding of the classifier?

- From my perspective, the proposed method is more like just an image painting with specific guidance, where the guidance comes from the gradients of the classification model. The quality of the explanation ability somehow entangles the ability of the classifier itself and the inpainting model. The quality (FID, etc,) of the generated samples is also highly entangled with the base inpainting models. This might be a problem if we want to understand the behaviors of the classifier.

- The visual counterfactual explanation is similar to the problem of attribution map generation, where the goal is to detect the regions that have the most influence on the prediction of the model either positively or negatively. Actually, one of the region generation methods is attribution map generation (IG). If we use the attribution map generation method, we can also generate some examples with high FID while with different predicted labels. I wondered what's the core difference between these two setups.

**Questions:**

- I'm wondering what will be generated if we force the label of the generated image to a very different class. For example, if we want to change the output of the model given the image of an animal to the class of a dining table, what does the generated image look like?

- There are some cases in which the model indeed makes use of unusual visual cues to predict the label of the image, like the background. We may also want to know that the model has learned such a prior from the dataset (for example, the model links the snow background to the wolf due to dataset bias). How can the proposed method discover such cases?

- The visualization shown in Figure 5 is interesting. At the same time, I noticed that after changes, several images changed the image's attributes significantly toward the target class (like the Flamingo and Cauliflower cases). I'm wondering what can we learn from these changes to understand the model's behaviors, as the model's prediction on these changed samples is very reasonable and as expected.

---

> ### Author Response · Authors · 2024-11-16
> **Response I, part 1/2**
>
> We are grateful to the reviewer for the insightful feedback. We discuss the mentioned topics below.
> - [W1.] We agree that the choice of region extraction method is important and using any automated approach may introduce additional bias to the explanation, e.g., some attribution methods focusing on specific patterns of the image or LangSAM's understanding of particular objects. Sometimes, some form of bias may be regarded as beneficial to our approach. For example, since many attribution methods are very faithful to the model, using highest attributions in the proposed way will focus on semantic features that are most important to the prediction. Moreover, because our approach allows the user to provide the region manually, one can explicitly test any desired bias or get rid of the bias induced by an automated approach.
> - [W2.] We also agree that combining generative models with guidance from the model of interest introduces inevitable entanglement. For example, one has to first ensure that the generative model is good enough to properly modify images. However, most of the previous works follow the specific path on basing their method on such entanglement (Augustin et al. (2022), Augustin et al. (2024), Jeanneret et al. (2022), Jeanneret et al. (2023), Jeanneret et al. (2024)), Weng et al. (2024)). Following this research direction, it seems that producing high-quality VCEs requires access to an accurate generative prior, which accurately approximates the original data distribution. This ensures that the signal from the classifier is used to produce a sample that resembles original data. We also agree that this combination might lead to conflicting modifications in the image. On the other hand, our approach is, to the best of our knowledge, the first diffusion-based method to show that the guidance signal may originate *only* from the classifier of interest, without the use of other regularization components, such as an $l_2$ loss or LPIPS (Zhang et al. (2018)), which might additionally bias the generation, complicating the explanatory process with classifier-unrelated features.
> - [W3.] Following and extending our previous answer, the main difference between the described approach and our method is the use of a generative prior that keeps the explanation close to the data manifold. This is required by the definition of a VCE, as we want to obtain a minimal *semantic* change and avoid any form of adversarial noise. The method described by the reviewer would indeed change the classifier's decision, but simultaneously be far from the original image in terms of semantics. Such an approach is much less informative to the users, since they cannot identify semantic features that are important to the model of interest.
>
> **References**
>
> Maximilian Augustin, Valentyn Boreiko, Francesco Croce, and Matthias Hein. Diffusion visual
> counterfactual explanations. In Advances in Neural Information Processing Systems, 2022.
> Maximilian Augustin, Yannic Neuhaus, and Matthias Hein. Analyzing and explaining image clas-
> sifiers via diffusion guidance. In Proceedings of the IEEE/CVF Conference on Computer Vision
> and Pattern Recognition, 2024.
>
> Guillaume Jeanneret, Lo¨ ıc Simon, and Fr´ ed´ eric Jurie. Diffusion models for counterfactual explana-
> tions. In Proceedings of the Asian Conference on Computer Vision, pp. 858–876, 2022.
>
> Guillaume Jeanneret, Lo¨ ıc Simon, and Fr´ ed´ eric Jurie. Adversarial counterfactual visual explana-
> tions. In Proceedings of the IEEE/CVF Conference on Computer Vision and Pattern Recognition,
> pp. 16425–16435, 2023.
>
> Guillaume Jeanneret, Lo¨ ıc Simon, and Fr´ ed´ eric Jurie. Text-to-image models for counterfactual
> explanations: a black-box approach. In Proceedings of the IEEE/CVF Winter Conference on
> Applications of Computer Vision, pp. 4757–4767, 2024.
>
> Nina Weng, Paraskevas Pegios, Aasa Feragen, Eike Petersen, and Siavash Bigdeli. Fast diffusion-
> based counterfactuals for shortcut removal and generation. In European Conference on Computer
> Vision, 2024.
>
> Richard Zhang, Phillip Isola, Alexei A Efros, Eli Shechtman, and Oliver Wang. The unreasonable
> effectiveness of deep features as a perceptual metric. In Proceedings of the IEEE conference on
> computer vision and pattern recognition, pp. 586–595, 2018.

---

> > ### Author Response · Authors · 2024-11-16
> > **Response I, part 2/2**
> >
> > - [Q1.] While our method does not in any way restrict the target label and the described approach can be easily verified, VCEs are typically evaluated on target classes that are semantically close to the model's prediction. This is mostly due to practical preferences. Most commonly, it is of greater interest to the receiver why the model predicted a given bird specie and not the other one, than a semantically unrelated class for which the feature changes are possibly much bigger and less interpretable. To address the curiosity, we updated the Appendix (**Unintuitive classes** subsection) with example RVCEs for unintuitive pairs of classes like, e.g., *missile* $\rightarrow$ *hamster*. As expected, the explanations depict unusual compositions of objects largely preserving the realism but lacking informativeness about the classifier.
> > - [Q2.] VCEs are indeed often used in the mentioned scenario of discovering unusual visual cues. As indicated, our method is capable of producing examples that represent such cases. Moreover, our approach extends this ability in two ways. First, the visual cues of interest can be manually picked by the user, enabling them to verify them on their own. Second, our approach ensures that some unusual visual cues are indeed the reason behind the prediction's change. For example, a standard VCE might modify both the standard and atypical feature, making their specific influence unclear. With our approach, one can choose regions of both of these features or only the specific one.
> > - [Q3.] The mentioned examples provide us with a better understanding of what the classifier actually judges to be a cauliflower or a flamingo. We agree that these changes are very reasonable and 'as expected', which can strengthen the trust of the user to the model. Certainly, the unusual visual cues are typically of much greater interest, since they identify a possible undesired behavior of the model.
> >
> > We hope that the above comments clarify the mentioned ambiguities from the manuscript. We are open to answering further questions for clarification. We also encourage the reviewer to possibly increase their score, since it seems that they judge our method as an effective approach to the considered problem.

---

> > > ### Author Response · Authors · 2024-12-01
> > >
> > > Dear reviewer,
> > > as the discussion period is coming to an end, we would greatly appreciate if you could provide feedback regarding the provided responses to your questions. Moreover, following the concerns of other reviewers during the rebuttal, we have conducted a number of additional experiments that extend our initial results with evidence supporting many claims and capabilities of our approach. Many of those also address your comments. Hence, we provide a brief summary to ensure that your concerns were properly addressed.
> > >
> > > **Weaknesses**
> > >
> > > **W1. As the regions (...). Will these methods introduce additional bias in the understanding of the classifier?**
> > >
> > > **and**
> > >
> > > **W2. From my perspective, (...). This might be a problem if we want to understand the behaviors of the classifier.**
> > >
> > > To better address how the proposed approach influences the user's understanding of the classifier's behavior and decision-making process, we conduct two independent user studies that deeply investigate the topic of their helpfulness to humans.
> > >
> > > The first study evaluated the effectiveness of RVCEs compared to standard VCEs and examined the benefits of incorporating an interactive process. Results showed that 86.6% of participants found RVCEs more effective than VCEs for understanding and explaining the model's decisions. This preference was attributed to the ability to make localized semantic changes, improved interpretability, and stronger alignment with human intuition. Additionally, 93.3% preferred our method over standard static VCE interactions when allowed to manually specify regions for RVCEs. This preference stemmed from the ability to validate regions aligned with human understanding and integrate input from domain experts.
> > >
> > > In the second study, RVCEs were used to assist another group in analyzing model failure cases and identifying minimal semantic changes required for correction. Five iterations of the experiment were conducted, starting with participants being shown the model’s incorrect initial prediction on a given image, the correct class, and the basic characteristics of the predicted and true classes. They were then provided with the region constraint generated by our method, along with a series of RVCEs illustrating the semantic changes needed to flip the model's decision to the correct class. Over 80% of participants found RVCEs effective for understanding why the model misclassified the image. Moreover, 90.9% successfully identified the missing semantic features necessary for accurate predictions and reported an improved understanding of failure cases after using RVCEs. All participants agreed that RVCEs were a valuable resource for explaining the model's decisions.
> > >
> > > For more details regarding the user studies, we refer to a new section **User studies** in the Appendix.
> > >
> > > Overall, we conclude that the first study effectively shows that RVCEs obtained with our automated approach are preferred by the users over standard VCEs. Moreover, even if they introduce some form of additional bias, the second part of the study highlights that the users regard the possibility to provide regions manually as highly valuable, precisely for the reasons connected to bias elimination. In terms of the second study, it shows that combining I2SB with the classifier of interest allows to better understand the behavior of the latter, mainly due to the ease of precisely identifying semantic features that are required to change in order to fix the initial misclassification.
> > >
> > > **Questions**
> > >
> > > **Q2. There are some cases (...). How can the proposed method discover such cases?**
> > >
> > > Regarding the method's ability to discover unusual visual cues, we empirically verify that, besides the possibility to manually indicate unconventional regions or segment them using a foundation text-to-segmentation model shown in the manuscript, our approach also proves to be effective using a fully automated approach. Specifically, we show that the method's capabilities are largely preserved even if the highest initial attributions are zeroed out if they exceed the value of some quantile $q$ before being used for region extraction. Below, we include the results of our method on the three main tasks considered in the paper obtained with the $\text{RCSB}^A$ configuration ($10\%$ of total area coverage) for $q\in\{ 0.3, 0.4, 0.5, 0.6, 0.7, 0.8, 0.9\}$. Hence, with decreasing $q$, the utilized region is based on features that can be considered decreasingly important to the classifier. This way, more *obvious* features can be ommitted, and the region is able to focus on unusual parts of the image, possibly leading to discovery of unconventional visual cues. Crucially, using attributions of smaller magnitude does not visibly impact our method's performance and efficiency, leaving plenty of room for exploration.

---

> > > > ### Author Response · Authors · 2024-12-01
> > > >
> > > > **Zebra -- Sorrel**
> > > >
> > > > | $q$  | FID  | sFID  | S³   | COUT  | FR    |
> > > > |------|------|-------|------|-------|-------|
> > > > | 0.9  | 5.2  | 13.8  | 0.87 | 0.62  | 91.6  |
> > > > | 0.8  | 4.6  | 13.3  | 0.87 | 0.53  | 88.0  |
> > > > | 0.7  | 3.9  | 12.7  | 0.88 | 0.48  | 85.7  |
> > > > | 0.6  | 3.6  | 12.4  | 0.89 | 0.45  | 82.4  |
> > > > | 0.5  | 3.6  | 12.4  | 0.89 | 0.40  | 78.5  |
> > > > | 0.4  | 3.7  | 12.3  | 0.89 | 0.38  | 77.7  |
> > > > | 0.3  | 3.9  | 12.6  | 0.89 | 0.40  | 79.2  |
> > > >
> > > > **Cheetah -- Cougar**
> > > >
> > > > | $q$  | FID  | sFID  | S³   | COUT  | FR    |
> > > > |------|------|-------|------|-------|-------|
> > > > | 0.9  | 6.1  | 17.1  | 0.92 | 0.88  | 99.8  |
> > > > | 0.8  | 4.4  | 15.7  | 0.91 | 0.80  | 99.1  |
> > > > | 0.7  | 3.5  | 15.1  | 0.91 | 0.74  | 97.4  |
> > > > | 0.6  | 2.9  | 14.6  | 0.92 | 0.68  | 94.3  |
> > > > | 0.5  | 2.7  | 14.5  | 0.92 | 0.65  | 93.1  |
> > > > | 0.4  | 2.8  | 14.6  | 0.92 | 0.65  | 93.4  |
> > > > | 0.3  | 3.1  | 14.9  | 0.92 | 0.69  | 95.7  |
> > > >
> > > > **Egyptian cat -- Persian cat**
> > > >
> > > > | $q$  | FID   | sFID  | S³   | COUT  | FR    |
> > > > |------|-------|-------|------|-------|-------|
> > > > | 0.9  | 13.5  | 31.4  | 0.86 | 0.89  | 99.9  |
> > > > | 0.8  | 11.0  | 28.6  | 0.85 | 0.84  | 99.3  |
> > > > | 0.7  | 9.2   | 26.9  | 0.85 | 0.79  | 98.5  |
> > > > | 0.6  | 7.8   | 25.5  | 0.85 | 0.73  | 96.1  |
> > > > | 0.5  | 6.9   | 24.7  | 0.86 | 0.69  | 94.6  |
> > > > | 0.4  | 6.6   | 24.5  | 0.87 | 0.65  | 93.1  |
> > > > | 0.3  | 7.5   | 25.3  | 0.86 | 0.67  | 94.2  |
> > > >
> > > > We believe that, in addition to unusual visual cues discovered by our method for images shown in, e.g., Figures 6. and 7. of our paper, these results further strengthen the reasons for which one could explore the RVCEs obtained with our approach to detect more unconventional features used by the classifcation model.
> > > >
> > > > **Q3. The visualization shown (...). I'm wondering what can we learn (...).**
> > > >
> > > > Following the conclusions from the second user study, even `reasonable` and `as expected` changes may prove very useful when trying to better understand the model's way of reasoning. Specifically, the study shows that such changes allow the user to find out about the minimal semantic change needed for the model to transition from misclassifying the image to predicting it correctly. This way, the user may feel better informed about the specifics of model's decision-making process.
> > > >
> > > > We hope that the above additional comments further address the reviewer's concerns and we look forward to discussing any other issues that need clarification.

---

> > > > > ### Author Response · Authors · 2024-12-01
> > > > >
> > > > > Dear reviewer,
> > > > > following up on our previous comment, we would be extremely grateful if you could refer to whether the responses provided above adequately address your concerns and possibly engage further in the discussion.

---

> ### Comment · Reviewer_PMZS · 2024-12-02
> **Thanks the authors for the clear response**
>
> I would gratefully thank the authors for the rebuttal in the short period, its indeed a hard work. Part of my concerns has been resolved especially the difference between the attribution map generation and the VCE. The newly added user study is good to show the alignment between human's understanding and the generated mask. However, I still hold one concern regarding to the root of the method. From the user study, we can conclude that the method can help person to find the most influence region aligned with human's perception. It is doubtful if the method can explain the wrong behavior of the model especially when the model's prediction is influence by some factors similar to adversarial attach. But this might be a problem beyond this manuscript. I would like to raise my score.

---

> > ### Author Response · Authors · 2024-12-02
> >
> > We sincerely thank the reviewer for appreciating our efforts and raising the score. Regarding the mentioned concern, we would like to highlight the results of the second conducted user study below.
> >
> > The study focuses explicitly on the scenario mentioned by the reviewer, i.e., it verifies whether the method `can explain the wrong behavior of the model`. Specifically, participants are presented with images which were initially misclassified by the model, together with the name of the predicted class, the ground truth class and a series of additional images that allow them to identify key semantic features which characterize both of these classes (for examples shown to the participants, see the **User studies** section in the Appendix). Then, the participants are presented with a series of RVCEs obtained with our method using the automated region extraction approach which *correct* the model's prediction, i.e., it predicts each RVCE as the ground truth class for the initial image. According to the participants themselves, in over 80% of the cases, this procedure allowed them to identify the semantic features that the model lacked in its initial prediction and that lead to correcting the decision once they appeared on the image. Moreover, 90.9% of participants stated that the explanations helped them in better understanding the initial misclassification. We argue that these empirical results indicate our method's ability to explain the wrong behavior of the model.
> >
> > We also agree with the reviewer that explaining the wrong behavior of the model when influenced by factors similar to adversarial attacks might be very challenging. However, we must also argue that this difficulty is more broadly connected to the very definition and concept of visual counterfactual explanations, and not our specific method per se. Since these explanations aim at modifying the image in a semantically meaningful way, i.e., aligned with the data distribution, they will be inherently limited in cases when the cause of the misclassification is adversarial and not semantic. For example, if the adversarial change required for misclassification is based on a single pixel, then changing this pixel to its original value is enough to correct the model, but such change can hardly be recognized as semantically meaningful. Hence, it seems unfair to assign this particular limitation exclusively to our method.
> >
> > The mentioned concern also connects with the challenge of evaluating counterfactual explanations, since typically there is no `ground truth` to compare with. As the reviewer mentions, this is a problem beyond our manuscript, and we recognize it as an important future research direction.
> >
> > To summarize, we are grateful for appreciating our efforts and encourage the reviewer to consider our paper independently from the inherent limitations of visual counterfactual explanations, which certainly exist and require more research. Our approach addresses many aspects which, following the quantitative and human evaluation, were problematic for previous approaches, but have been very effectively addressed by what our paper introduces. This is supported by a large array of experiments on a very diverse pool of datasets and edge cases, where our method was shown to outperform all of the previous ones.

---

### Official Review · Reviewer_2FX9 · 2024-11-04

**Soundness:** 4
**Presentation:** 4
**Contribution:** 3
**Rating:** 8
**Confidence:** 4

**Summary:**

The authors propose a method for generating region-constrained visual counterfactual explanations, arguing that the region constraint leads to improved interpretability of the resulting counterfactual explanations (compared to unconstrained CEs). For this purpose, they leverage i) Image-to-Image Schrödinger Bridges, ii) automatically determined regions of interest, using either classical visual attribution methods or a text-to-object segmentation foundation model (LangSAM), and iii) several tweaks for stabilizing classifier guidance of the diffusion process. The proposed method outperforms several baselines on previously identified particularly challenging counterfactual questions on ImageNet.

**Strengths:**

- The paper is very well-written overall and generally a pleasure to read; the various technical aspects of the method are very well-motivated, rigorously and clearly presented, and demonstrated to be useful in ablation experiments.
- Region-constrained counterfactual explanations are certainly very useful (although I would argue that this approach is not completely novel, cf. my comments below) and I believe the explicit problem formulation and framing here can help move this field forward. The extensive and clearly explained examples are also much appreciated.
- The I2SB-based conditional inpainting approach appears to perform very well.
- An interesting approach for (ADAM-derived) gradient stabilization and normalization in classifier-guided diffusion

**Weaknesses:**

**The relationship to several prior works should be discussed more comprehensively**, and possibly some of these methods should be included as baselines. The explicitly novel contributions should be highlighted more clearly.
- Both [Jeanneret et al. (2023)](https://openaccess.thecvf.com/content/CVPR2023/papers/Jeanneret_Adversarial_Counterfactual_Visual_Explanations_CVPR_2023_paper.pdf) and [Weng et al. (2024)](https://link.springer.com/chapter/10.1007/978-3-031-73016-0_20) also use automatically determined region constraints and conditional inpainting to obtain region-constrained counterfactual explanations. In several places it reads as if the authors claim novelty on region-constrained CEs *per se*, or the use of conditional inpainting for this purpose.
- Concerning the challenge of obtaining meaningful gradients with respect to noisy (intermediate) images, as discussed on page 4, this has been addressed before in various approaches such as DiME (by computing gradients with respect to denoised images instead, which is computationally expensive). Both FastDiME and [He et al.](https://openreview.net/forum?id=o3BxOLoxm1) have (more?) efficient implementations of the Tweedie approach.
- [Karunratanakul et al. (2023)](https://openaccess.thecvf.com/content/ICCV2023/papers/Karunratanakul_Guided_Motion_Diffusion_for_Controllable_Human_Motion_Synthesis_ICCV_2023_paper.pdf) is also very related and contains many of the same ingredients presented here. Again, the precise differences should be discussed.

**For SOTA claims, more extensive experiments and standard benchmarks would be needed.**
- Performance is only evaluated quantitatively on three particular tasks previously identified to be particularly challenging for competitor methods. It seems unfair to specifically compare on the case where baselines are known to perform poorly, but not on the ones where they perform well. More generally, a SOTA claim for VCEs would require more extensive experiments. E.g., a comparison on CelebA-smile would already be interesting.
- As noted above, I would suggest adding several more baseline methods such as DiME, FastDiME, and possibly combinations of these methods. E.g. it should be relatively straightforward to implement an I2SB version of FastDiME. This would help further disentangle the benefits provided by the different method components (I2SB, different auto-masking compared to prior work, gradient estimation & stabilization) presented here.

**Overclaims regarding trustworthiness and 'causality' of the resulting explanations**. While I am generally on board concerning the utility of region-constrained VCEs, the authors do go quite a bit overboard in their claims about them, I believe.
- If a VCE is constrained to only change a specific part of the image, this may yield to more intuitively appealing explanations, but the model might actually still be looking at weird shortcut features in other parts of the image (outside the mask). One would then miss this crucial information. This could happen in the proposed method e.g. due to non-localized shortcuts (e.g. [Wang et al.](https://openaccess.thecvf.com/content/ICCV2023/html/Wang_What_do_neural_networks_learn_in_image_classification_A_frequency_ICCV_2023_paper.html)), where every individual pixel of the background would receive a low importance score. It would also miss weak shortcut influence, where the shortcut doesn't completely dominate predictions but still subtly skews confidences scores, such as in racial/ethnic shortcuts in medical imaging ([Zou et al.](https://www.science.org/doi/full/10.1126/science.adh4260), [Yang et al.](https://www.nature.com/articles/s41591-024-03113-4)). Personally, I would be *very* careful with explanation methods that explicitly constrain the explanation to those parts of an image that we believe to be useful a priori. (Cf. e.g. the case study in Fig. 5 in regard to which it is claimed that "RCSB allows for causal inference about the model’s reasoning" - but if we constrain explanations to LangSAM segmentations, we might miss that the classification model is actually primarily looking elsewhere.) The whole setup could thus be seen as strongly *increasing* the risk for confirmation bias, which the authors claim to combat using their method. Also cf. the example in Fig. 1, where it could actually be considered very informative that the counterfactual explanation changes the copyright caption (the authors mention this as a flaw), because it might indicate potential shortcut learning! As a final example in this regard, the authors write that "In this case, RCSB allows for increasing trust in the model, as making the lemons more orange correctly modifies its decision." - but again, this trust is unwarranted: the importance of the object color might in fact be negligible compared to the importance of some background shortcut feature.
- Relatedly, the relative importance of different features is lost in region-constrained VCEs. The fact that one *can* change the head of a bird such that it is classified differently does not mean that these specific properties of the head are actually important to the model's classification. Again, this severely limits any claims of "inferring causally about the model's predictions" - we still don't know which features were actually important for the classification.

To reiterate, I am generally sympathetic to the approach pursued here. However, the discussion of its merits and limitations (!) should be much more nuanced.

**Questions:**

In addition to my questions & suggestions presented above under 'Weaknesses', just a few minor remarks:
- In Fig. 1, what is the "recent SOTA method" used?
- In the caption of Fig. 2, the authors write that "Intermediate images of I2SB are much closer to the data manifold." - I assume that by this, the authors mean that most of the image is 'natural' throughout the whole generation procedure? I was a bit confused here at first, because the trajectory of the cat's face looks very similar in both rows (as would be expected).
- Did the authors consider using the *inverse* of their mask (i.e., only allowing the model to change the background) for assessing a model's robustness to shortcut features?

---

> ### Author Response · Authors · 2024-11-24
> **Response I, part 1/X**
>
> We sincerely thank the reviewer for providing such detailed feedback. We deeply appreciate the insightful analysis of our approach. Below, we first list new results on which we further base the response to the reviewer's concerns.
>
> - We extend our evaluation scheme with three new datasets:
>
>     a. CelebA-HQ following previous works ([Jeanneret et al. (2022)](https://arxiv.org/abs/2203.15636), [Jeanneret et al. (2023)](https://openaccess.thecvf.com/content/CVPR2023/papers/Jeanneret_Adversarial_Counterfactual_Visual_Explanations_CVPR_2023_paper.pdf), [Farid et al. (2023)](https://arxiv.org/abs/2310.06668)) which serves as a testbed for higher resolution ($256\times256$) face images ($30000$ samples).
>
>     b. CelebA also following previous works ([Weng et al. (2024)](https://link.springer.com/chapter/10.1007/978-3-031-73016-0_20) and the above) as a larger dataset of lower resolution face images ($128\times128$, $\sim200 000$ samples).
>
>     c. MNIST as a proof-of-concept sanity check on very low resolution ($32\times32$) to further extend the difficulty spectrum.
>
>     For each dataset, we first train I2SB on the task of inpainting the freeform (20%-30% area) masks, starting from a pretrained diffusion checkpoint (from the work of [Lugmayr et al. (2022)](https://arxiv.org/abs/2201.09865) for CelebA-HQ, [Jeanneret et al. (2022)](https://arxiv.org/abs/2203.15636) for CelebA and from scratch for MNIST). Specific details about the above experiments are now included in new sections in the Appendix: **Other benchmarks** and **Shape modification**.
>
> - We perform two independent user studies:
>
>     a. the first one focused on user preference and explanation usefulness when presented with standard VCEs vs. RVCEs, and the added value coming from the interaction with the explanation creation, where regions can be provided manually by the user.
>
>     b. the second one verifying whether the users are able to better understand the misclassification of the model and identify minimal semantic change to correct it when presented with RVCEs generated by RCSB.
>
>     The results are described in detail in a new section **User studies** in the Appendix.
>
> - We adapt the implementations of DiME to ImageNet and FastDiME to CelebA-HQ and ImageNet to extend the evaluation further. For both methods, we perform an extensive grid search to find optimal hyperparameters on a small subset of images and evaluate the best configuration in a large-scale experiment.
>
> - We provide additional results, where RCSB is evaluated using our automated region extraction approach with the use of lower-importance attributions (**Appendix: Lower-level attributions**).
>
> - We provide a detailed comparison of RCSB to previous works (**Appendix: Comparison to previous works**) and thoroughly discuss its limitations (**Appendix: Limitations**).
>
> **Weaknesses**
>
> **W1. The relationship to several prior works should be discussed more comprehensively.**
>
> Before referring to prior works, we first relate to the definition of RVCEs provided in Eq. 2 of our work, i.e. RVCEs are sampled from a distribution $p(\mathbf{x} \mid \argmax_{y'}f(y' \mid \mathbf{x})=y, (\mathbf{1} - \mathbf{R}) \odot \mathbf{x} = (\mathbf{1} - \mathbf{R}) \odot \mathbf{x}^*)$, where $\mathbf{x}^*$ - original image, $y$ - target class label, $\mathbf{R}$ - region, $f$ - classifier. Please note that this definition a. does not assume the dependence of $\mathbf{R}$ on any other components of the problem (e.g., the classifier $f$) and b. assumes that $\mathbf{R}$ is determined prior to the sampling process and kept fixed. We also emphasize that our understanding of *inpainting* follows the typical definition from the literature (see, e.g., [Liu et al. (2023)](https://arxiv.org/abs/2302.05872), [Rombach et al. (2022)](https://arxiv.org/abs/2112.10752)) and refers to the process of infilling after the region (mask) is provided.
>
> The method ACE of [Jeanneret et al. (2023)](https://openaccess.thecvf.com/content/CVPR2023/papers/Jeanneret_Adversarial_Counterfactual_Visual_Explanations_CVPR_2023_paper.pdf) divides the explanation generation process into pre-explanation (I) and post-processing (II) phases (Algorithms 1. and 2. in their Appendix). Phase I is responsible for modifying $\mathbf{x}^*$ with diffusion model-based denoising and influencing it through a PGD attack with the classifier $f$. Phase II first extracts a binary mask based on the absolute change between $\mathbf{x}*$ and the pre-explanation. Then, the region of the mask is unconditionally (without the use of classifier $f$) inpainted with the diffusion model. Therefore, ACE explicitly assumes that $\mathbf{R}$ depends on $f$ and was not shown to work with $f$-independent regions (contrary to our method). Moreover, they do not perform *conditional* inpainting, since it depends only on the diffusion model, while our approach conditions it with $f$.

---

> > ### Author Response · Authors · 2024-11-24
> > **Response I, 2/X**
> >
> > [Weng et al. (2024)](https://link.springer.com/chapter/10.1007/978-3-031-73016-0_20) propose three approaches: FastDiME, FastDiME-2 and FastDiME-2+. FastDiME modifies guidance of DiME with the use of denoised estimate at each timestep and limits the changes through inpainting only to the most differring region which changes throughout generation. FastDiME-2 first runs FastDiME (w/o Mask), meaning that the changes are not constrained to any region, to obtain a counterfactual. Then, a mask is extracted from it based on thresholded difference to original image, and guidance is performed once again using a fixed mask. FastDiME-2+ extracts the mask by thresholding the result of initial FastDiME run and the guided inpainting is performed again with a fixed mask. In the end, these approaches either do not constrain the changes with a region or extract a classifier-dependent region. Moreover, the initial runs used to obtain a mask for 2-step approaches resemble our automated region extraction method, but require performing an independent denoising procedure with classifier guidance (i.e., using heavy forward passess of the U-Net), while our approach uses light-weight attribution method algorithms to extract the region.
> >
> > Based on the above, we argue that these methods do not sample explanations that fully agree with our definition (Eq. 2), since both approaches assume that $\mathbf{R}$ depends at least partially on $f$, while FastDiME may additionally modify $\mathbf{R}$ throughout the generation. Moreover, none of them showed that picking $\mathbf{R}$ independently of $f$ retains their effectiveness. We elaborate on that aspect further in the experimental results.
> >
> > Regarding the challenge of obtaining meaningful gradients with respect to noisy images, we agree with the reviewer that this was identified by prior works, which we also mentioned in lines 174-187. In terms of the implementation of the Tweedie approach, both [He et al. (2024)](https://openreview.net/pdf?id=o3BxOLoxm1) and [Weng et al. (2024)](https://link.springer.com/chapter/10.1007/978-3-031-73016-0_20) approximate the gradient of the form $\frac{d g(\hat{\mathbf{x}}_0(\mathbf{x}_t))}{d \mathbf{x}_t} = \frac{d g(\hat{\mathbf{x}}_0(\mathbf{x}_t))}{d\hat{\mathbf{x}}_0(\mathbf{x}_t)} \frac{d \hat{\mathbf{x}}_0(\mathbf{x}_t)}{d \mathbf{x}_t}$, where $g$ represents, for example, some classifier-dependent function, by estimating the (U-Net) Jacobian $\frac{d \hat{\mathbf{x}}_0(\mathbf{x}_t)}{d \mathbf{x}_t}$ instead of using autodifferentiation. Although the latter does not mention that explicitly, FastDiME (and its variants) replaces it with identity matrix. While it is true that this approach leads to better efficiency, we emphasize that it is only an approximation of the true Jacobian. We have experimented with this approach previously and during the rebuttal, and found that using autodifferentation performs much better in our case (although with longer runtime) than using some form of approximation. In terms of additional differences to these works, MPGD does not touch upon the topic of explainable artificial intelligence (XAI), which our work is entirely concerned with.
> >
> > The work of [Karunratanakul et al. (2023)](https://openaccess.thecvf.com/content/ICCV2023/papers/Karunratanakul_Guided_Motion_Diffusion_for_Controllable_Human_Motion_Synthesis_ICCV_2023_paper.pdf) deals with the problem of controllable human motion synthesis. The method utilizes two separate SGMs for trajectory and trajectory-conditioned motion generation respectively. Generally, both denoisers introduce some form of guidance - the first one being a standard approach utilizing some goal function, with the second one resembling conditional inpainting adapted to additional conditioning from the keyframe locations. Despite the entire pipeline performing the task of guided generation, this work largely deviates from our approach. We focus specifically on the XAI domain, use a different class of generative models (tractable Schrödinger Bridges) that map arbitrary distributions (assuming access to paired data) and propose a series of adaptations of the classifier's gradients that are not present in the mentioned work, e.g., improved optimization with ADAM and adaptive normalization.

---

> > > ### Author Response · Authors · 2024-11-24
> > > **Response I, 3/X**
> > >
> > > In a general comparison to the aforementioned works, we emphasize that they adapt standard diffusion models (SGMs) to the task at hand, while our method utilizes a more general class of models (tractable Schrödinger Bridges), which naturally require a different treatment, e.g. when correcting the classifier's gradients on the generative trajectory. This is one of the novelties introduced by our work. While some similarities to previous methods exist (e.g., the usage of Tweedie estimate in FastDiME), we highlight that prior works dealt with a problem of different nature. None of them showed that utilizing only the gradient of the classifier (without an $l_2$ or LPIPS loss), enhanced with ADAM stabilization and adaptive normalization, can be such an effective solution when combined with I2SB.
> > >
> > > We thank the reviewer for pointing out that the paper lacks a detailed comparison of this type. It is now included in a new section **Comparison to previous works** in the Appendix due to space limits of the main part.
> > >
> > > **W2. For SOTA claims, more extensive experiments and standard benchmarks would be needed.**
> > >
> > > Our choice of the ImageNet benchmark was based on prior works ([Jeanneret et al. (2023)](https://openaccess.thecvf.com/content/CVPR2023/papers/Jeanneret_Adversarial_Counterfactual_Visual_Explanations_CVPR_2023_paper.pdf), [Augustin et al. (2022)](https://arxiv.org/abs/2210.11841), [Augustin et al. (2024)](https://arxiv.org/abs/2311.17833)) recognizing it as a very challenging testbed due to high complexity of the data (over 1 000 000 samples with 1000 distinct classes) combined with high resolution ($256\times256$). For example, the work of [Jeanneret et al. (2023)](https://openaccess.thecvf.com/content/CVPR2023/papers/Jeanneret_Adversarial_Counterfactual_Visual_Explanations_CVPR_2023_paper.pdf) was considered SOTA on CelebA and CelebA-HQ datasets at the moment of publication, but struggled greatly on ImageNet. Hence, good performance on ImageNet typically translates to high generalizability of the method. In terms of the chosen tasks, we specifically follow previous works ([Jeanneret et al. (2023)](https://openaccess.thecvf.com/content/CVPR2023/papers/Jeanneret_Adversarial_Counterfactual_Visual_Explanations_CVPR_2023_paper.pdf), [Farid et al. (2023)](https://arxiv.org/abs/2310.06668)) to ensure that the comparison is fair. To ensure that our method indeed sets a new SOTA, we follow and extend the reviewer's suggestion, adding CelebA and CelebA-HQ to our evaluation scheme, so that the comparison is performed with all recently published diffusion-based methods. As an additional sanity check, we add evaluation on MNIST, ensuring that RCSB is able to perform well even in the low-resolution scenario.
> > >
> > > In terms of other methods, we adapt the implementation of DiME to ImageNet and FastDiME variants to CelebA-HQ and ImageNet. The hyperparameters of each method are tuned with a large grid search on a small subset of images, and the best configuration is then used following the setting from our paper. Note that we transferred the original source code of these methods to our codebase, which allows to not interfere with their initial implementation. For a more detailed description of the extended evaluation, please head to a newly added section **Other benchmarks** in the Appendix.
> > >
> > > In the following, we briefly mention and discuss the results obtained with our method on CelebA and CelebA-HQ. On CelebA, we highlight that for both *smile* and *age* classes RCSB outperforms previous SOTA (ACE) on FVA, FS, MNAC, COUT and FR, while staying very close in terms of FID, sFID and CD. In particular, following the ImageNet results, our method achieves the best sparsity (COUT), which was a very challenging criterion for previous approaches. Moreover, the nature of our approach explains well the higher values of CD. Since this metric measures how far the correlation of face attributes in the explanations deviates from the original counterpart, it is intuitive that a semantically aligned region constraint might break many correlations of the original data. Here, we must argue that CD is not, therefore, a pure measure of performance, but rather an indicator of a specific property which might or might not desired.

---

> ### Author Response · Authors · 2024-11-24
> **Response I, 4/X**
>
> **CelebA (smile)**
>
> | Method        | FID  | sFID | FVA  | FS   | MNAC  | CD   | COUT  | FR    |
> |---------------|-------|------|------|------|-------|------|-------|-------|
> | DiVE          | 29.4  | -    | 97.3 | -    | -     | -    | -     | -     |
> | DiVE $^{100}$ | 36.8  | -    | 73.4 | -    | 4.63  | 2.34 | -     | -     |
> | STEEX         | 10.2  | -    | 96.9 | -    | 4.11  | -    | -     | -     |
> | ACE $\ell_1$  | 1.27  | 3.97 | 99.9 | 0.87 | 2.94  | 1.73 | 0.78  | 97.6  |
> | ACE $\ell_2$  | 1.90  | 4.56 | 99.9 | 0.87 | 2.77  | 1.56 | 0.62  | 84.3  |
> | DiME          | 3.17  | 4.89 | 98.3 | 0.73 | 3.72  | 2.30 | 0.53  | 97.0  |
> | FastDiME      | 4.18  | 6.13 | 99.8 | 0.76 | 3.12  | 1.91 | 0.44  | 99.0  |
> | FastDiME-2    | 3.33  | 5.49 | 99.9 | 0.77 | 3.06  | 1.89 | 0.44  | 99.4  |
> | FastDiME-2+   | 3.24  | 5.23 | 99.9 | 0.79 | 2.91  | 2.02 | 0.41  | 98.9  |
> | RCSB (ours)   | 2.98  | 4.79 | 100.0| 0.91 | 2.24  | 2.78 | 0.87  | 99.8  |
>
> **CelebA (age)**
>
> | Method        | FID  | sFID | FVA  | FS   | MNAC  | CD   | COUT  | FR    |
> |---------------|-------|------|------|------|-------|------|-------|-------|
> | DiVE          | 33.8  | -    | 98.1 | -    | 4.58  | -    | -     | -     |
> | DiVE $^{100}$ | 39.9  | -    | 52.2 | -    | 4.27  | -    | -     | -     |
> | STEEX         | 11.8  | -    | 97.5 | -    | 3.44  | -    | -     | -     |
> | ACE $\ell_1$  | 1.45  | 4.12 | 99.6 | 0.78 | 3.20  | 2.94 | 0.72  | 96.2  |
> | ACE $\ell_2$  | 2.08  | 4.62 | 99.6 | 0.80 | 2.94  | 2.82 | 0.56  | 77.5  |
> | DiME          | 4.15  | 5.89 | 95.3 | 0.67 | 3.13  | 3.27 | 0.44  | 99.0  |
> | FastDiME      | 4.82  | 6.76 | 99.2 | 0.74 | 2.65  | 3.80 | 0.36  | 98.6  |
> | FastDiME-2    | 4.04  | 6.01 | 99.6 | 0.75 | 2.63  | 3.80 | 0.37  | 99.3  |
> | FastDiME-2+   | 3.60  | 5.59 | 99.7 | 0.77 | 2.44  | 3.76 | 0.32  | 98.7  |
> | RCSB (ours)   | 2.94  | 4.94 | 99.9 | 0.88 | 2.14  | 3.63 | 0.81  | 99.3  |
>
> On CelebA-HQ, the comparison remains largely the same. Our method stays very close to SOTA results of ACE in terms of FID, sFID and CD, while outperforming all other approaches on FVA, FS, MNAC, COUT and FR.
>
> **CelebA-HQ (smile)**
>
> | Method       | FID   | sFID  | FVA    | FS    | MNAC  | CD    | COUT  | FR    |
> |--------------|-------|-------|--------|-------|-------|-------|-------|-------|
> | DiVE         | 107.0 | -     | 35.7   | -     | 7.41  | -     | -     | -     |
> | STEEX        | 21.9  | -     | 97.6   | -     | 5.27  | -     | -     | -     |
> | DiME         | 18.1  | 27.7  | 96.7   | 0.67  | 2.63  | 1.82  | 0.65  | 97.0  |
> | ACE $\ell_1$ | 3.21  | 20.2  | 100.0  | 0.89  | 1.56  | 2.61  | 0.55  | 95.0  |
> | ACE $\ell_2$ | 6.93  | 22.0  | 100.0  | 0.84  | 1.87  | 2.21  | 0.60  | 95.0  |
> | LDCE         | 13.6  | 25.8  | 99.1   | 0.76  | 2.44  | 1.68  | 0.34  | -     |
> | FastDiME-2+  | 16.51 | 31.4  | 99.9   | 0.87  | 1.43  | 4.16  | 0.28  | 87.1  |
> | RCSB (ours)       | 3.04  | 20.0  | 100.0  | 0.93  | 1.22  | 3.22  | 0.83  | 98.9  |
>
> **CelebA-HQ (age)**
>
> | Method       | FID   | sFID  | FVA    | FS    | MNAC  | CD    | COUT  | FR    |
> |--------------|-------|-------|--------|-------|-------|-------|-------|-------|
> | DiVE         | 107.5 | -     | 32.3   | -     | 6.76  | -     | -     | -     |
> | STEEX        | 26.8  | -     | 96.0   | -     | 5.63  | -     | -     | -     |
> | DiME         | 18.7  | 27.8  | 95.0   | 0.66  | 2.10  | 4.29  | 0.56  | 97.0  |
> | ACE $\ell_1$ | 5.31  | 21.7  | 99.6   | 0.81  | 1.53  | 5.40  | 0.40  | 95.0  |
> | ACE $\ell_2$ | 16.4  | 28.2  | 99.6   | 0.77  | 1.92  | 4.21  | 0.53  | 95.0  |
> | LDCE         | 14.2  | 25.6  | 98.0   | 0.73  | 2.12  | 4.02  | 0.33  | -     |
> | FastDiME-2+  | 26.0  | 40.3  | 99.6   | 0.81  | 3.15  | 4.36  | 0.31  | 92.6  |
> | RCSB (ours)       | 4.92  | 27.3  | 100.0  | 0.96  | 1.47  | 5.16  | 0.80  | 99.4  |
>
> Regarding the comment connected to combining RCSB with FastDiME, we note that the ablation study included in the main part of our paper contains an adaptation of
> Manifold Constrained Gradient (MCG, [Chung et al. (2022)](https://arxiv.org/abs/2206.00941)), which performs the task of inpainting using standard SGMs and adopts the Tweedie trick for the gradient at each step. There, the Jacobian is computed using autodifferentiation. Hence, this variant of MCG is effectively a combination of FastDiME (use of Tweedie, but with exact computation of Jacobian) with our automated region extraction approach providing a fixed region for the entire conditioning process.
>
> **W3. Overclaims regarding trustworthiness and 'causality' of the resulting explanations.**
>
> We thank the reviewer for such comprehensive analysis of the different aspects of RVCEs. We will try to clarify the claims made in the paper, as it is true that they may not have been conveyed properly.

---

> > ### Author Response · Authors · 2024-11-24
> > **Response I, 5/X**
> >
> > We certainly agree with the reviewer's view on the fact that, despite constraining the changes to a specific region, the model might still be looking at other shortcut features outside the mask. In the paper, we do not claim to solve this problem. Our main motivation stems from the definition of a VCE, which is based on identifying minimal semantic modification in the image that changes the decision of the classifier. We argue that an RVCE with region constraint disconnected from the shortcut feature, which effectively flips the model's decision, is an entirely valid explanation, since even if the shortcut feature is still used, the model's prediction has changed, which solves the task of counterfactual explanation generation.
> >
> > While we understand the concern that `one would then miss this crucial information` (about the shortcut feature being used), there are no theoretical guarantees that any counterfactual explanation is actually able to provide such information, since even if the explanation modified a given feature, one cannot be certain that it happened due to the model's use of it. Consequently, this cannot be also guaranteed in practice, as counterfactual explanation generation is based on some mixture of the signal from the classifier and approximate data prior (generative model in this case), which also influences the image.
> >
> > In terms of non-localized shortcuts, we do not entirely agree with the reviewer, since our framework makes no assumptions about the structure of the region. In practice, we verified that even when using raw attributions (in the form of pixels dispersed across the entire image), RCSB is still able to change the model's decision, although in this case they will not be semantically relevant, which goes against the definition of a VCE. Moreover, regarding the work of [Wang et al. (2023)](https://openaccess.thecvf.com/content/ICCV2023/html/Wang_What_do_neural_networks_learn_in_image_classification_A_frequency_ICCV_2023_paper.html), we think that VCEs are not a proper tool for detecting low-frequency shortcuts, as the mentioned paper already provides sample-based explanations for the model's decision in the form of low-frequency versions of the original image, but those can hardly be considered semantically valid, i.e., originating from the data distribution. We are open to further discussion about how such VCE would like.
> >
> > Once again, in terms of weak shortcut influence, VCEs are not defined as explanations that must verify or detect their existence. We understand that they can be used as a tool in such task (e.g., [Weng et al. (2024)](https://link.springer.com/chapter/10.1007/978-3-031-73016-0_20)), but the ongoing research also does not treat it as its primary focus (cite). To the best of our understanding, choosing a region that contains racial/ethnic features is possible, and hence could also be used in the context of our method, since we already show the plausibility of user-defined regions.
> >
> > In terms of methods that `explicitly constrain the explanation to those parts of an image tha we believe to be useful a priori`, we want to firstly emphasize that Fig. 5 depicts RVCEs obtained with the automated approach, which uses attribution methods to find the region constraint, and was not biased by our a priori knowledge. Regarding the experiment with LangSAM segmentations, our understanding of `causal inference about the model's reasoning` is based on being able to infer that the new prediction of the model was caused solely by modifying the region containing precisely the indicated object. Undoubtedly, the model still uses the entire image to make the prediction and we do not claim that it does not look at other features. However, it is a fact that its decision changes, even if it `is actually primarily looking elsewhere`, and hence the explanation is a valid VCE.
> >
> > Regarding Fig. 1, we agree that it could be considered very informative for any explanation to indicate the model's usage of a caption. However, we do not mention the modification of this caption in the VCE from Fig. 1 as a flaw. Rather, in lines 52-54, we describe that, because this change happens simultaneously with modifications of other features, it is difficult (if not impossible) to verify its actual impact on the model. For example, this caption might have been modified due to the inability of the generative model to perfectly reconstruct it, but could also indicate shortcut learning, making the explanatory process ambiguous. With RVCEs, one could add a region constraint connected solely with the caption to verify that.
> >
> > Considering the *Lemon* $\rightarrow$ *Orange* task in Fig. 6, we once again agree that the model might be using some background shortcut feature. We acknowledge the point about the increased trust being unwarranted and decided to remove the mentioned claim by replacing it with a statement about alignment with human intuition.

---

> > > ### Author Response · Authors · 2024-11-24
> > > **Response I, 6/X**
> > >
> > > In terms of the point made by the reviewer about the loss of relative importance of different features in RVCEs, we must argue that since no ground truth exists to verify the correctness of a counterfactual explanation (at least in non-synthetic tasks considered in our work), one is not able to actually verify this relative importance in any kind of explanations of counterfactual type (neither standard VCEs nor RVCEs). This is the exact problem that we address with our method - since one does not know if even small magnitude of change of some feature influenced the decision, it might be better to simply leave most of the features as they are and constrain the changes to a specific region. We argue that the fact that we can change the model's decision using only, e.g., the head of a bird does in fact bring added value to the process. This is because we explicitly show that modifying the head only is sufficient to make the model think differently. We do not claim that now the model relies solely on the new features. Rather, we state that the entire image, with the head region modified only, is now predicted differently by the model. Because this region is provided in simple form, one can infer causally about the prediction, i.e., *the prediction changed because the head changed*, but this does not mean that one can now identify all of the relationships between the image features for the model, which we do not state.
> > >
> > > We also want to mention how our initial claims about RVCEs translate to real-world scenarios by briefly mentioning the results of two performed user studies. In the first one, focused on the usefulness of RVCEs when compared with standard VCEs and the added value coming from the interactive process, 86.6% of participants found RVCEs to be more useful than VCEs in understanding and explaining the model's decision-making. The individual reasons for the vote in favor of RVCEs were in general focused on more localized semantic change, better interpretability and alignment with human intuition. Moreover, 93.3% answered in favor of our method when presented with a comparison of standard (static) interaction with VCEs vs. the possibility to provide the regions for RVCEs manually. These positive votes were mostly motivated by the possibility to verify regions that align with human understanding of the problem and potential incorporation of domain experts. In the second study, RVCEs were used to help a different group of participants in understanding model's failure cases and the minimal semantic change needed to correct them. There, in over 80% of cases, RVCEs were found to be an effective tool for the considered problem. 90.9% of participants were able to indicate the specific semantic features that were missing for the model to predict the image correctly and concluded that they better understand the failure case after observing RVCEs. All participants considered RVCEs to be a useful tool in explaining the model's decision-making. For more details, please see the newly added section **User studies** in the Appendix.
> > >
> > > Finally, we want to emphasize that while we regard the problem of shortcut learning and detection as highly valuable, our paper is not concerned with it. In the past, VCEs have proven to be an effective tool for detecting such features, but their formulation and definition are considered independently from that problem. One last point that we would like to address is the case of the method's possibility to miss `weak shortcut influence`. In the newly added section **Lower-level attributions** in the Appendix, we provide quantitative results from an experiment in which the initial attributions were first zeroed out before being used by the automated region extraction if they exceeded the *q*th quantile. There, we considered $q\in\{ 0.3, 0.4, 0.5, 0.6, 0.7, 0.8, 0.9\}$ and used $\text{RCSB}^A$ configuration, where the region's area is equal to 10% of total image coverage. Below, we include the table from the mentioned section describing the results. These indicate that RCSB largely preserves its performance from the case where only the highest attributions are considered (Table 1. in our paper). Importantly, this means that one may explore the influence of regions with much lower attributions (e.g., weak influence shortcuts) using our method and still obtain RVCEs very effectively. We hope that these results at least partially address the reviewer's concern in this case.
> > >
> > > **Zebra -- Sorrel**
> > >
> > > | $q$  | FID  | sFID  | S³   | COUT  | FR    |
> > > |------|------|-------|------|-------|-------|
> > > | 0.9  | 5.2  | 13.8  | 0.87 | 0.62  | 91.6  |
> > > | 0.8  | 4.6  | 13.3  | 0.87 | 0.53  | 88.0  |
> > > | 0.7  | 3.9  | 12.7  | 0.88 | 0.48  | 85.7  |
> > > | 0.6  | 3.6  | 12.4  | 0.89 | 0.45  | 82.4  |
> > > | 0.5  | 3.6  | 12.4  | 0.89 | 0.40  | 78.5  |
> > > | 0.4  | 3.7  | 12.3  | 0.89 | 0.38  | 77.7  |
> > > | 0.3  | 3.9  | 12.6  | 0.89 | 0.40  | 79.2  |

---

> > > > ### Author Response · Authors · 2024-11-24
> > > > **Response I, 7/X**
> > > >
> > > > **Cheetah -- Cougar**
> > > >
> > > > | $q$  | FID  | sFID  | S³   | COUT  | FR    |
> > > > |------|------|-------|------|-------|-------|
> > > > | 0.9  | 6.1  | 17.1  | 0.92 | 0.88  | 99.8  |
> > > > | 0.8  | 4.4  | 15.7  | 0.91 | 0.80  | 99.1  |
> > > > | 0.7  | 3.5  | 15.1  | 0.91 | 0.74  | 97.4  |
> > > > | 0.6  | 2.9  | 14.6  | 0.92 | 0.68  | 94.3  |
> > > > | 0.5  | 2.7  | 14.5  | 0.92 | 0.65  | 93.1  |
> > > > | 0.4  | 2.8  | 14.6  | 0.92 | 0.65  | 93.4  |
> > > > | 0.3  | 3.1  | 14.9  | 0.92 | 0.69  | 95.7  |
> > > >
> > > > **Egyptian cat -- Persian cat**
> > > >
> > > > | $q$  | FID   | sFID  | S³   | COUT  | FR    |
> > > > |------|-------|-------|------|-------|-------|
> > > > | 0.9  | 13.5  | 31.4  | 0.86 | 0.89  | 99.9  |
> > > > | 0.8  | 11.0  | 28.6  | 0.85 | 0.84  | 99.3  |
> > > > | 0.7  | 9.2   | 26.9  | 0.85 | 0.79  | 98.5  |
> > > > | 0.6  | 7.8   | 25.5  | 0.85 | 0.73  | 96.1  |
> > > > | 0.5  | 6.9   | 24.7  | 0.86 | 0.69  | 94.6  |
> > > > | 0.4  | 6.6   | 24.5  | 0.87 | 0.65  | 93.1  |
> > > > | 0.3  | 7.5   | 25.3  | 0.86 | 0.67  | 94.2  |
> > > >
> > > > **Questions**
> > > >
> > > > **Q1.** The recent SOTA method is DVCE ([Augustin et al. (2022)](https://arxiv.org/abs/2210.11841)). Since we treated Fig. 1 as a conceptual introduction, we did not want to bias the reader in any way, hence decided to skip the name.
> > > >
> > > > **Q2.** Indeed, we meant that most of the image is `natural` throughout the generation process. In the initial steps of the reverse diffusion process, images from SGMs are highly distorted with noise, making it almost impossible to identify the cat in the image. For I2SB, the complement of the region is kept the same as the original throughout the entire process.
> > > >
> > > > **Q3.** One of the practical limitations of I2SB is that the ImageNet checkpoint was trained using masks with 20-30% area coverage, meaning that it struggles to correctly inpaint masks covering area much greater than that (in practice, the quality starts degrading over 45% area coverage, which we mention in lines 399-401). Hence, we leave the mentioned experiment as future work, when we verify that larger areas are also possible with I2SB, which is entirely plausible from a theoretical point of view.
> > > >
> > > > We hope that the above comments effectively address the reviewer's concerns. Once again, we are very thankful for such constructive feedback and look forward to further discussion.

---

> ### Comment · Reviewer_2FX9 · 2024-11-28
>
> I would like to thank the authors for this incredibly thorough and comprehensive rebuttal in such a short period of time! This must have been a huge amount of work and is much appreciated.
>
> The various additional experiments do add a lot of value to the paper and provide significant further support to the arguments made by the authors, I believe.
>
> The very detailed comparison with prior work is also very much appreciated; I believe this will be very valuable for future readers. Just one minor nitpick on this: the argument that some of the prior works do not fit *exactly* into the authors' Eq. 2 really seems quite superficial. E.g. FastDiME does conditional inpainting and could really also use any mask, e.g. LangSAM-based as suggested here. The differences in the core setup really seem quite negligible. This is fine - there are enough other novel contributions in this piece - but then they should also not be unnecessarily overstated. I would strongly suggest citing prior works that also perform conditional inpainting either in section 2 or the conditional inpainting part of section 3, which currently still reads as if the authors were the first to propose this. (Given that esp. FastDiME seems like the most closely related prior work, one could also ask whether this should not be included in the results in the main paper, but I will leave this decision to the authors' discretion.)
>
> I do have some concerns about the interpretation of the newly added user study, however. I concur with everything that the authors write on the subject of the trustworthiness of the resulting explanations. I fully agree that the purpose of VCEs is neither to detect shortcuts nor to fully explain a model, that all the explanations generated in the paper are valid VCEs, and that baseline attribution maps also have many problems. *However*, there is a large risk when specifically designing an explanation method that is constrained to only change semantically meaningful features of an image: since such explanations align well with human intuition (as the authors themselves note), users will be strongly biased to assume that 'if the model has understood that changing the color of a lemon turns it into something like an orange, then it is likely that the model has well-internalized the concepts of lemons and oranges and that it will generally perform well'. However, as I indicated in my initial review and as the authors also seem to agree, this may be a false conclusion - the model can still be arbitrarily non-robust. In other words, the constraining of explanations to semantically meaningful variations may induce trust on the user's side that is unwarranted. This is actually reflected well in what the authors write about their user study: "The answers generally focused on the semantic change being more localized, easier to interpret and better aligned with human intuition". The fact that an explanation is 'aligned with human intuition' should be completely irrelevant for evaluating the utility of an explanation method: if the model does something very weird, I want to see that, and not an explanation that looks intuitively pleasing but does not reflect the model's actual decision process well. As far as I understand it, there was no evaluation in this study of whether the users *actually* understood the model's decision process better, or whether they just *felt* they did? As it stands then, the user study seems to show that users like these explanations, but not whether the explanations are any good in the sense that they really help them understand the model (better). See e.g. [Davis et al.](https://ieeexplore.ieee.org/abstract/document/9307625) or [Jacovi et al.](https://dl.acm.org/doi/pdf/10.1145/3442188.3445923), specifically section 8.2 on the distinction between *warranted* and *unwarranted* trust.
>
> Nevertheless, in light of the otherwise excellent rebuttal and significantly improved manuscript, I am already raising my score.

---

> > ### Author Response · Authors · 2024-11-29
> >
> > We sincerely thank the reviewer for appreciating our efforts and a very fruitful discussion with detailed comments. Regarding the user study, we agree with the reviewer that the mentioned aspect is of high relevance in the context of our approach. For example, one could try to create a synthetic dataset, where the model's decision-making process can be assumed to be *known* and which fits well with the concept of RVCEs, in order to verify their helpfulness to the user in better understanding the model. Since the scale of such an evaluation can already be rated as high and touches on a concept which our work only introduces, we believe this is an important future research direction.

---

> > > ### Comment · Reviewer_2FX9 · 2024-11-30
> > >
> > > Absolutely, I was certainly not expecting such a study to be added here. (I believe people sometimes also evaluate users' model understanding by having them predict the model's decisions for new examples. There is a lot of interesting work on this subject in the XAI/HCI fields.) For this piece, I would just suggest carefully rephrasing a few sentences in relation to that user study accordingly.
> > >
> > > Thank you likewise for a very productive discussion phase! :-)

---

### Official Review · Reviewer_nGZZ · 2024-11-04

**Soundness:** 3
**Presentation:** 3
**Contribution:** 3
**Rating:** 6
**Confidence:** 3

**Summary:**

This paper addresses the task of Visual Contour Factual Explanations (VCE) by introducing regional constraints to enhance the specificity and clarity of the explanations. Additionally, it proposes a method, RCSB, to generate counterfactual results based on region-specific information. The effectiveness of the proposed approach is evaluated on both generated and manually designed regions.

**Strengths:**

The paper is clearly written, with most details of the proposed method presented effectively. It includes numerous qualitative results that demonstrate the method’s effectiveness and clearly define the problem.

**Weaknesses:**

1. The paper demonstrates that the proposed method achieves better performance when a semantic region constraint is applied. However, it would be helpful to explore the effects when the region is not semantically relevant—such as when using a randomly selected area with similar shape and size—to understand how critical semantic relevance is to the method’s effectiveness.



2. In Table 1, three configurations of the proposed method are presented as settings A, B, and C. While it appears these configurations differ based on specific hyperparameter choices, could you provide additional context or rationale for highlighting these particular settings? Are there theoretical or practical reasons to show them in this table?



3. The results for LangSAM-based regions have a considerably higher FID score than those for other regions, suggesting that this style of regions may pose unique challenges. Could you clarify why this region is so different or particularly difficult for the method? Additional explanation on this would provide valuable insight into the model’s limitations.



4. The paper briefly mentions "confirmation bias," but could you expand on how this bias is defined in the original problem and mitigated by the proposed method?

**Questions:**

See strengths and weaknesses.

---

> ### Author Response · Authors · 2024-11-16
> **Response I, part 1/2**
>
> We sincerely thank the reviewer for the valuable feedback. Below, we address the mentioned weaknesses and concerns.
> - [W1.] Thank you for suggesting this analysis. We agree that it is an insightful scenario. Please note that an experiment of this kind has already been performed and described in our paper (**Discovering complex patterns with interactive RVCEs** subsection, results in Table 2. (B) of our work), but since it was easy to overlook, we elaborate on it in the following. In the paper, we simulate user interaction for the region proposal by evaluating our method on randomly assigned *freeform* masks from the work of Saharia et al. (2022) (for examples of these masks, see the work of Liu et al. (2023)). Importantly, these masks represent blob-like shapes with similar shape and size to the regions obtained with, e.g., our automated approach, but are not semantically relevant to the image as they are assigned randomly across the entire dataset. Table 2. (B) shows that the semantic relevance certainly helps our method in performing better, but is not critical to its effectiveness (by comparing Table 2. (B) to, e.g., Table 1.). This is intuitive, since freeform masks have no connection with the classifier's prediction, but should still give our method sufficient capabilities to modify the given region so the model's decision changes. To verify that, we also updated the Appendix (**Freeform masks** subsection) with example RVCEs in *sorrel* $\rightarrow$ *zebra* task from Table 2. (B). We observe that our method mostly focuses on modifying the intersection of the random mask with parts that, intuitively, the classifier should be using (e.g., horse skin), while leaving others (e.g., background, sky) mostly unchanged. These observations explain well why the performance of our method with freeform masks is still very comparable to that of the automated approach.
> - [W2.] Regarding Table 1., we included three different configurations mainly to highlight the method's versatility across different areas of the extracted regions (respectively 10%, 20% and 30% of total image area for A, B and C). For each area, we found its respective configuration manually on a held-out set of samples, but observed in general that our method is not very sensitive to hyperparameter changes (see Table 6. in the Appendix). Due to lack of space, we did not elaborate on that aspect in the manuscript and are thankful for pointing it out.
>
> **References**
>
> Guan-Horng Liu, Arash Vahdat, De-An Huang, Evangelos Theodorou, Weili Nie, and Anima
> Anandkumar. I2SB: Image-to-Image Schr¨ odinger Bridge. In International Conference on Ma-
> chine Learning, pp. 22042–22062. PMLR, 2023.
>
> Chitwan Saharia, William Chan, Huiwen Chang, Chris Lee, Jonathan Ho, Tim Salimans, David
> Fleet, and Mohammad Norouzi. Palette: Image-to-image diffusion models. In ACM SIGGRAPH
> 2022 conference proceedings, pp. 1–10, 2022.

---

> ### Author Response · Authors · 2024-11-16
> **Response I, part 2/2**
>
> - [W3.] Regarding the results for LangSAM and increased FID and sFID (Table 2. (A)), we follow the justification provided in lines 404-409. The regions extracted with LangSAM entirely cover the main objects representing the given class, which our method modifies to depict objects judged as a different class by the classifier. This naturally justifies that the distribution of the generated explanations shifts far from the original data distribution (for the initial class). While this shift is expected, it may suggest that the generated samples could be lacking in realism. However, as indicated by the S$^3$ metric in Table 2. (A), this does not seem to be the case, since the representational similarity of the images is well-preserved. Moreover, the additional influence on FID stems from the LangSAM masks typically covering an area greater than total 30%. This allows modifying a larger part of the image than, e.g., the results for the automated approach, further justifying the inflation of FID. Hence, we do not consider these specific results as an indication of our method's limitation, as other metrics still indicate its high effectiveness.
>
> - [W4.] We agree that the confirmation bias aspect could be introduced better in the manuscript. Here, we describe it more deeply using the images from Figure 1. of our work. Specifically, previous VCEs modify the given image with no constraints on the region. This leads to an explanation that simultaneously modifies the bird's head, feathers, the branch, background and the text caption in the bottom-right corner. Confirmation bias refers to us, humans, having a tendency to search for or favor the information that confirms our prior beliefs. In this case, this may lead to, based on the explanation, concluding that the model bases its reasoning on the features that changed the most visibly and that are consistent with intuition, i.e., change in feathers and head. However, the change of the caption and branch are much less intuitive and visible, but may constitute the main reasons behind the model's new prediction due to spurious correlation of the data. Our method overcomes this limitation, since it restricts modifications only to the features of interest indicated by the prespecified region.
>
> We hope that the above comments clarify the reviewer's concerns. Thank you for appreciating the well written paper and precisely set research goals. The proposed method accomplishes them while greatly outperforming previous SOTA and creates a new benchmark in the field of counterfactual explanations for image data.  We are open to elucidating any further ambiguities and whatever remains unclear. We would also like to encourage the reviewer to increase the proposed score if sufficient justifications were provided or suggest some additional directions to help in that decision.
>
> **References**
>
> Guan-Horng Liu, Arash Vahdat, De-An Huang, Evangelos Theodorou, Weili Nie, and Anima Anandkumar. I2SB: Image-to-Image Schr¨ odinger Bridge. In International Conference on Ma- chine Learning, pp. 22042–22062. PMLR, 2023.
>
> Chitwan Saharia, William Chan, Huiwen Chang, Chris Lee, Jonathan Ho, Tim Salimans, David Fleet, and Mohammad Norouzi. Palette: Image-to-image diffusion models. In ACM SIGGRAPH 2022 conference proceedings, pp. 1–10, 2022.

---

> > ### Author Response · Authors · 2024-12-01
> >
> > Dear reviewer,
> > due to the discussion period coming to an end, we highly encourage you to inform us whether the provided answers properly address your concerns. Moreover, during the rebuttal, we have conducted a number of additional experiments that extend our initial results with evidence supporting many claims and capabilities of our approach. Since many of them also address your comments, we provide a brief summary of the related ones below.
> >
> > **Weaknesses**
> >
> > **W1. The paper demonstrates that (...) when the region is not semantically relevant (...). **
> >
> > To further comment on the semantic relevance of the chosen regions, we perform an experiment in which our automated approach extracts the region by first zeroing out the attributions with values over some quantile $q$. This way, the region loses semantic relevance to the most 'important' features and focuses on what the attribution method judges as less influential on the classifier. Below, we include the results of our method on the three main tasks considered in the paper. Here, we use the $\text{RCSB}^A$ configuration ($10\%$ of total area coverage) with $q\in\{ 0.3, 0.4, 0.5, 0.6, 0.7, 0.8, 0.9\}$. Hence, with decreasing $q$, the utilized region loses the initial semantic relevance with the most important features. This way, more *obvious* features can be ommitted, and the region is able to focus on unusual parts of the image. Importantly, despite using attributions of smaller magnitude, this approach does not visibly impact our method's performance and efficiency, further highlighting its versatility even when the initial semantic relevance is lost.
> >
> > **Zebra -- Sorrel**
> >
> > | $q$  | FID  | sFID  | S³   | COUT  | FR    |
> > |------|------|-------|------|-------|-------|
> > | 0.9  | 5.2  | 13.8  | 0.87 | 0.62  | 91.6  |
> > | 0.8  | 4.6  | 13.3  | 0.87 | 0.53  | 88.0  |
> > | 0.7  | 3.9  | 12.7  | 0.88 | 0.48  | 85.7  |
> > | 0.6  | 3.6  | 12.4  | 0.89 | 0.45  | 82.4  |
> > | 0.5  | 3.6  | 12.4  | 0.89 | 0.40  | 78.5  |
> > | 0.4  | 3.7  | 12.3  | 0.89 | 0.38  | 77.7  |
> > | 0.3  | 3.9  | 12.6  | 0.89 | 0.40  | 79.2  |
> >
> > **Cheetah -- Cougar**
> >
> > | $q$  | FID  | sFID  | S³   | COUT  | FR    |
> > |------|------|-------|------|-------|-------|
> > | 0.9  | 6.1  | 17.1  | 0.92 | 0.88  | 99.8  |
> > | 0.8  | 4.4  | 15.7  | 0.91 | 0.80  | 99.1  |
> > | 0.7  | 3.5  | 15.1  | 0.91 | 0.74  | 97.4  |
> > | 0.6  | 2.9  | 14.6  | 0.92 | 0.68  | 94.3  |
> > | 0.5  | 2.7  | 14.5  | 0.92 | 0.65  | 93.1  |
> > | 0.4  | 2.8  | 14.6  | 0.92 | 0.65  | 93.4  |
> > | 0.3  | 3.1  | 14.9  | 0.92 | 0.69  | 95.7  |
> >
> > **Egyptian cat -- Persian cat**
> >
> > | $q$  | FID   | sFID  | S³   | COUT  | FR    |
> > |------|-------|-------|------|-------|-------|
> > | 0.9  | 13.5  | 31.4  | 0.86 | 0.89  | 99.9  |
> > | 0.8  | 11.0  | 28.6  | 0.85 | 0.84  | 99.3  |
> > | 0.7  | 9.2   | 26.9  | 0.85 | 0.79  | 98.5  |
> > | 0.6  | 7.8   | 25.5  | 0.85 | 0.73  | 96.1  |
> > | 0.5  | 6.9   | 24.7  | 0.86 | 0.69  | 94.6  |
> > | 0.4  | 6.6   | 24.5  | 0.87 | 0.65  | 93.1  |
> > | 0.3  | 7.5   | 25.3  | 0.86 | 0.67  | 94.2  |
> >
> > **W4. The paper briefly mentions "confirmation bias," **
> >
> > As the notion of confirmation bias is mainly connected to the human perception, we conduct two independent user studies to verify how the proposed method influences the user's understanding of the model's decision-making process.
> >
> > The first study compared the effectiveness of RVCEs with standard VCEs and explored the benefits of incorporating an interactive process. Results indicated that 86.6% of participants found RVCEs more effective for understanding and explaining the model's decisions. This preference was attributed to RVCEs’ ability to enable localized semantic changes, enhance interpretability, and better align with human intuition. Additionally, 93.3% favored RVCEs over standard static VCE interactions when allowed to manually specify regions. This was due to their ability to validate regions aligned with human understanding and incorporate domain expert input.

---

> > > ### Author Response · Authors · 2024-12-01
> > >
> > > In the second study, RVCEs were used to assist another group in analyzing model failure cases and identifying minimal semantic changes required for correction. The experiment was conducted over five iterations. Participants were first shown the model’s incorrect prediction on a given image, the correct class, and the key characteristics of both the predicted and true classes. They were then provided with the region constraint generated by our method and a series of RVCEs illustrating the semantic changes needed to correct the model’s prediction. Over 80% of participants found RVCEs effective in understanding why the model misclassified the image. Furthermore, 90.9% successfully identified the missing semantic features necessary for accurate predictions and reported an improved understanding of failure cases after using RVCEs. All participants agreed that RVCEs were a valuable tool for explaining the model's decisions.
> > >
> > > Details regarding both of these studies can be found in a new section **User studies** in the Appendix.
> > >
> > > We believe that the results of these studies further support that our method effectively addresses the notion of confirmation bias. This is mainly reflected by the first study showing that users are able to precisely localize the changes, leaving no room for undetected or unobserved changes. Also, the second study makes effective use of this property in clarifying the model's reasons for misclassification, which is clearly missing in standard VCEs, which may lead to confirmation bias in this case due to small but not clearly visible modifications.
> > >
> > > We would greatly appreciate if the reviewer could provide feedback on whether the above comments properly address the initial concerns and engage in further discussion.

---

> > > > ### Author Response · Authors · 2024-12-01
> > > >
> > > > Dear reviewer,
> > > > following up on our previous comment, we would be extremely grateful if you could refer to whether the responses provided above adequately address your concerns and possibly engage further in the discussion.

---

### Author Response · Authors · 2024-11-24
**Summary of added content**

We sincerely thank all the reviewers for the insightful feedback and apologize for the late responses, caused mainly by the exceptionally high (6) total number of reviews our paper has received. Below, we summarize the changes and additions to the main manuscript and the Appendix, which attempt to address all of the reviewers concerns.

1. We extended our quantitative and qualitative evaluation with three new benchmarks:

    a. CelebA and CelebA-HQ were added to provide a fair comparison with all previously published diffusion-based methods for VCE generation,

    b. MNIST was added as additional proof-of-concept and to showcase the method's ability to modify shapes in addition to texture and color.

2. We performed two independent user studies:

    a. the first one addresses the usefulness and helpfulness of the introduced RVCEs when compared to standard VCEs, and the potential added value coming from the interactive process, which allows the users to manually provide the region constraint,

    b. the second one aims at evaluating whether our RCSB method is able to help the users in understanding the model's failure cases and identifying the minimal semantic change to correct the prediction.

3. We added new sections in the Appendix based on the reviewers requests:

    a. Limitations - deeply investigates the method's limitations, including the reviewer's concerns,

    b. Comparison to previous works - provides a thorough theoretical and practical comparison of our approach to previous works,

    c. Freeform masks - provides qualitative examples to one of the reviewer's concerns connected to the scenario where the method is evaluated on regions that are not semantically relevant to the image content,

    d. Unintuitive classes - provides qualitative examples as an answer to whether the method is able to provide explanations where the class pairings are not semantically connected,

    e. Shape modification - provides qualitative and quantitative results to evaluate that the method is able to modify shapes in addition to texture and color,

    f. Lower-level attributions - provides empirical proof of the method's ability to generate RVCEs when using attributions of smaller importance,

    g. Other classifiers - provides additional quantitative results for other classifiers from ImageNet as requested by one of the reviewers,

    h. Other benchmarks - includes quantitative results for CelebA, CelebA-HQ and MNIST with additional details.

4. In the main manuscript:

    a. the related works section now includes more details about the relationship of ACE and FastDiME to our method, and some initially missing abbreviations,

    b. the definition of RVCEs in the Method section was clarified to explicitly indicate that the region does not have to depend on the classifier,

    c. the claim about increased trust in lines 417-418 was replaced with a statement about alignment with human intuition, following the suggestion of one of the reviewers,

    d. we fixed some typos that were found in the meantime.

We note that the above additions constitute extensions of the content within the initial manuscript and were motivated solely by the reviewers requests and concerns. Despite the late phase of the rebuttal, we are open to further discussion about our work.

---

### Author Response · Authors · 2024-11-28

Due to the extended discussion period, we strongly encourage reviewers to provide feedback on our comments. We are aware of the heavy workload and would be very grateful if you could find the time to address our responses.

---

### Meta-Review · Area_Chair_pbrQ · 2024-12-19

**Metareview:**

The paper initially received mixed reviews, 656653. There were a number of major issues raised:

1. what is the effect if the region is not semantically relevant? [nGZZ]
2. What is the purpose of the three settings A, B, and C [nGZZ]
3. Why do LangSAM-based regions have higher FID, and why is it so difficult for the proposed method? [nGZZ]
4. how is confirmation bias mitigated by the proposed method? [nGZZ, p98R]
5. missing discussion about prior works [2FX9]
6. needs more evaluation (both datasets and baselines) [2FX9]
7. overclaims need to be tempered [2FX9]
8. there is a risk when constraining the explanation to semantically meaningful variations that the explanation will not show the actual decision process [2FX9]
9. does using attribution maps or grouping methods bias the understanding of the classifier? [PMZS]
10. the explanation ability entangles the ability of the classifier and that of the pinpointing model, which is problematic for understanding the classifier [PMZS]
11. difference between attribution map generation (IG) and visual counterfactuals is unclear. [PMZS]
12. what happens when we force the label to a different class? [PMZS]
13. can the method discover cases where unusual visual cues are used, e.g., the background context is use due to dataset bias? [PMZS]
14. How to interpret some examples in Fig 5 where the model's prediction is reasonable and expected? [PMZS]
15. Limited technical novelty [p98R]
16. Cannot capture changes in shape due to inpainting in a specific region. Missing results on less favorable test cases with shape changes [Rswz]
17. unclear whether the proposed method is better, since the best counterfactual image would flip the decision with minimal visible change (which is not quite useful) [Rswz]
18. can it help explain adversarial attacks? [Rswz]
19. FID score is misleading because the proposed method only changes a region [TMN9]
20. missing user study [TMN9]
21. need more use cases of the method to show its usefulness [TMN9]

The authors wrote a response. During the discussion period, 2FX9, nGZZ, p98R were satisfied with the response. TMN9 was also positive, but noted that Point 19 was not properly addressed: "Still, the paper should name that limitation of the metric at least and could have thought of a region-based metric or similar."

PMZS was mostly satisfied but noted there was still a concern about the new user study, which 2FX9 also noted:

- PMZS: "However, I still hold one concern regarding to the root of the method. From the user study, we can conclude that the method can help person to find the most influence region aligned with human's perception. It is doubtful if the method can explain the wrong behavior of the model especially when the model's prediction is influence by some factors similar to adversarial attach. But this might be a problem beyond this manuscript."
- 2Xf9: "I do have some concerns about the interpretation of the newly added user study. .... the constraining of explanations to semantically meaningful variations may induce trust on the user's side that is unwarranted. This is actually reflected well in what the authors write about their user study: "The answers generally focused on the semantic change being more localized, easier to interpret and better aligned with human intuition". The fact that an explanation is 'aligned with human intuition' should be completely irrelevant for evaluating the utility of an explanation method: if the model does something very weird, I want to see that, and not an explanation that looks intuitively pleasing but does not reflect the model's actual decision process well. As far as I understand it, there was no evaluation in this study of whether the users actually understood the model's decision process better, or whether they just felt they did? As it stands then, the user study seems to show that users like these explanations, but not whether the explanations are any good in the sense that they really help them understand the model (better)."

Regarding this concern, the AC noted that it is mainly about the user study where regions are defined by users. For the fully-automated version, then the region limitation inherits the limitations of the attribution method employed.

Finally, Rswz raised concerns about the limitation of the method to explain shape changes or adversarial attacks. The AC thought that the authors responsed well to these questions (Rswz did not participate)

Overall, the paper received positive support with updated scores 866655 with reviewers noting the potential usefulness and good performance, despite its limitations. These limitations are interesting avenues for future work.  The AC agreed and recommends accept. The paper should be further edited to address the discussion points. In particular a summary of the above  limitations should be included in the main paper (not the supplemental), for full disclosure.

**Additional Comments On Reviewer Discussion:**

see above

---

### Decision · Program_Chairs · 2025-01-22

Accept (Poster)